# SpurLens: Automatic Detection of Spurious Cues in Multimodal LLMs

## Abstract

Unimodal vision models are known to rely on spurious correlations, but it remains unclear to what extent Multimodal Large Language Models (MLLMs) exhibit similar biases despite language supervision. In this paper, we investigate spurious bias in MLLMs and introduce SpurLens, a pipeline that leverages GPT-4 and open-set object detectors to automatically identify spurious visual cues without human supervision. Our findings reveal that spurious correlations cause two major failure modes in MLLMs: (1) over-reliance on spurious cues for object recognition, where removing these cues reduces accuracy, and (2) object hallucination, where spurious cues amplify the hallucination by over 10x. We investigate various MLLMs and datasets, and validate our findings with multiple robustness checks. Beyond diagnosing these failures, we explore potential mitigation strategies, such as prompt ensembling and reasoning-based prompting, and conduct ablation studies to examine the root causes of spurious bias in MLLMs. By exposing the persistence of spurious correlations, our study calls for more rigorous evaluation methods and mitigation strategies to enhance the reliability of MLLMs.

## 1 Introduction

Multimodal large language models (MLLMs) (Wang et al., 2024; Liu et al., 2024a; Meta, 2024; OpenAI, 2024a) have seen rapid advances in recent years. These models leverage the powerful capabilities of large language models (LLMs) (OpenAI, 2024b; Touvron et al., 2023) to process diverse modalities, such as images and text. They have demonstrated significant proficiency in tasks such as image perception, visual question answering, and instruction following.

Despite these advancements, MLLMs still exhibit critical visual shortcomings (Tong et al., 2024a;b). One such failure is object hallucination (Li et al., 2023; Hu et al., 2023; Lovenia et al., 2023; Leng et al., 2024), where MLLMs generate semantically coherent but factually incorrect content, falsely detecting objects that are not present in the input images, as exemplified in Figure 1. We hypothesize that many of these failures stem from a well-known robustness issue in deep learning models: spurious bias – the tendency to rely on non-essential input attributes rather than truly recognizing the target object (Ye et al., 2024). While spurious correlation reliance has been well-documented in single-modality image classifiers, the extent to which it persists in MLLMs remains unclear.

Understanding spurious correlations is crucial because they can lead to systematic failures. Consider the case of a fire hydrant (Figure 2). When a fire hydrant appears in a street scene, Llama-3.2 (Meta, 2024) correctly recognizes it 96% of the time. However, when placed in unusual contexts, such as a warehouse, accuracy drops to 83%, suggesting an over-reliance on contextual cues rather than the object itself. Conversely, when shown a street scene without a fire hydrant, the model hallucinates its presence 7 times more often than when given a random image without a hydrant, such as an indoor environment lacking street-like features.

In this work, we systematically study spurious bias in MLLMs. We introduce *SpurLens*, a pipeline designed to rank images based on the presence of spurious cues. For each object, we leverage GPT-4 to generate a list of potential spurious cues and then rank images using open-set object detectors that assess the presence of these cues. This ranking enables a fine-grained evaluation of how spurious correlations influence both object recognition and hallucination in MLLMs.

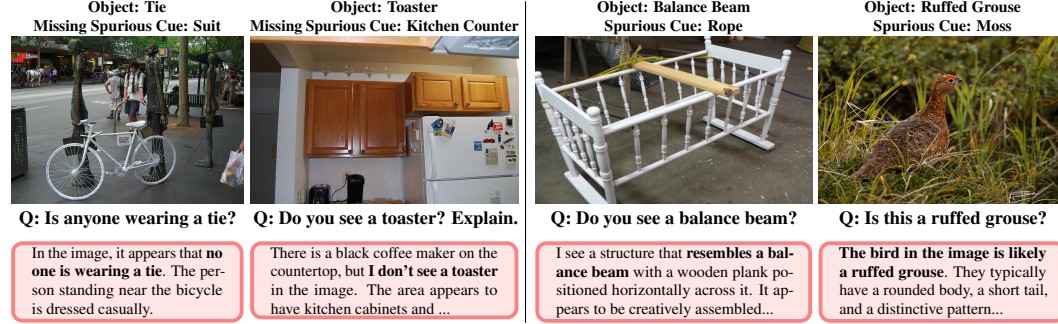

**Object: Tie**
**Missing Spurious Cue: Suit**
**Q: Is anyone wearing a tie?**
In the image, it appears that **no one is wearing a tie**. The person standing near the bicycle is dressed casually.

**Object: Toaster**
**Missing Spurious Cue: Kitchen Counter**
**Q: Do you see a toaster? Explain.**
There is a black coffee maker on the countertop, but **I don't see a toaster** in the image. The area appears to have kitchen cabinets and ...

**Object: Balance Beam**
**Spurious Cue: Rope**
**Q: Do you see a balance beam?**
I see a structure that **resembles a balance beam** with a wooden plank positioned horizontally across it. It appears to be creatively assembled...

**Object: Ruffed Grouse**
**Spurious Cue: Moss**
**Q: Is this a ruffed grouse?**
**The bird in the image is likely a ruffed grouse**. They typically have a rounded body, a short tail, and a distinctive pattern...

Figure 1: Some failures of GPT-4o-mini identified by *SpurLens*. (**Left**) The model fails to recognize objects in the absence of spurious cues. (**Right**) Spurious cues trigger hallucinations.

Our results reveal that MLLMs over-rely on spurious cues, leading to two major failures. The first is *object recognition* failures: when spurious cues are removed, recognition performance significantly drops. For example, GPT-4o-mini (OpenAI, 2024a) sees a 20.4% accuracy drop on COCO (Lin et al., 2014), from 88.0% to 67.6% (Table 1). The second is *object hallucination*: spurious cues trigger hallucinations, causing models to misidentify objects. For GPT-4o-mini, hallucination rates increase 26-fold when spurious cues are present (Table 2). We illustrate some of these failures in Figure 1.

*SpurLens* offers three key advantages over prior work. First, *SpurLens* enables the detection of a broader range of spurious cues as it is not limited to predefined features. Second, it provides natural language descriptors for detected cues, offering insight into model biases. Finally, our method requires no human supervision, making it scalable and adaptable across datasets and objects. We validate the reliability and robustness of *SpurLens* through human studies and analysis of its components. Its outputs align closely with human judgments and remain consistent when using different language models to propose spurious cues, highlighting its generality and reproducibility.

While *SpurLens* allows us to systematically analyze spurious correlations in MLLMs, some key questions remain: from where do these biases originate, and can they be mitigated? To address these, we conduct a series of ablation studies designed to isolate different factors contributing to spurious bias. First, we investigate whether spurious bias persists even when all visual evidence of an object is artificially removed by dropping the corresponding visual tokens. We find that even when target objects have been excised from images, MLLMs tend to hallucinate their presence at a high rate. Next, we explore whether alternative prompting strategies can mitigate spurious reliance. We find that prompt ensembling and reasoning-based prompting offer slight situational improvements but do not significantly reduce spurious bias. Even informing the MLLM of spurious features in the prompt does not meaningfully improve performance, indicating that the spurious biases identified by *SpurLens* are fundamental issues. Finally, we examine whether spurious correlations can be measured purely from the MLLM image embeddings. We find that the vision encoder alone also exhibits spurious biases, reinforcing that the issue extends beyond multimodal fusion errors. These findings indicate that spurious bias is a deeply rooted problem in MLLMs and that simple mitigation strategies are insufficient to fully address it.

## 2 RELATED WORK

**Spurious Correlation**: Spurious correlations have been extensively studied in the context of deep neural network classifiers (e.g., ViT (Alexey, 2020)), with various approaches proposed to detect and mitigate the issue (Sagawa et al., 2019a; Kirichenko et al., 2022; Noohdani et al., 2024). However, these studies primarily focus on uni-modal settings (image classification tasks). Some research (Wang et al.; Varma et al., 2024; Kim et al., 2023) has explored spurious correlations in CLIP (Radford et al., 2021), framing the problem in terms of zero-shot performance across vision and language modalities. Ye et al. (2024) introduces a visual question answering (VQA) benchmark designed to evaluate MLLMs' reliance on spurious correlations using open-source image datasets. Zheng et al. (2024) also proposes a framework to quantify the varying degrees of robustness of Vision-Language Models (used as few-shot image classifiers) against spurious bias.

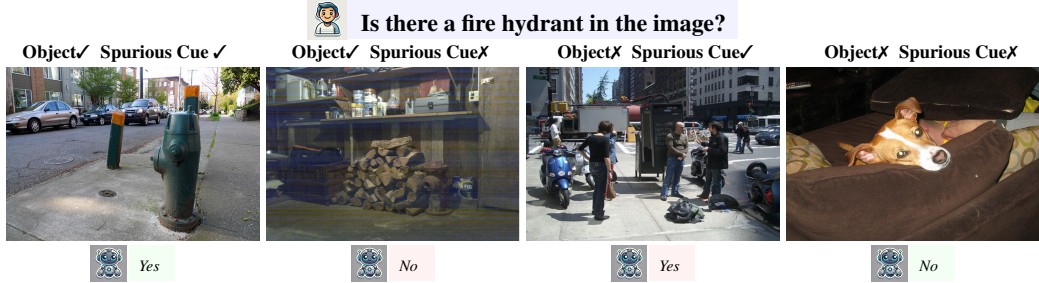

Figure 2: (**Left**) Object Recognition — the model overrelies on spurious cues for recognition. (**Right**) Object Hallucination — spurious cues amplify hallucinations.

**Interpretable Spurious Biases**: Some works investigate language-interpretable and semi-automatic detection of spurious biases in vision models. Eyuboglu et al. (2022) and Zhang et al. (2023) both identify interpretable high-error slices using cross-modal embeddings; however, the former requires additional feature annotations on each sample, while the latter requires human selection of potential text attributes. Wiles et al. (2023) uses clustering of failure cases and image captioning to identify interpretable spurious features; however, they use image generation models to produce synthetic datasets for iterative refinement. In contrast to these methods, *SpurLens* does not require human supervision, and consistently identifies strong failures in real-world image datasets.

**Ranking Images by Spuriosity**: One approach to detecting spurious bias in image classifiers is to rank images within their classes based on spuriosity, the degree to which common spurious cues are present. Prior work, such as HardImageNet (Moayeri et al., 2022) and Moayeri et al. (2023), utilized deep neural features from an interpretable network combined with human supervision to identify these cues. In contrast, *SpurLens* eliminates the need for human supervision by leveraging object detectors to automatically rank images based on spurious cues.

**Failures of Multimodal Systems**: Some studies have introduced frameworks to automatically identify critical shortcomings of MLLMs (Tong et al., 2024a;b). Tong et al. (2024b) highlight MLLMs' struggles with basic visual understanding, attributing these issues to weaknesses in CLIP-based vision encoders. Conversely, Tong et al. (2024a) focuses on the language modality.

**Object Hallucination**: Object hallucination is an actively studied topic in MLLMs (Hu et al., 2023; Leng et al., 2024; Lovenia et al., 2023; Li et al., 2023; Zhou et al., 2023). POPE (Li et al., 2023) presents the first systematic study on object hallucination in MLLMs, providing an evaluation framework to quantify hallucination. Similarly, Leng et al. (2024) provides additional evidence of object hallucination in MLLMs. Lovenia et al. (2023) argues that hallucination should be distinguished from general incorrectness, introducing a separate benchmark for evaluating object hallucination in MLLMs. Hu et al. (2023) proposes CIEM, an automatic pipeline that leverages an annotated image-text dataset along with an LLM to generate factual and contrastive question-answer pairs for evaluating hallucination in MLLMs. The authors also use CIEM to instruction-tune MLLMs in an attempt to mitigate hallucination. Additionally, Zhou et al. (2023) introduces LURE, a post-hoc method that reduces object hallucination in MLLMs by reconstructing less hallucinatory descriptions. We emphasize that, although our evaluation framework follows POPE (Li et al., 2023) and we adopt the evaluation metrics from Leng et al. (2024), our study is not focused on evaluating object hallucination in MLLMs. The key difference between our work and the studies in this section is that we evaluate the reliance of MLLMs on spurious biases, rather than object hallucination itself.

## 3 PROBLEM SETTING

We briefly present our theoretical framework for studying spurious bias in a multimodal setting. We adopt the common setup described in Ye et al. (2024), where $\mathcal{X}$ and $\mathcal{Y}$ represent the vision and language modalities, respectively. Given an image input $x \in \mathcal{X}$ and a text prompt $y \in \mathcal{Y}$, a MLLM learns the mapping $\phi : \mathcal{X} \times \mathcal{Y} \to \mathcal{O}$ such that $o = \phi(x, y)$, where $o \in \mathcal{O} \subset \mathcal{Y}$ denotes the response.

To study spurious biases, for an object, we define $\mathcal{C} \subset \mathcal{X}$ as the set of images that include target object $t$, and $\mathcal{S} \subset \mathcal{X}$ as the set of images containing spurious cue $f$ associated with that object. We can

partition $\mathcal{X}$ into 4 subsets: Object with Spurious Cue ($\mathcal{C} \cap \mathcal{S}$), Object without Spurious Cue ($\mathcal{C} \cap \mathcal{S}^c$), Spurious Cue without Object ($\mathcal{C}^c \cap \mathcal{S}$), and Baseline (Object and Spurious Cue Free; $\mathcal{C}^c \cap \mathcal{S}^c$). For example, if $t =$"fire hydrant" and $f =$"road", $\mathcal{C}$ is all images including a fire hydrant, and $\mathcal{S}$ is all images including a road. For prompt $y =$"Is there a fire hydrant in the image?", the correct response $o$ is "yes" for $x \in \mathcal{C}$ and "no" for $x \notin \mathcal{C}$. Examples of all four types of images are found in Figure 2.

Note that spurious attributes may also exist in the language modality. In our experiments, we minimize their impact by averaging over multiple simple binary prompts. Moreover, due to the flexibility of language and the extensive amount of text data, spurious attributes in the text modality often exhibit only weak correlations with specific responses, such as "yes" or "no" (Ye et al., 2024).

Similar to Leng et al. (2024), we used two core metrics for our experiments: **Perception Accuracy (PA)** and **Hallucination Rate (HR)**. Formally:

$$\textbf{PA} := p(o = \text{'yes'} \mid x \in \mathcal{C}, y), \quad \textbf{HR} := p(o = \text{'yes'} \mid x \in \mathcal{C}^c, y) \tag{1}$$

**PA** measures the model's ability to accurately perceive objects that are present, and **HR** quantifies the hallucination of non-existent objects. Higher **PA** scores indicate better perception, while lower **HR** scores reflect greater robustness against object hallucinations. To study spurious bias, we analyze how **PA** and **HR** change with and without spurious cues for a given target object $t$. Formally, for object recognition the Spurious Gap (Moayeri et al., 2023) is defined as $\textbf{PA Gap} := \textbf{PA}_s - \textbf{PA}_c$, where:

$$\textbf{PA}_s := p(o = \text{'yes'} \mid x \in \mathcal{C} \cap \mathcal{S}, y) \qquad \textbf{PA}_c := p(o = \text{'yes'} \mid x \in \mathcal{C} \cap \mathcal{S}^c, y) \tag{2}$$

Similarly, for object hallucination, the Spurious Gap is defined as $\textbf{HR Gap} := \textbf{HR}_s - \textbf{HR}_c$, where:

$$\textbf{HR}_s := p(o = \text{'yes'} \mid x \in \mathcal{C}^c \cap \mathcal{S}, y) \qquad \textbf{HR}_c := p(o = \text{'yes'} \mid x \in \mathcal{C}^c \cap \mathcal{S}^c, y) \tag{3}$$

A high Spurious Gap indicates that the MLLM is highly dependent on spurious cues to identify the presence of object $t$. Appendix B provides further theoretical intuition for these choices of metrics. We believe that modern MLLMs are susceptible to spurious correlations, and thus exhibit high PA and HR Gaps in many cases. We estimate these values by applying selected MLLMs to images from all four partitions of $\mathcal{X}$, once they have been identified.

## 4 SpurLens: Automating Discovery of Spurious Cues

The spuriosity rankings in HardImageNet (Moayeri et al., 2022; 2023) are constructed using neural features extracted from Salient ImageNet (Singla & Feizi, 2022). This process relies on human supervision to identify spurious features for each class. However, it is limited in scope, detecting only a few spurious features per class. Notably, for nearly two-thirds of ImageNet classes, no spurious cues were detected at all (Moayeri et al., 2023). To study spurious correlations in MLLMs across a broader range of objects and datasets, we develop *SpurLens*, a pipeline designed to produce interpretable spuriosity rankings of images, from which the Spurious Gap can be estimated (outlined in Figure 3).

Suppose that, for a given MLLM $\mathcal{M}$ and target object $t$, we aim to identify spurious cues for $t$. Assume that we have access to a large dataset of images, denoted as $\{\mathcal{I}_j\}_{j=1}^N$, and that the presence of $t$ in each image is known; if we are testing object recognition, all images must contain the target object $t$; if we are testing object hallucination, all images must not contain $t$.

**Proposing Spurious Features** We use GPT-4 to generate a list of objects or background elements that commonly appear in images of $t$. We lemmatize each suggested object, remove duplicates, and remove features that share words with target object name. We then ask GPT-4 several filtering questions to ensure that the proposed objects match the qualifications for being spurious features, and to preemptively eliminate objects that may be challenging for the object detector (e.g., features that are not easily detectable such as "sunlight", or features that are part of the corresponding target object such as "screen" for "laptop"). Note that this filtering reduces the set of potential spurious features in exchange for improved robustness in object detection. See Appendix A for details of these prompts.

Works such as Leng et al. (2024); Zhou et al. (2023); Kim et al. (2024) identify spurious correlations through the frequent co-occurrence of objects in MLLM-generated image captions. Our method avoids this computational cost, and the easily-modifiable prompt structure may suggest a more diverse pool of potential spurious objects.

**Identifying Spurious Objects** To identify the presence of these spurious features $f_i$ in the images $\mathcal{I}_j$, we use the OWLv2 open-set object detector Minderer et al. (2024). For each image, we query

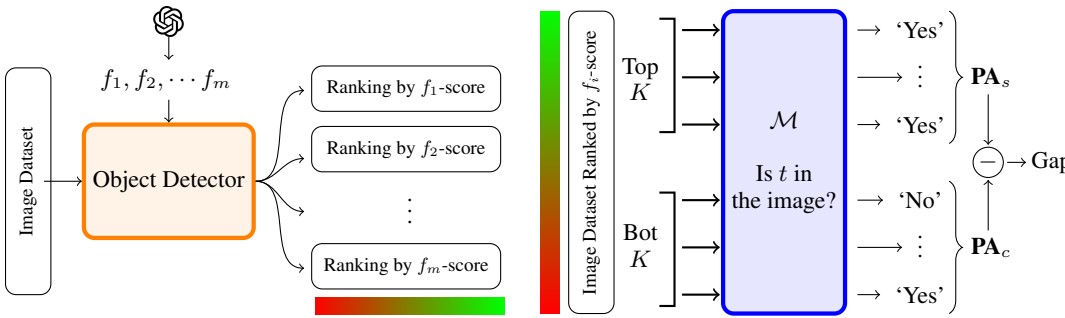

Figure 3: **An overview of** *SpurLens*. **Left**: GPT-4 is used to propose (and filter) spurious features $f_i$. OWLv2 annotates an image dataset with these spurious features, and its scores are used to rank the image dataset for each feature. **Right**: for a given ranking, the top-$K$ and bottom-$K$ images are passed to an MLLM with a prompt. The responses are aggregated to compute the Spurious Gap.

OWLv2 with all potential spurious features and obtain several triplets consisting of a bounding box $b \in [0,1]^4$, label $f_i$, and confidence score $c \in [0,1]$. Let $\mathcal{O}(\mathcal{I}_i)$ denote the set of such triplets produced by OWLv2 for image $\mathcal{I}_i$. We define the $f_i$-score of $\mathcal{I}_j$ as

$$S(f_i, \mathcal{I}_j) := \max\left(\{0\} \cup \{c : (b, f_i, c) \in \mathcal{O}(\mathcal{I}_j)\}\right) \tag{4}$$

For each potential spurious feature $f_i$, we sort the images by $f_i$-score to obtain a ranking. Validation of object detection results is provided in Appendix F.

**Spurious Gaps** For each ranking corresponding to feature $f_i$, let $\mathcal{U}^+_{t,f_i}, \mathcal{U}^-_{t,f_i} \subset \{\mathcal{I}_j\}_{j=1}^N$ be the images with the $K$-highest and $K$-lowest $f_i$-scores respectively. For each of these images, we apply the model $\mathcal{M}$ paired with three prompts $p_k(t)$, $1 \leq k \leq 3$ (see Appendix A for details). Each prompt asks $\mathcal{M}$ if it sees the target object $t$ in the image, and elicits a Yes/No response; we use three prompts to mitigate the bias due to word choice. We define **PA** (or similarly **HR**) of $\mathcal{M}$ on image $\mathcal{I}$ as:

$$\mathbf{PA}(\mathcal{M}, \mathcal{I}, t) = \frac{1}{3} \sum_{k=1}^3 \mathbb{1}\left(\mathcal{M}(\mathcal{I}, p_k(t)) = \text{"Yes"}\right) \tag{5}$$

We then estimate the Spurious Gap as

$$\mathbf{PA}_s = \frac{1}{K} \sum_{\mathcal{I} \in \mathcal{U}^+_{t,f_i}} \mathbf{PA}(\mathcal{M}, \mathcal{I}, t), \ \mathbf{PA}_c = \frac{1}{K} \sum_{\mathcal{I} \in \mathcal{U}^-_{t,f_i}} \mathbf{PA}(\mathcal{M}, \mathcal{I}, t), \ \mathbf{PA \ Gap} = \mathbf{PA}_s - \mathbf{PA}_c \tag{6}$$

In the object recognition case, the Spurious Gap measures the difference in recognition accuracy between images with and without $f_i$, based on the top-$K$ and bottom-$K$ ranked images in the $f_i$-score ranking. A large positive PA Gap indicates that $f_i$ is truly spurious for $t$, as the model recognizes $t$ better when the spurious feature is present. In the object hallucination case, we apply the same methodology but to images that do not contain $t$. Here, a positive HR Gap means that the model hallucinates the presence of $t$ more frequently when $f_i$ is present, indicating that $f_i$ acts as a spurious cue that triggers hallucination. After computing the Gap for all potential spurious features, we select the feature with the largest Gap as the most influential spurious cue.

**Validation** In Appendix E, we compare GPT-4's spurious feature proposals with other LLMs (e.g. Gemini) and observe substantial overlap, suggesting that our results are not dependent on the choice of language model. In Appendix I, we study the sensitivity of the computed Gaps to the choice of $K$, and find that the Gaps stabilize for all models as $K$ increases, indicating the *SpurLens* results are robust to this parameter. In Appendix F, we assess the accuracy of the OWLv2 object detector with two human studies. First, we randomly sample detection outputs and check whether human annotators agree with the presence/absence of the spurious cue in the top- and bottom-ranked images; we find high agreement: 93% on COCO and 90% on ImageNet. Second, we for a random subset of spurious cues identified by *SpurLens*, we replace the object detector results with human annotations, and compute Gaps based on the human-labeled top- and bottom-ranked images; we observe strong alignment with the *SpurLens* gaps. This demonstrates that, while the object detector may itself be subject to some spurious biases, our methodology is sufficiently robust to enable reliable detection of large spurious gaps. In Appendix N, we compare *SpurLens* gaps to a random baseline, providing evidence that the signals *SpurLens* detects are statistically significant.

Finally, we note that *SpurLens* emphasizes precision, rather than recall or completeness: it aims to consistently and robustly find strong spurious cues, rather than the absolute best cue for each model and dataset. While our results cannot be used to directly compare models, we do find significant spurious biases in all models that we evaluate. Although *SpurLens* uses binary-resemblance prompts to evaluate MLLMs (for ease of evaluation and minimizing other sources of bias), we find many instances of the identified biases manifesting in open-ended generation (see Appendices C, D, and R).

## 5 RESULTS

**Models** For our experiments, we evaluate three open-source MLLMs, Qwen2-VL-7B-Instruct (Wang et al., 2024), Llama-3.2-11B-Vision-Instruct (Meta, 2024), and LLaVA-v1.6-mistral-7B (Liu et al., 2023), along with one closed-source model, GPT-4o-mini (OpenAI, 2024a).

**Datasets** We use three image datasets in our evaluation: HardImageNet (Moayeri et al., 2022), ImageNet (Russakovsky et al., 2015), and COCO (Lin et al., 2014). To maintain comparability with prior work, we use a subset of 100 classes from ImageNet as chosen in Spurious ImageNet (Neuhaus et al., 2023). These datasets offer diverse visual contexts, enabling analysis of spurious correlations across different domains. They also vary in complexity: HardImageNet is curated to highlight spurious correlations, while COCO contains diverse real-world scenes, providing a broader evaluation of MLLM robustness. In contrast, older benchmarks such as CelebA (Liu et al., 2015) and Waterbirds (Sagawa et al., 2019b) were designed for unimodal classifiers trained end-to-end, and thus fail to challenge MLLMs in a zero-shot setting. See Appendices G and H for further discussion.

**Evaluation** For each class in all three datasets, we initially generate 32 potential spurious features, which are then passed through various filters (see Appendix A). We compute the PA/HR Gap for each feature, and choose the most influential one as the *SpurLens* feature for that class. For each dataset, we take the class-wise average of the chosen PA/HR Gaps and report the final aggregated metrics.

### 5.1 OBJECT RECOGNITION

The results are presented in Table 1. We use $K = 50$ for HardImageNet and the ImageNet subset, and $K = 100$ for COCO. We observe that performance decreases across all models when spurious cues are absent, with the magnitude of this drop varying between models and datasets. Some qualitative examples of GPT-4o-mini failures are provided in Appendix D. Additional qualitative examples are found in Appendix R, which compares the behavior of open-source models with and without spurious cues. While the models generate detailed and accurate descriptions when spurious cues are present, the descriptions become less accurate when they are absent.

We conduct a class-wise analysis of spurious bias: the distribution of PA across ImageNet and COCO classes for Qwen2-VL (Wang et al., 2024) is visualized in Figure 5 (see Appendix Q for other models). When spurious cues are absent, accuracy decreases and variance increases, indicating that models struggle to generalize without these cues. The distributions also confirm that spurious bias is highly class-dependent. We provide a detailed breakdown of the strongest spurious cues and their corresponding Spurious Gaps as identified by *SpurLens*, for each model and class, in Appendix S.

### 5.2 OBJECT HALLUCINATION

We visualize an example of a spurious cue leading to hallucination, as identified by *SpurLens*, in Figure 4. Given an MLLM and a large image dataset, *SpurLens* detects the most influential spurious cues based on their Spurious Gap. For LLaVA (Liu et al., 2024a) with target object "bird", *SpurLens* identifies "feeder" as the most influential spurious cue. In Figure 4, we demonstrate that this cue can indeed trigger hallucinations in the model. We provide more qualitative examples in Appendix C.

Recall that to compute HR Gaps with *SpurLens*, for each class we require an image dataset that does not contain the target object. COCO is a large-scale object segmentation dataset, whose 80 categories are grouped into 10 supercategories. For each target object, we run *SpurLens* on images from the same supercategory that do not contain the target object. We exclude three COCO classes (keyboard, dining table, and sports ball) from our final analysis because these objects frequently appear in the dataset without being annotated, making HR measurements unreliable for these categories. For ImageNet, for each of the 100 classes, we randomly sample 5,000 images outside of the class to serve

Table 1: **Object Recognition.** Perception Accuracy (PA%) across different datasets and models. $PA_s$ and $PA_c$ are accuracy on high-spurious and spurious-free images. Results are averaged over classes.

| Dataset | HardImageNet | | | ImageNet Subset | | | COCO | | |
|---|---|---|---|---|---|---|---|---|---|
| Model | $PA_s$ | $PA_c$ | Gap | $PA_s$ | $PA_c$ | Gap | $PA_s$ | $PA_c$ | Gap |
| Qwen2-VL | 99.3 | 93.4 | 5.9 | 98.4 | 91.7 | 6.7 | 95.3 | 80.1 | 15.2 |
| Llama-3.2 | 93.1 | 78.1 | 15.0 | 90.6 | 72.9 | 17.7 | 95.0 | 80.0 | 14.8 |
| LLaVA-v1.6 | 96.0 | 82.8 | 13.2 | 93.0 | 77.3 | 15.7 | 95.6 | 79.6 | 16.0 |
| GPT-4o-mini | 95.0 | 82.4 | 12.6 | 95.7 | 83.8 | 11.9 | 88.0 | 67.6 | 20.4 |

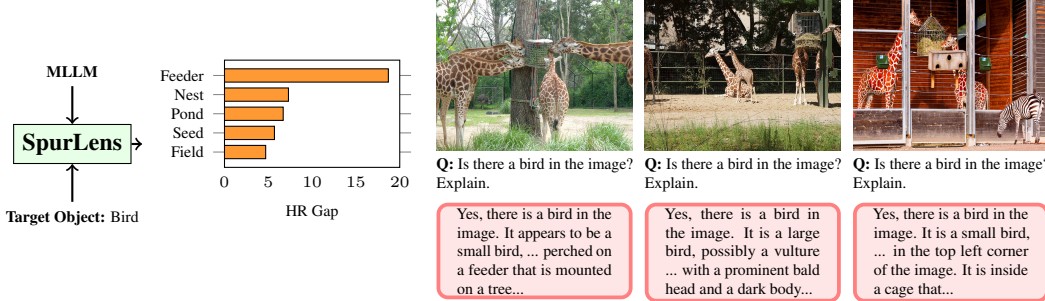

Figure 4: An example hallucination identified by *SpurLens*: (**Left**) *SpurLens* ranks the most influential spurious cues. (**Right**) The spurious cue, 'feeder' in this case, triggers hallucination in the model.

as our dataset. We compute rankings with $K = 50$ for ImageNet and $K = 100$ for COCO. As an ablation, we provide an alternative setup using these two datasets in Appendix L.

The quantitative results are presented in Table 2. We observe that for both COCO and ImageNet, the hallucination rate is significantly higher when spurious cues are present. In COCO, the class-wise averaged HR is at least 5 times higher in high-spurious images across all models; the effect is more pronounced in ImageNet, with it being at least 18 times higher. These results demonstrate that spurious bias makes MLLMs vulnerable to hallucination, as models fail to distinguish between true object presence and misleading contextual cues.

Similar to PA, we conduct a class-wise analysis, to examine hallucination across objects. The distribution of HR for Qwen2-VL is visualized in Figure 5 (see Appendix Q for other models). When spurious cues are present, hallucinations become significantly more pronounced. HR varies across classes, indicating that some objects are more susceptible to hallucination than others. We provide detailed class-wise results, in Appendix S.

## 6 ABLATION STUDIES

### 6.1 DROPPING VISUAL CUES

Both HardImageNet (Moayeri et al., 2022) and COCO (Lin et al., 2014) have pixel-level mask annotation of the target object. We leverage these to create artificial negative samples by removing visual cues of the target object and and evaluate to what extent the models hallucinate their presence.

All open-source MLLMs have a vision encoder that patchifies the image, tokenizes each patch, and processes the resulting visual tokens. We drop the tokens associated with the target object before they are processed by the vision encoder (Jain et al., 2022). Since different MLLMs have varying architectures, we implement this token-dropping approach separately for each model; see Appendix K for implementation details. We also test hallucination rates on blank (fully black) images, where all visual cues are removed.

The results for this experiment are shown in Table 3. We observe significantly higher HR rates compared to Table 2. This poor performance likely arises because these images are out-of-distribution

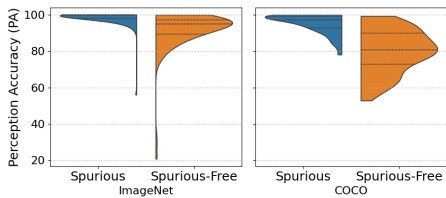 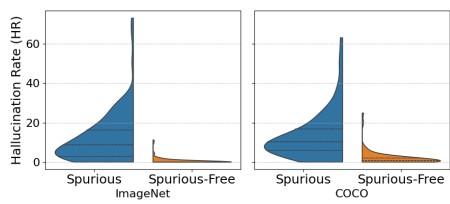

Figure 5: Per-class PA and HR distributions for Qwen2-VL across two datasets.

Table 2: Hallucination rates (HR %) for high-spurious vs spurious-free images. When spurious cues are present, hallucination rates increase.

| Model | ImageNet Subset | | | COCO | | |
|---|---|---|---|---|---|---|
| | $HR_s$ | $HR_c$ | Gap | $HR_s$ | $HR_c$ | Gap |
| Qwen2-VL | 12.1 | 0.5 | 11.6 | 13.5 | 1.8 | 11.7 |
| Llama-3.2 | 9.0 | 0.5 | 8.6 | 18.1 | 3.8 | 14.4 |
| LLaVA-v1.6 | 18.4 | 1.0 | 17.4 | 21.8 | 2.7 | 19.2 |
| GPT-4o-mini | 5.2 | 0.2 | 5.1 | 11.2 | 1.4 | 9.8 |

Table 3: Hallucination rates (HR %) after object token-dropping. Models hallucinate even when target object features are removed.

| Model | HardImageNet | | | COCO | | |
|---|---|---|---|---|---|---|
| | $HR_s$ | $HR_c$ | $HR_b$ | $HR_s$ | $HR_c$ | $HR_b$ |
| Qwen2-VL | 50.2 | 37.2 | 6.6 | 35.3 | 15.1 | 5.5 |
| Llama-3.2 | 31.5 | 20.4 | 4.9 | 34.8 | 15.1 | 4.0 |
| LLaVA-1.6 | 45.7 | 34.6 | 0.0 | 40.0 | 16.4 | 0.0 |

(OOD) samples, making them more challenging for the models. However, even under these OOD conditions, we observe that HR remains higher when spurious cues are present, reinforcing that models strongly rely on spurious correlations when the target object is absent. We also find that $HR_b$, the hallucination rate on blank (fully black) images, is not always zero, meaning some models hallucinate objects even when given a completely blank input. This highlights that spurious bias in MLLMs is not solely dependent on visual features.

## 6.2 MITIGATION

*SpurLens* is developed as a diagnostic tool that automatically identifies interpretable causes of spurious bias, which can be used for mitigation. However, in this section, we establish that the spurious biases identified are a fundamental issue and cannot be trivially overcome through language-based techniques. We focus on Chain-of-Thought (CoT) prompting, which encourages language models to generate rationales for their solution to a base prompt. Zero-shot CoT has previously been shown to improve performance on various benchmarks in both textual and multimodal scenarios (Wei et al., 2023; Kojima et al., 2022; Luo et al., 2024; Zhang et al., 2024).

We evaluate five strategies. (1) Prompt Ensembling, where we take the majority vote among multiple queries with different prompt phrasing. (2) Dual Prompting, where we first ask for a general image description before prompting for object recognition. (3) Guiding Prompting, where we explicitly instruct the model to focus on key details and avoid misleading background cues. (4) Spurious List, where we provide the model with a list of potential spurious features (the list of GPT-4 suggested features that *SpurLens* analyzes). (5) Spurious Top, where we provide the model with the strongest spurious feature as identified by *SpurLens* (note that this is improper, as it incorporates *SpurLens*' knowledge of the full dataset). The prompts used in these strategies are provided in Appendix A.

The results are provided in Table 4. PA is evaluated on HardImageNet, while HR is evaluated on Spurious ImageNet (Neuhaus et al., 2023). While the ensemble/reasoning-based strategies offer slight improvements on certain models and metrics, none of the strategies perform significantly better or have substantially smaller Spurious Gap compared to the baseline. The Spurious Top strategy "cheats" compared to the other strategies by using *SpurLens*' analysis and identification of the strongest spurious cue on each class' dataset, which is the same dataset the model is being evaluated on; yet, it also does not perform well relatively. Further discussion of these strategies and their relative performance can be found in Appendix J. The continued presence of the Spurious Gap across all strategies suggests that language-based methods, while influencing the model's behavior, cannot eliminate the underlying reliance on spurious cues; Spurious Gaps detected by *SpurLens* are a more fundamental issue that cannot be resolved solely through prompting techniques.

Table 4: **Mitigation Attempt via Prompting.** Perception Accuracy (PA%) and Hallucination Rate (HR%) across different prompting strategies and models. The Spurious Gap persists in all strategies.

| Model | Metric | Baseline | Ensemble | Guiding | Dual | Spur. List | Spur. Top |
|-------|--------|----------|----------|---------|------|-----------|-----------|
| Qwen2 | $PA_s$ | 98.1 | **98.5** | 97.5 | 95.2 | 96.5 | 97.0 |
|       | $PA_c$ | 91.9 | **92.4** | 89.3 | 82.3 | 85.9 | 88.9 |
|       | $HR_s$ | 9.5 | 8.6 | 16.4 | **4.0** | 5.5 | 4.8 |
|       | $HR_c$ | 1.2 | 1.0 | 0.8 | **0.4** | 0.5 | 0.8 |
|       | $HR_b$ | 3.3 | **0** | **0** | **0** | **0** | 1.0 |
| Llama | $PA_s$ | 95.1 | **96.8** | 95.6 | 95.2 | 73.3 | 95.2 |
|       | $PA_c$ | 84.7 | **87.0** | **87.0** | 84.4 | 63.2 | 85.6 |
|       | $HR_s$ | 11.0 | **7.5** | 15.1 | 7.6 | 4.8 | 10.3 |
|       | $HR_c$ | 2.2 | 1.5 | 4.1 | 3.1 | **1.0** | 2.0 |
|       | $HR_b$ | 7.0 | **0** | 4.7 | **0** | 0.7 | 0.6 |
| LLaVA | $PA_s$ | 91.4 | 92.4 | **99.2** | 89.3 | 94.8 | 89.6 |
|       | $PA_c$ | 85.2 | 86.4 | **91.2** | 72.0 | 81.2 | 82.7 |
|       | $HR_s$ | 12.5 | 11.0 | 24.6 | **5.6** | 11.1 | 9.5 |
|       | $HR_c$ | 2.1 | 1.9 | 5.8 | **0.8** | 1.6 | 1.4 |
|       | $HR_b$ | **0** | **0** | **0** | **0** | **0** | **0** |

## 6.3 EXAMINING SPURIOUSITY IN THE VISION ENCODER

As suggested in Tong et al. (2024a;b), MLLMs' poor performance on perception tasks may be attributed to poor information or visual ambiguity in the vision encoder features. To study this, we train binary classifiers on vision encoder features and perform the same experiments as before to compute Spurious Gaps for HardImageNet classes. By isolating the vision encoder, we ablate away any biases that could come from the language model component.

For each image in HardImageNet, we obtain a vector by averaging the embeddings of all image patches. For each class, we train a logistic regression binary classifier that determines whether a given embedding is of an image from that class. We evaluate the bias of this classifier by computing the PA Gap with the $K$ most and least-spurious images of that class. This experiment is done for various compositions and sizes of the training dataset; further details and results are found in Appendix M.

The results in this simple setting reaffirm key observations from our main experiments. The spurious biases measured from these linear classifiers are of similar magnitude to those measured from the Qwen2-VL experiments ($\sim 6\%$ class-wise average PA Gap). This suggests the vision embeddings are affected by spurious bias, and that language supervision did not significantly effect our main experimental results. While the training dataset size and composition for the linear classifiers effected the size of the Gap, spurious bias was still present and significant in all cases. Finally, the magnitude of spurious bias is highly class-dependent, which affirms our distributional analysis of the effect.

## 7 CONCLUSION

We present *SpurLens*: an adaptable, automated, and interpretable system for identifying spurious cues for objects. We use *SpurLens* to find spurious cues for ImageNet and COCO classes and quantitatively evaluate the Spurious Gaps caused by the presence/absence of these features for several MLLMs. Our findings show that modern MLLMs are still highly susceptible to spurious correlations. We hope that our process to discover and measure spurious biases guides future developments for effective mitigation and downstream tasks dependent on accurate visual perception.

**Limitations.** Our method assumes that the image dataset being studied is reasonably large and diverse, so that many of the spurious features proposed appear often enough to be evaluated. Filtering the set of proposed spurious features helped to minimize errors in object detection, although it likely decreased the maximum spurious gaps we could measure. While the method is computationally efficient, scaling linearly with dataset size, runtime may become a bottleneck for very large datasets. Finally, *SpurLens* is diagnostic in nature: it reveals when and where models rely on spurious cues but does not identify their root causes or a robust mitigation strategy; this is an avenue for future work.

## REPRODUCIBILITY STATEMENT

All of the open-source models referenced in Section 5 were accessed through HuggingFace. The prompts used for generating the potential spurious features, filtering, and evaluating MLLMs is found in Appendix A. The *SpurLens* code used to generate and evaluate spurious features is submitted as supplementary material, and will be made publicly available and linked in the camera-ready version.

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

# A  PROMPTS

## A.1  EVALUATION

We used the following three prompts for our main experiments when querying the MLLMs.

---

**Prompts to Evaluate Perception/Hallucination for an Image**

- Do you see a CLASSNAME in the image? Answer with 'Yes' or 'No'.
- Is there a CLASSNAME in the image? Answer with 'Yes' or 'No'.
- Determine whether there is a CLASSNAME in the image. Reply with 'Yes' or 'No'.

---

The following prompts were used in Section 6.2.

---

**Prompting Strategies**

**Ensemble:** We performed a majority vote among the following prompts:
**1)** Do you see a CLASSNAME in the image? Answer with 'Yes' or 'No'.
**2)** Is there a CLASSNAME in the image? Answer with 'Yes' or 'No'.
**3)** Determine whether there is a CLASSNAME in the image. Reply with 'Yes' or 'No'.

**Guiding:**
Do you see a CLASSNAME in the image? Describe all objects in the image. Pay attention to key details that confirm their presence. Be mindful of misleading background features, but do not ignore real objects. Finally, answer with 'Yes' or 'No'.

**Dual:** We first ask the model for a general image description before prompting for object detection

1. Describe the most prominent objects in this image.
2. Is there a CLASSNAME in the image? Answer with 'Yes' or 'No'.

**Spurious List:** Do you see a CLASSNAME in the image? Describe all objects in the image. Pay attention to key details that confirm their presence. Be mindful of misleading background features, but do not ignore real objects. For example, spurious cues like CUES_LIST may appear but are not directly related to the CLASSNAME. Focus on distinguishing the CLASSNAME from such irrelevant features. Finally, answer with 'Yes' or 'No'.

**Spurious Top:** Is there a CLASSNAME in the image? Be aware that the presence or absence of a STRONGEST_CUE does not necessarily indicate the presence or absence of a CLASSNAME. Answer with 'Yes' or 'No'.

---

## A.2  FEATURE GENERATION

We used the following prompts with GPT-4 to generate potential spurious features for a given class.

---

**Generating Potential Spurious Features**

The prompts began with one of the following two options:

- "List N objects that commonly appear in images of a CLASSNAME."
- "List N background elements that commonly appear in images of a CLASSNAME."

To each of these was appended:
"The objects cannot be part of a CLASSNAME. List exactly one item on a every consecutive line, followed by a period and a one sentence explanation. The object must be physical and discernable in an image. The object name must be less than two words. Do not number the responses. Do not output anything else."

---

We used several prompts with GPT-4 to filter the proposed spurious features, keeping only those features which get the desired answer for all questions. We provide all our prompts below.

---

**Filtering Prompts**

- "Can a FEATURENAME exist without a CLASSNAME?" (Desired answer: "Yes".)
- "Is a FEATURENAME part of a CLASSNAME?" (Desired answer: "No".)
- "Do all or almost all FEATURENAME have a CLASSNAME?" (Desired answer: "No".)
- "Do all or almost all CLASSNAME have a FEATURENAME?" (Desired answer: "No".)

---

Now we eliminate objects that may be challenging for the object detector (for example, features that are not easily detectable such as "sunlight", or features that are physically part of the corresponding target object such as "screen" for "laptop"). We use the following prompts for this matter.

---

**Filtering Prompts: Detectability**

Determine whether the provided object or feature is visualizeable in an image. An object is visualizeable in an image if the object has a physical presence, and it is always clear what pixels in the image comprise the specific object. Be conservative when labeling a feature as not detectable; only do so if you are completely sure. Respond with 'Yes' or 'No' only.

Here are some example responses:
'sunlight': No
'trail': Yes
'walk': No
'fluoride': No
'toothpaste': Yes
'algae': No
'water': Yes

Determine whether the following object or feature is visualizeable:

'SPUR FEATURE':

---

**Filtering Prompts: Vocabulary**

Determine whether the meaning of the provided feature might be too difficult most people to understand without background context. A feature is too difficult if the feature is too niche to a specific context, is very uncommon, or has an unusual spelling. Be conservative in labelling a feature as too difficult; only do so when you are completely sure that most people would not know the correct meaning without additional information. Respond with 'Yes' or 'No' only.

Here are some examples:
'saddle': No
'equine': Yes
'grille': Yes
'trunk': No
'liana': Yes
'vine': No

Determine whether the following feature is too difficult:

'SPUR FEATURE':

**Filtering Prompts: Synonyms**

Determine whether two objects provided are synonyms of each other, or instances of each other. This is not asking whether the two objects are similar. Only answer 'Yes' if the two terms generally refer to the same object. Respond with 'Yes' or 'No' only.

Here are some examples:
'car', 'vehicle': Yes
'truck', 'bumper': No
'surfboard', 'skimboard': Yes
'remote', 'game controller': Yes
'motorcycle', 'bike': Yes
'motorcycle', 'pedal': No
'cradle', 'rocker': Yes
'bed', 'sleeping mat': Yes
'laptop', 'tablet': No
'backpack', 'purse': No

Determine whether the following two objects are synonyms or instances of each other:

'SPUR FEATURE', 'TARGET OBJECT':

**Filtering Prompts: Separable**

Determine whether the two objects provided are inseparable from each other. Answer 'Yes' one of the objects is part of the other, or if they are nearly always found together. This is not asking whether the two objects are similar or related. Only answer 'Yes' if most people could not distinguish between the two objects when shown an example, or whether it is almost impossible to find one object without the other. Respond with 'Yes' or 'No' only.

Here are some examples:
'cell phone', 'screen': Yes
'ski', 'snowboard': No
'bed', 'nightlight': No
'oven', 'stove': Yes
'boat', 'anchor': Yes
'canoe', 'sail': No
'train', 'railroad': Yes
'train', 'traffic signal': No

Determine whether the following two objects are inseparable from each other:

'SPUR FEATURE', 'TARGET OBJECT':

**Filtering Prompts: Composition**

Determine whether the first object is part of the second object. Respond with 'Yes' if the first object is frequently physically attached to the second object, refers to some component of the second object, or is a property of the second object. This is not asking whether the two objects are similar or often seen together. Only answer 'Yes' if you generally cannot have the second object without the first object. Respond with 'Yes' or 'No' only.
Here are some examples:
'power cord', 'hair dryer': Yes
'bookmark', 'book': No
'handlebar', 'bicycle': Yes
'label', 'wine bottle': Yes
'collar', 'dog': No
'drinking glass', 'wine bottle': No
'soil', 'plant pot': Yes
'rod', 'pull-up bar': Yes

Answer for the following object or term.

'SPUR FEATURE', 'TARGET OBJECT':

---

**Filtering Prompts: Confusion**

Determine whether an instance of the first object in a photograph could be easily confused as being the second object type. This is not asking whether the two objects are similar or often seen together. Only answer 'Yes' if the two objects look so similar to each other that most people would not be able to tell the difference between them in an image when viewed from certain angles. Respond with 'Yes' or 'No' only.

Here are some examples:
'knife', 'fork': Yes
'chopstick', 'fork': No
'balloon', 'kite': Yes
'airplane', 'kite': No
'parking space', 'parking meter': No
'parking space', 'parking lot': Yes
'coffee cup', 'cup', : Yes
'straw', 'cup': No
'juice', 'cider': Yes
'barrel', 'composter': Yes
'soil', 'mulch': Yes
'double bass', 'guitar': No

Answer for the following object or term.

'SPUR FEATURE', 'TARGET OBJECT':

---

# B    THEORETICAL INTUITION FOR GAP METRICS

## B.1    PERCEPTION ACCURACY

Our notion of multimodal spurious bias can be considered a relaxation of the framework proposed in Ye et al. (2024); using our notation,

$$p(o = \text{``yes''} \mid x \in \mathcal{C} \cap \mathcal{S}, y) \gg p(o = \text{``yes''} \mid x \in \mathcal{C}, y) \tag{7}$$

Our definition of **PA Gap** $= \mathbf{PA}_s - \mathbf{PA}_c$ from Equation 2 is related to Equation 7 by the law of total probability. To illustrate this, let $\alpha = p(x \in S^c \mid x \in \mathcal{C}, y)$. We derive the relationship as follows:

$$\mathbf{PA}_s - \mathbf{PA} = \mathbf{PA}_s - \alpha\mathbf{PA}_c - (1 - \alpha)\mathbf{PA}_s = \alpha(\mathbf{PA}_s - \mathbf{PA}_c) \tag{8}$$

Thus, the **Spurious Gap** represents the multimodal spurious bias, scaled by $\frac{1}{\alpha}$. Note that $\alpha$ reflects the strength of the spurious correlation. If the spurious cue is almost always present when the object is present, $\alpha$ would approach zero, resulting in the Spurious Gap being much larger than the bias.

## B.2    OBJECT HALLUCINATION

Define $\beta = p(x \in S \mid x \in \mathcal{C}^c, y)$. From definition of **HR** and using the law of total probability, we derive:

$$\mathbf{HR} = (1 - \beta)\mathbf{HR}_c + \beta\mathbf{HR}_s \tag{9}$$

We argue that $\beta$ is near zero because the set $\mathcal{C}^c$ is much larger than $\mathcal{S}$. Additionally, we assume that the model is resistant to object hallucination on baseline images. This assumption is based on the fact that, among the four subsets partitioning $\mathcal{X}$, baseline is the largest. Formally, $\mathbf{HR}_c \approx 0$. Our empirical evidence supports this assumption. Combining these two assumptions with Equation 9, we obtain:

$$\frac{1}{\beta}\mathbf{HR} \approx \mathbf{HR}_s \tag{10}$$

Equation 10 demonstrates that even though the general hallucination rate for a model might be low, the presence of spurious cues can amplify it significantly on a large scale.

## C   QUALITATIVE EXAMPLES OF OBJECT HALLUCINATION

We provided some of the hallucinations identified by *SpurLens* in Figure 6 and Figure 7.

## D   QUALITATIVE EXAMPLES OF GPT-4O-MINI SPURIOUS GAP

We provided some of the object recognition failures identified by *SpurLens* in Figure 8.

## E   CONSISTENCY ACROSS FEATURE-PROPOSAL LLMS

*SpurLens* relies on GPT-4 for generating spurious features for each target object. Our usage of this particular model may incur some bias in what features are proposed, or which target objects get better-aligned features for object detection and recognition tasks. To study this, we experiment with feature generation on two additional LLMs: Qwen2.5 72B, and Gemini 2.5 Flash.

For both the 15 HardImageNet classes and the 79 COCO classes, we generate potential spurious features with both LLMs using the same method as described in Section 4. We use the same prompts to generate the base list of features, and perform basic de-duplication, lemmatization, and filtering using the 4 prompts that determine whether the proposed object follows the definition of a "spurious feature". However, we do not perform the extensive filtering with the prompts that utilize in-context examples, as we wish to study the distribution of features proposed without concern for the object detector.

To compare the lists of features generated by these two alternative LLMs with those generated by GPT-4, we utilize semantic similarity matching. For each dataset class, we take the list $\{f_i\}_{i=1}^n$ generated by GPT-4 and the list $\{\tilde{f}_j\}_{j=1}^m$ generated by the alternative LLM, and compute the pair-wise similarity as the cosine similarity of embeddings using the all-MiniLM-L6-v2 embedding model. For each feature proposed by the alternative LLM, we select highest similarity with respect to all GPT-4 proposed features:

$$S(\tilde{f}_j) = \max_i \cos(f_i, \tilde{f}_j)$$

We wish to examine the distribution of this value within a class. We introduce a thresholded metric, the "proposal similarity at $\alpha$", defined as the proportion of features for which the $S$-value is at least $\alpha$:

$$\text{PS}_\alpha = \frac{1}{m} \sum_{j=1}^m \mathbb{1}\left(S(\tilde{f}_j) > \alpha\right)$$

Empirically, we find that a similarity score above $\alpha = 0.7$ means that the two features are extremely similar apart from some adjective (for example, $\cos(\text{"trees"}, \text{"pine trees"}) = 0.7994$), and a similarity score above $\alpha = 0.5$ means that the objects are similar in subject and closely related. We compute values of $\text{PS}_{0.7}$ and $\text{PS}_{0.5}$ for both alternative LLMs, for all HardImagNet and COCO classes. We summarize the class-wise averaged results in Table 5. This demonstrates significant similarity in the features proposed by the alternative LLMs with those proposed by GPT-4o. The significant overlap we find with other LLMs shows that using GPT-4o does not introduce significant bias in our methodology. Further, since *SpurLens* focuses on precision rather than recall, the diversity of generations is a more significant factor, and is demonstrated to be adequate for this application.

## F   HUMAN STUDY VALIDATION

We conduct two human studies to validate the precision and reliability of the rankings produced by *SpurLens*.

### F.1   AGREEMENT WITH OBJECT DETECTOR

In the first study, we evaluate the quality of the spurious cue detection step. For a randomly selected subset of classes (20 from COCO and 25 from ImageNet), we asked human annotators to label

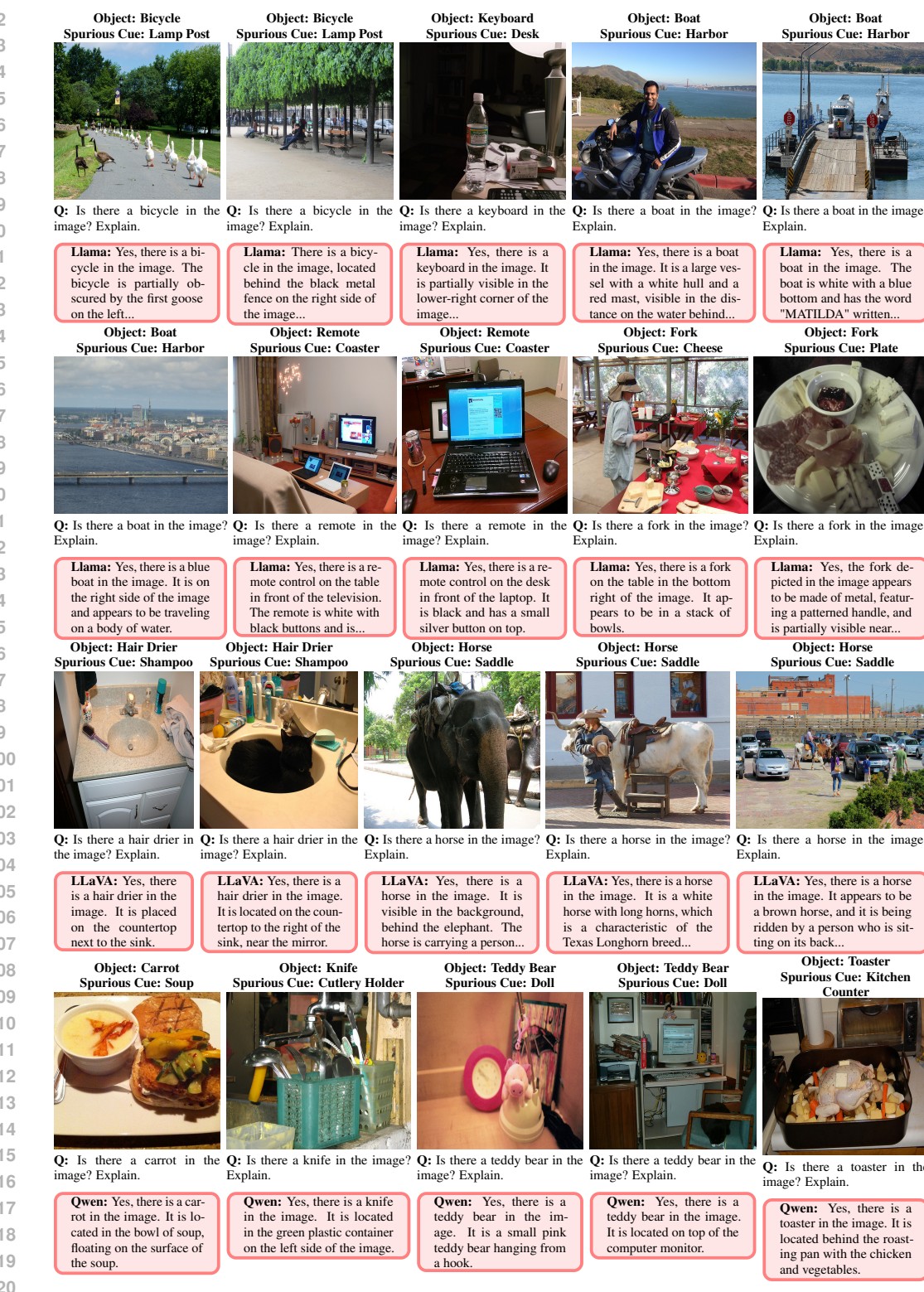

Figure 6: Hallucinations identified by *SpurLens*, part 1

whether the spurious cue was present in the top 10 and bottom 10 images ranked by *SpurLens*. We then compute agreement between the object detector and human judgments. As shown in Table 6,

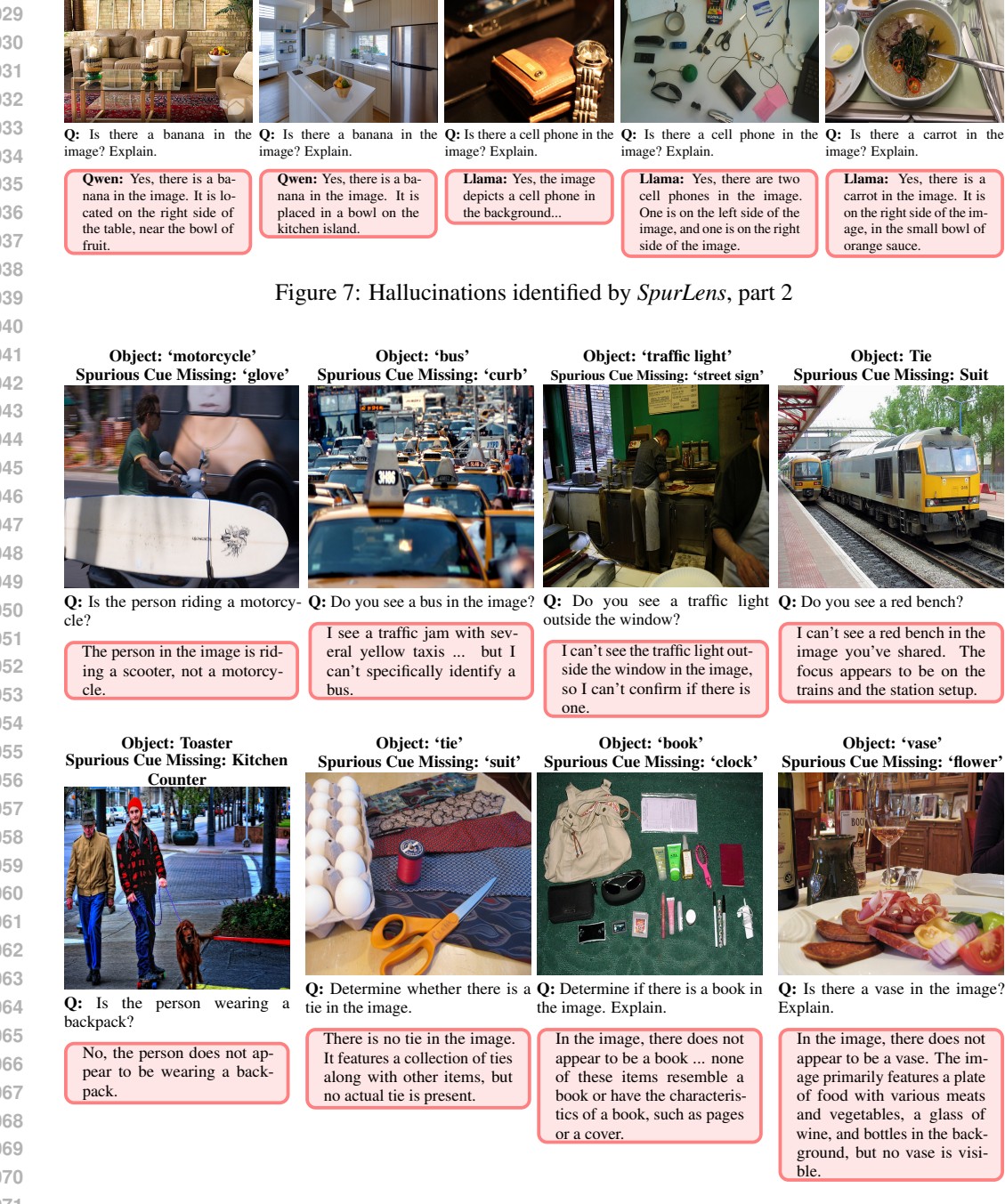

Figure 7: Hallucinations identified by *SpurLens*, part 2

Figure 8: Some failures of GPT-4o-mini (accessed in March 2025) on COCO object perception tasks when spurious features are *not* present.

human agreement is high, suggesting that OWLv2 reliably distinguishes the presence or absence of cues in most cases.

To provide qualitative support, we also visualize example detections. We randomly selected 4 COCO classes and uniformly sampled images from the top- and bottom-ranked sets (based on cue presence) from one of the *SpurLens* rankings. Example detections are shown in Figure 10.

Table 5: Metrics describing the similarity of features proposed by alternative LLMs compared with those generated with GPT-4. All numbers are averaged class-wise. For both models, nearly half of the features are near-equivalent, and the majority are very similar.

| Dataset | HardImageNet | | COCO | |
|---|---|---|---|---|
| Model | $PS_{0.7}$ | $PS_{0.5}$ | $PS_{0.7}$ | $PS_{0.5}$ |
| Gemini 2.5 Flash | 52.4 | 67.0 | 42.9 | 61.9 |
| Qwen 2.5 72B | 49.2 | 69.4 | 42.0 | 62.9 |

| Qwen 2.5 72B | GPT-4o | Gemini 2.5 Flash |
|---|---|---|
| water | water | water |
| rain | rain | rain |
| cloud | cloud | cloud |
| bird | bird | bird |
| leaf | leaf | leaf |
| fruit | fruit | fruit |
| flower | flower | flower |
| branch | tree branch | branch |
| insect | insect | insect |
| tree | tree | tree |
| sunlight | sunlight | forest |
| moss | moss | sunlight |
| soil | fallen logs | sun |
| butterfly | sky | log |
| grass | nest | sky |
| vine | liana | nest |
| fern | rock | liana |
| egg | vine | rock |
| feather | nut | vine |
| sloth | underbrush | nut |
| snake | shade | air |
| dirt | fallen leaves | soil |
| | stream | canopy |
| | banana | dirt |
| | bark | mud |
| | foliage | |
| | bamboo | |
| | coconut | |

Figure 9: Spurious features proposed for HardImagNet class "howler monkey" by Qwen2.5 72B, GPT-4o, and Gemini 2.5 Flash.

It is worth noting that we also experimented with using Grounding DINO (Liu et al., 2024b) as an open-set object detector in place of OWLv2. However, we found that its detections were qualitatively less consistent and reliable. As a result, we use OWLv2 exclusively in our pipeline.

## F.2 AGREEMENT ON SPURIOUS GAP

In the second study, we assess whether the gaps computed by *SpurLens* align with those derived from human-annotated spurious cues. We randomly sampled 20 object-cue pairs from COCO and asked human annotators to label whether the associated spurious feature was present in each image. Using these annotations, we recompute the PA Spurious Gap with $K = 50$, and compare it to the gap produced by *SpurLens* using the same $K$. As shown in Figure 11, the two sets of gap values are highly correlated (Pearson $r = 0.988$), supporting the claim that *SpurLens* produces human-aligned and reliable rankings.

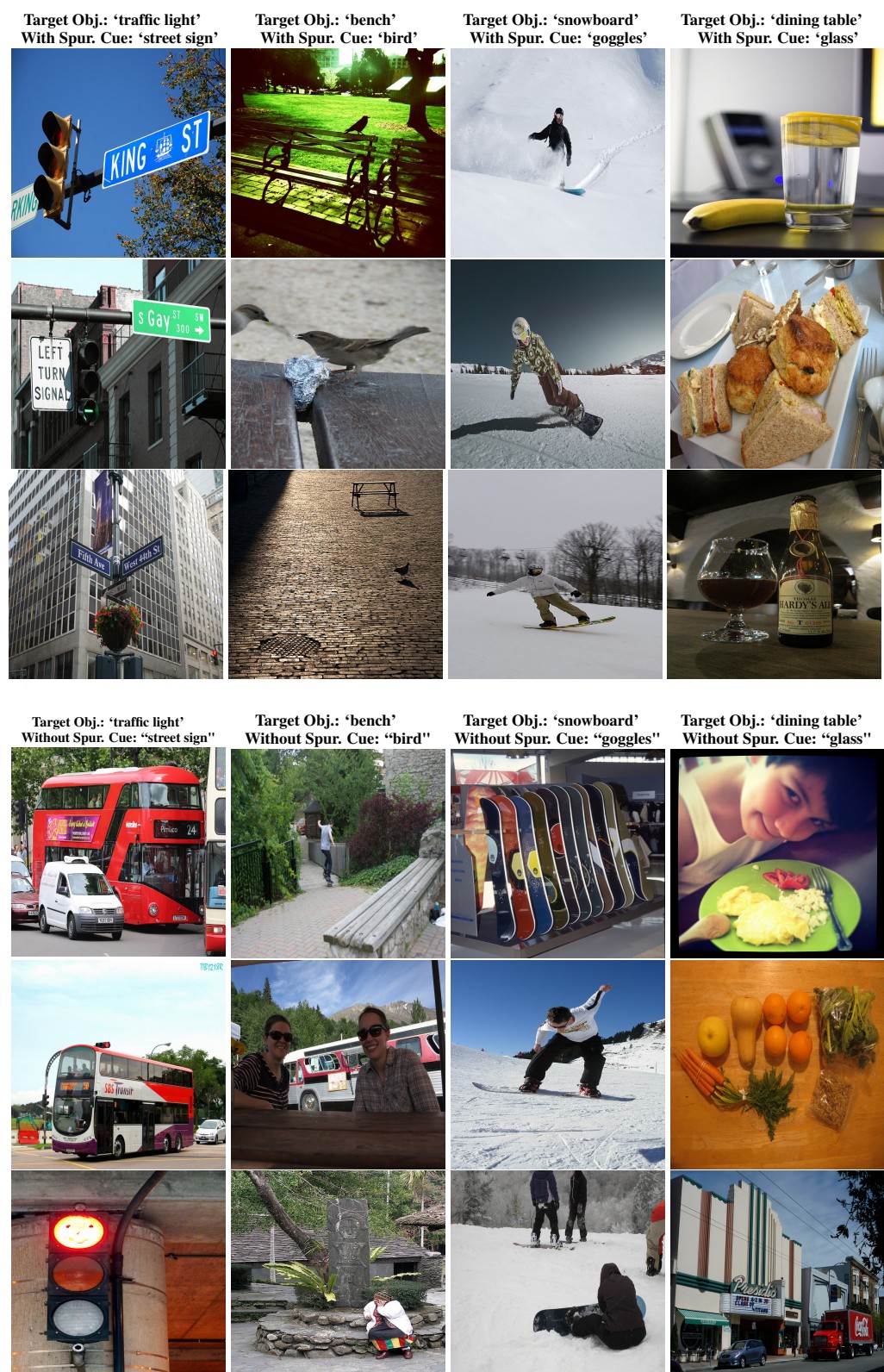

Figure 10: For the randomly-chosen COCO classes 'traffic light' (10), 'bench' (15), 'snowboard' (36), and 'dining table' (67), we randomly choose one of the spurious features we evaluated, and then randomly chose images from the top-100 and bottom-100 in the object detection ranking for that class/feature pair.

Table 6: Human agreement with object detector results on top/bottom 10 ranked images across 20 (COCO) and 25 (ImageNet) randomly selected classes.

| Dataset | Top Agreement | Bottom Agreement | Average |
|---------|---------------|------------------|---------|
| COCO | 0.89 | 0.96 | 0.925 |
| ImageNet | 0.84 | 0.948 | 0.894 |

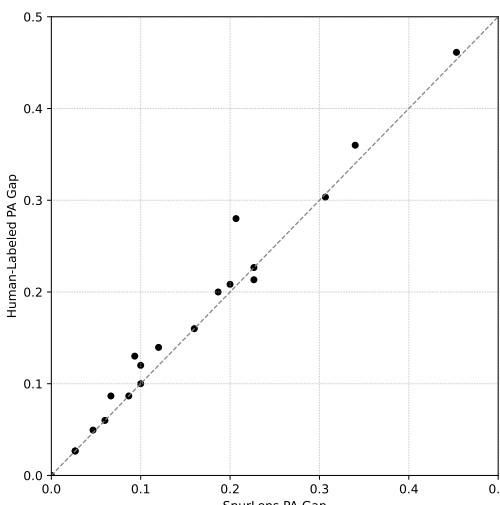

Figure 11: Comparison of SpurLens PA Gap and human-labeled PA Gap (both computed with $K = 50$) across 20 COCO classes. The strong correlation (Pearson $r = 0.988$) indicates that SpurLens produces rankings consistent with human judgments.

## G  CELEBA EVALUATION

While *SpurLens* focuses on spurious biases related to object recognition and hallucination, prior work has extensively studied other forms of bias, such as correlations between gender and blond hair in the CelebA (Liu et al., 2015) dataset, particularly in unimodal classification settings (Noohdani et al., 2024; Sagawa et al., 2019a; Kirichenko et al., 2022). Following this line of work, we examine whether such spurious correlations persist in MLLMs.

We adopt a similar evaluation setup: using the prompt "Does the person in the image have blond hair? Answer with 'Yes' or 'No'," we measure accuracy across groups with and without the spurious feature (gender). Results are shown in Table 7. While Spurious Gap still exists, we argue that it is not particularly meaningful in this context. In prior work, Unimodal classifiers trained with ERM performed poorly on the worst group (e.g., blond males), often below 50%. In contrast, MLLMs like Qwen2-VL achieve over 96% accuracy on this group in a zero-shot setting. Moreover, the CelebA annotations are noisy, and in many cases where Qwen's predictions differ from the labels, the true hair color is ambiguous even to humans. Figure 12 shows qualitative examples.

## H  WATERBIRDS EVALUATION

Similar to CelebA, the Waterbirds dataset has often been used to study the effect of spurious correlations in vision classifiers. Unlike CelebA, however, it is a synthetic dataset with a simplified structure. This may pose issues for MLLMs, which may err due to the synthetic, out-of-distribution nature of the inputs rather than the spurious correlations. As our goal is to draw conclusion about the performance of modern MLLMs in practical, real-world settings, the results of such an analysis may not be fully applicable.

Table 7: Qwen2-VL accuracy on CelebA across gender and hair color. The spurious feature in this setting is gender. All numbers are percentages.

| Hair Color | Male | Female | Gap |
|---|---|---|---|
| Blond | 96.63 | 98.28 | 1.65 |
| Non-blond | 81.70 | 74.84 | 6.86 |

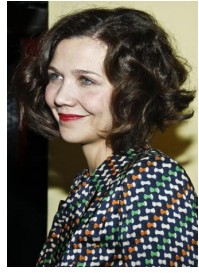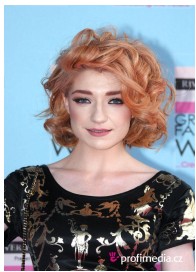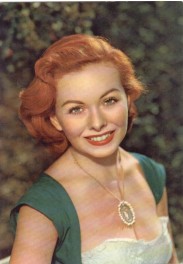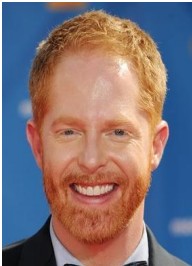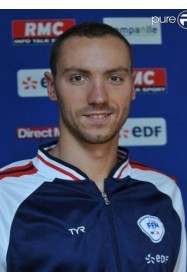

Figure 12: Qualitative examples from CelebA where Qwen-VL's answer ("Not blond") differs from the annotated ground truth ("Blond"). Hair color is often ambiguous, even for humans.

Additionally, Waterbirds was designed for evaluating spurious correlations in a training setup. In the training phase, the dataset creates strong spurious cues (e.g., landbirds predominantly being on land and waterbirds on water). This creates a high spurious correlation that the model learns to rely on. However, during testing, the dataset balances these cues, which causes a drop in classifier accuracy. In contrast, our work focuses on a zero-shot setup, where the model is not explicitly trained on such cues, and any spurious correlations that arise are more naturally occurring. Thus, we believe evaluating on such a synthetic, domain-specific setup would not align with the goal of evaluating MLLMs in more complex, real-world scenarios.

Nevertheless, we still explore Waterbirds' applicability within the context of MLLMs. We use the following prompt: "We call a bird 'waterbird' if it is a seabird (albatross, auklet, cormorant, frigatebird, fulmar, gull, jaeger, kittiwake, pelican, puffin, or tern) or waterfowl (gadwall, grebe, mallard, merganser, guillemot, or Pacific loon). Is there a waterbird in the image? Answer with 'Yes' or 'No'." This prompt is similar to our standard prompts, prepended by a definition of the target class "waterbird" (following the definition provided by Sagawa et al. (2019a)). The results are presented in Table 8. We observe that all MLLMs examined still exhibited significant spurious gaps, which we interpret as a clear indicator of residual spurious correlations in the models.

## I  HYPERPARAMETER SENSITIVITY ANALYSIS

We study how the Spurious Gap varies with the hyperparameter $K$, which controls the number of top- and bottom-ranked images used to compute the gap. As shown in Figure 13, all models exhibit a similar trend: the Spurious Gap decreases for larger values of $K$ and stabilizes as $K$ increases. This behavior is expected: smaller $K$ focuses on more extreme samples and amplifies the gap, while larger $K$ smooths the estimate by averaging over a broader set of images.

Importantly, the gap values begin to stabilize around $K \approx 60$, indicating that *SpurLens* produces consistent rankings and that our reported gaps are not highly sensitive to this parameter. These trends

Table 8: Accuracy of various open-source models on Waterbirds. All numbers are percentages.

| Model | Waterbird (Water) | Waterbird (Land) | Landbird (Water) | Landbird (Land) |
|---|---|---|---|---|
| Qwen | 94.58 | 81.35 | 64.02 | 96.03 |
| LLaVA | 59.33 | 13.24 | 77.38 | 99.76 |
| Llama | 94.34 | 79.90 | 38.99 | 70.88 |

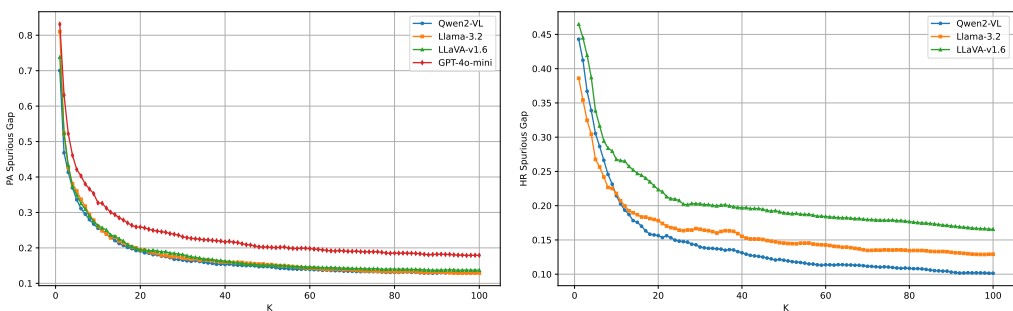

Figure 13: PA (**Left**) and HR (**Right**) Spurious Gap as a function of $K$, the number of top/bottom images used in evaluation. Gaps stabilize for $K \sim 60$, suggesting robustness to this parameter.

support the robustness of our method and suggest that our main findings are not artifacts of a specific $K$ choice.

## J PROMPTING MITIGATION DISCUSSION

This is an extended discussion of the attempted prompt-based mitigation from Section 6.2.

Table 4 includes $HR_b$: the hallucination rates on blank (fully black) images. While $HR_b$ is not always zero in the Baseline setting, it drops to zero with Ensemble or Dual Prompting, suggesting that these strategies help suppress some hallucinations.

The results in Table 4 reveal a trade-off between accuracy and hallucination rates across the reasoning-oriented strategies. Dual Prompting reduces HR, but at the cost of lower PA, while Guiding Prompting improves PA but increases hallucinations, suggesting that encouraging the model to focus on key details might reinforce reliance on spurious cues. In contrast, Prompt Ensembling provides a minor but consistent improvement across all cases, reducing HR while slightly increasing PA, making it the most balanced strategy. None of these increases or decreases are very significant, and none of these strategies resolve the Spurious Gap.

The final two strategies, Spurious List and Spurious Top, aim to leverage the textual information retrieved by the *SpurLens* pipeline. Since SpurLens leverages LLMs to identify spurious biases in natural language for each class, a natural idea is to provide a list of such potential features to the examined LLMs, which may cause them to recognize and/or avoid the bias; this is the basis of the Spurious List strategy. After running the *SpurLens* pipeline, we know which of the proposed spurious features is strongest for each class. Providing only this feature (rather than the list of all features, not all of which may be spurious) is not proper, as it contains information from the entire class dataset upon which we are also evaluating. Nevertheless, even with this additional information, we find that spurious bias still clearly persists. In fact, the results for the Spurious List and Spurious Top strategies are often weaker than the reasoning and ensembling-based strategies

The Spurious Gap remains present across all prompting strategies, suggesting that while prompt engineering can influence model behavior, it does not eliminate the underlying reliance on spurious cues. This indicates that the Spurious Gaps detected by *SpurLens* are a more fundamental issue that cannot be resolved solely through prompting techniques.

## K TOKEN DROPPING

### K.1 IMPLEMENTATION

We implemented token-dropping for Qwen2-VL, Llama-3.2 11B Vision Instruction, and LLaVa 1.6. We will focus on Qwen2-VL, as the discussion for the other models is roughly similar.

Suppose that we have an image of a target object, and have a bitmask of the object for that image. We aim to drop visual tokens from Qwen2-VL's processing of the image, without affecting its vision

of other areas. After some basic resizing, Qwen2-VL breaks the image into patches of size $14 \times 14$ pixels. However, adjacent $2 \times 2$ tokens are later merged into a single token by a simple MLP layer (Wang et al., 2024). Therefore, when dropping tokens, we must effectively work with a "token size" of $28 \times 28$ pixels.

For a given mask/image pair, after reshaping the mask in the same way that Qwen does, we condense the mask into a boolean for each $14 \times 14$ pixel patch, where a patch is `false` if any part of the mask lies within the $28 \times 28$ pixel area of the token that the patch is a part of, and `true` otherwise. (This is essentially flood-filling the the inverted mask into $28 \times 28$ pixel regions, then extracting all $14 \times 14$ chunks as booleans.) This condensed mask is then passed through the model into the vision encoder, where it is used to index the patch embeddings such that only the patches outside the expanded mask are kept. The mask indexing is also applied to the rotary positional embeddings, before both the positional and patch embeddings are passed to the patch-to-token merger. The mask information is also be passed to the input processor in order to add the correct number of vision placeholder tokens for the image.

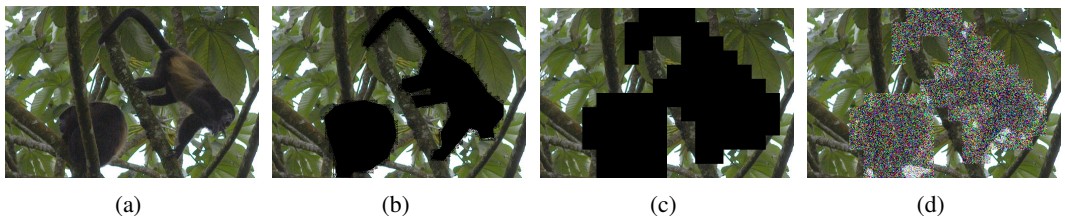

|      |      |      |      |
|:----:|:----:|:----:|:----:|
| (a)  | (b)  | (c)  | (d)  |

Figure 14: An example for token-dropping. (a) shows the original figure of a howler monkey, and (b) shows the mask of this image from HardImageNet. (c) Illustrates the regions that are dropped by token-dropping. In (d), the areas of the image that will be dropped are corrupted with uniform random noise.

Figure 14 illustrates the procedure using an image/mask pair from the "howler monkey" class of HardImageNet. Figure 14 (a) and (b) show the image and mask, and (c) shows the regions of the image that would be dropped, which is a superset of the mask. (Note that we visualize this as a mask, but the model would not see black masked-out regions – the tokens for those regions would not be present at all.)

## K.2 VALIDATION

We devise a small demonstration to show the validity of our method: we pass images to Qwen2-VL with the prompt "Is something wrong with this image?". The responses with various images from Figure 14 are as follows:

- With original image (a): "The image appears to be a close-up of tree branches and leaves. There doesn't seem to be anything inherently wrong with the image; it's a natural scene."

- With the masked image (b): "Yes, there appears to be an issue with the image. The silhouette of the animal seems to be floating in the air without any visible connection to the tree branches or leaves. This suggests that the image might be digitally altered or manipulated, as it is not a natural occurrence for an animal to be floating in the air without support."

- With the token-dropped image (visualized as a mask in (c)): "The image appears to be a close-up of tree branches and leaves. There doesn't seem to be anything inherently wrong with the image; it's a natural scene."

- We add uniform random noise to the area of the image that will be dropped to obtain (d). We then pass in the corrupted image with the same tokens dropped as in (c). The model answers: "The image appears to be a close-up of tree branches and leaves. There doesn't seem to be anything inherently wrong with the image; it's a natural scene."

The model does not notice any irregularities in the image in the token-dropping scenario, and further does not notice any noise corruption under the area that is dropped, confirming that the model does not see the dropped regions at all.

Table 9: Comparison of HR Gaps (as %) on artificial negative samples from HardImageNet. The Gaps are computed with $K = 50$, and the class-wise average is reported.

| Model | Token-Dropping | | | Masking | | |
|---|---|---|---|---|---|---|
| | $HR_s$ | $HR_c$ | HR Gap | $HR_s$ | $HR_c$ | HR Gap |
| Qwen2-VL | 50.2 | 37.2 | 13.0 | 53.1 | 35.7 | 17.4 |
| Llama-3.2 | 31.5 | 20.4 | 11.1 | 44.0 | 27.3 | 16.7 |
| LLaVA-1.6 | 45.7 | 34.6 | 11.1 | 40.5 | 31.3 | 9.2 |

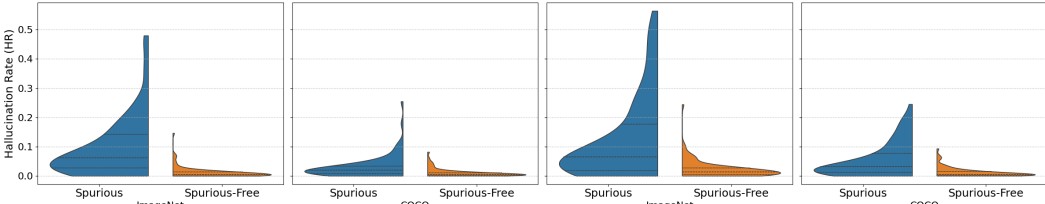

Figure 15: The distribution of HR across Spurious ImageNet and COCO classes for (**Left**) Qwen2-VL and (**Right**) LLaVA-v1.6. When spurious cues are present, hallucination amplifies.

### K.3 COMPARISON WITH MASKING

In Section 6.1, we computed HR Gaps on HardImageNet classes, using token-dropping to create artificial samples without the target object. Here, we compare those results to the results if we were to use masking (filling in the masked area with 0 RGB pixel values) to create the artificial negative samples.

The results are in Table 9. We find relatively similar class-wise averaged performance between the two constructions. Both methods give very large HR estimates, likely because both are out-of-distribution (OOD) images. Nevertheless, the HR Gap in both scenarios provides additional evidence of spurious correlation effects.

## L SPURIOUS BIAS IN MLLMS: EVIDENCE WITHOUT *SpurLens*

In this section, we provide evidence that spurious bias in MLLMs can be observed even without using *SpurLens*. Instead of systematically identifying high-spurious images, we leverage existing benchmarks, Spurious ImageNet (Neuhaus et al., 2023) and COCO (Lin et al., 2014), to demonstrate this issue.

Spurious ImageNet is a subset of 100 classes from ImageNet (Russakovsky et al., 2015), containing 75 images per class that include spurious cues but lack the target object. For baseline images, we randomly selected 75 images from ImageNet for each class.

COCO is a large-scale object detection, segmentation, and captioning dataset with rich annotations. It includes 80 categories organized into 10 supercategories (e.g., kitchen, furniture, vehicle). We argue that using only supercategory annotations is sufficient to support our claims. For each category, we randomly selected 500 images from the same supercategory but belonging to different categories as our spurious images. Additionally, we selected 500 images from different supercategories as spurious cue-free images.

We illustrate some hallucinations produced by GPT-4o-mini in this setup in Figure 16. The quantitative results for this experiment are shown in Table 10. For COCO, even with our simple setup for identifying spurious images, the HR is at least three times higher than that of the baseline images. For Spurious ImageNet, which systematically identifies spurious cues for each class, the ratios are even larger. Additionally, we visualized the HR distribution across different classes in Figure 15. Our findings align with Moayeri et al. (2023), highlighting that the bias from reliance on spurious cues is critically class-dependent.

Table 10: Hallucination Rates (HR) for Spurious ImageNet and COCO datasets. When spurious cues are present, the hallucination rate increases by at least threefold. * indicates that for GPT, due to API costs, we selected 100 images from COCO per category, compared to 500 images for other models.

| Model | Spurious ImageNet | | COCO | |
|---|---|---|---|---|
| | HR$_s$ | HR$_c$ | HR$_s$ | HR$_c$ |
| Qwen2-VL | **9.5** | 1.1 | **3.1** | 1.0 |
| Llama-3.2 | **11.0** | 2.1 | **9.6** | 3.1 |
| LLaVA-v1.6 | **12.5** | 2.2 | **5.5** | 1.2 |
| GPT-4o-mini | **3.2** | 0.3 | **2.6**$^*$ | 0.5$^*$ |

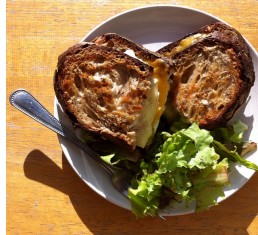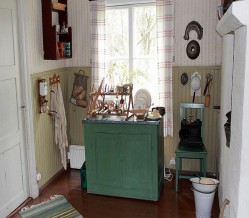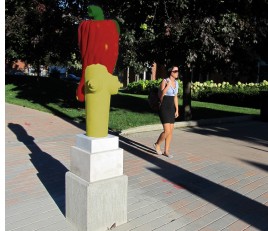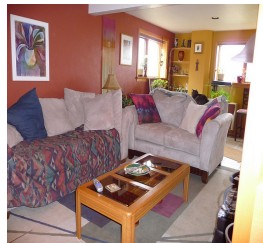

**Q:** Is there a spoon in the image? Explain.

Yes, there is a spoon in the image. It is positioned to the left of the plate, which contains a sandwich and some lettuce. The spoon has a decorative handle...

**Q:** Is there an oven in the image? Explain.

Yes, there is an oven in the image. It appears to be a small black stove located on the right side of the room, next to a green cabinet...

**Q:** Is there a parking meter in the image? Explain.

Yes, there appears to be a parking meter in the image, but it has been creatively designed to look like a piece of art. It resembles a cartoonish figure...

**Q:** Is there a cell phone in the image? Explain.

Output: Yes, there is a cell phone visible in the image. It appears to be placed on the coffee table, towards the left side. The phone is small and has a simple design...

Figure 16: Some failures of GPT-4o (accessed in January 2025) identified by our method. The model hallucinates when spurious cues are present.

## M  VISION ENCODER EXPERIMENTS

To further analyze the problem of spurious correlations, we perform a similar analysis to the main paper in a highly simplified setting, and show that several intuitions and results discussed hold true. We focus on the Qwen2-VL vision encoder, train various binary linear classifiers on HardImageNet data, and demonstrate that the effect of spurious correlations can still be found.

First, for each image in HardImageNet-train, we use the Qwen2-VL vision encoder to get the embedding for each $14 \times 14$ token in the image. We average the embeddings of all tokens to get a single representative embedding (vector of length 3584) for each image.

For each HardImageNet class, we train a logistic regression binary classifiers to determine whether a given image embedding is of that class. Let the target class be $c \in [1, 15]$. Consider the embeddings from HardImageNet-train of class $c$; after removing the $K = 100$ least spurious and $K = 100$ most spurious images (according to the original HardImageNet ranking), we randomly sample $x$ of the remaining images as the positive examples in our training dataset (where $x$ variable). For the negative examples in the training dataset, we randomly sample $f \times x$ embeddings from HardImageNet-train excluding class $c$ (where $f \geq 1$ is variable). This construction of the dataset allows us to control (1) the scale of the training dataset through $x$, and (2) how unbalanced the dataset is through $f$ (the positive:negative sample ratio in the training dataset is $1 : f$).

After training the logistic regression classifier on this dataset, we estimate the PA Gap $= \text{PA}_s - \text{PA}_c$ as before by computing the accuracy on the $K = 100$ most spurious and $K = 100$ least spurious images in HardImageNet-train of class $c$ (note that these were excluded from the training dataset). Additionally, we run the classifier on all of HardImageNet-val to get a better understanding of its overall accuracy.

Because the training datasets are randomly sampled, for each value of $x$ and $f$ and class $c$, we train 10 such classifiers with this procedure and average the results for $\text{PA}_s$, $\text{PA}_c$, the PA Gap, and

the HardImageNet-val accuracies. This experiment is performed for $x \in \{100, 200, \ldots 600\}$ and $f \in \{1, 2, 3, 5, 7\}$.

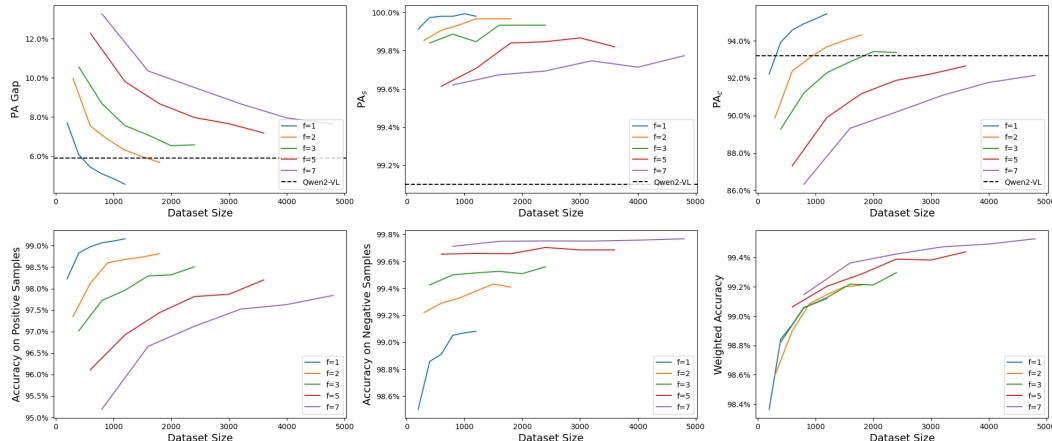

Figure 17: PA Gap results for various binary linear classifiers trained on mean-pooled image embeddings. The plots show various dataset sizes and levels of sample balance; each point is the classwise average (over 15 HardImageNet classes) of the metric shown; the metric for each class is the average of the results of 10 experiments on random training datasets, as previously described. The PA Gap (top left) is the difference between $\text{PA}_s$ (top middle) and $\text{PA}_c$ (top right). Accuracies are evaluated on HardImageNet-val on the target class (bottom left) and all classes except the target class (bottom middle). The weighted accuracy (bottom-right) is a 1:$f$ weighted average of these two. For the top three images, the PA values for Qwen2-VL from the main experiments are also depicted for comparison.

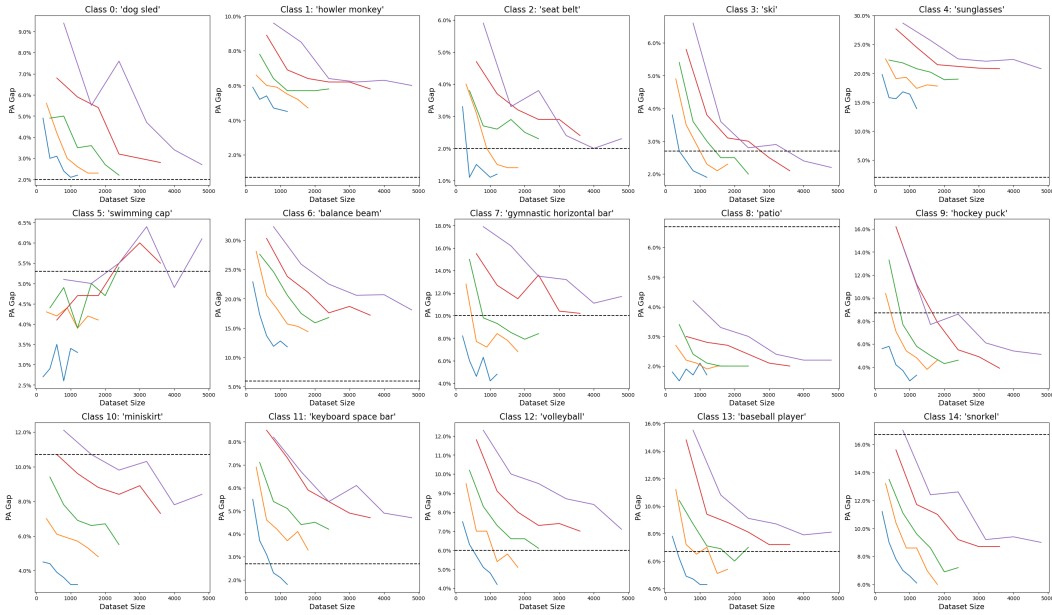

Figure 18: PA Gaps separately for each HardImageNet class for all experiments in Figure 17. The colors for different values of $f$ are the same as in that figure.

The (class-wise averaged) results for the PA metrics are found in Figure 17. Note that the training dataset size (horizontal axis) is computed as $x \times (1 + f)$. There are a few point to note. First, the classwise-averaged PA Gap measured for Qwen2-VL in our main experiments is of similar magnitude to the PA Gaps measured for these binary classifiers, which indicates that the vision encoder is largely

responsible for spurious bias effect and that the language model component did not significantly influence our results. Second, increasing dataset size tends to improve accuracy on both positive and negative samples, increase perception accuracy for both spurious and non-spurious images, and decreases the spurious gap; this is fairly intuitive. Third, for more unbalanced training datasets (larger values of $f$), the accuracy on positive samples tends to be lower, which is expected in a binary classification problem. However, the figure shows that in evaluating PA, there are far more errors for non-spurious images than there are for spurious images. When the model has seen a very wide set of images (which is true in the case of MLLMs), images without spurious features tend to be misclassified much more often than high-spurious images. This suggests that the source of spurious bias lies within the image embedding being insufficient to robustly distinguish between classes.

While the trends are somewhat clear for class-wise averaged results, there is significant variation between classes. Figure 18 presents the results of the experiments for each class separately. We see a significant variation in the scale of PA Gap as well as in the level of decrease in PA Gap as dataset size increases. Thus, while the effects discussed are generally true, the impact of spurious correlations in the image embeddings on perception is highly class-dependent.

Finally, we directly compare the gaps obtained through these vision encoder logistic regressions with the PA gaps from evaluation on Qwen2-VL on the HardImageNet rankings. Using $f = 7$, $K = 50$, and taking the maximum possible value of $x$ for each dataset (that being $2K$ less than the size of the HardImageNet-train for that class), we obtain a PA Gap on the vision encoder logistic regression classifier for each HardImageNet class. The class-wise average of these gaps is 0.082. Comparing these gaps class-wise against the PA Gaps obtained when evaluating Qwen2-VL on the HardImageNet rankings (with $K = 50$), we find a linear correlation coefficient of 0.25. This suggests that the vision encoder is partially responsible for the observed spurious bias in the full MLLM.

# N  SPURLENS VALIDATION

We apply SpurLens to the 15 HardImageNet classes. As discussed in Section 4, we use GPT-4 to generate 32 potential spurious features for each class, which are subsequently filtered through GPT filters and manual review of object detector results. For each model, after computing the PA Gap (with $K = 50$) for each potential spurious feature , we select the feature with the largest PA gap as the PA gap for the class; we then average these results class-wise.

We compare the PA Gaps of the spurious ranking selected by SpurLens to the PA Gap (with $K = 50$) obtained with the original HardImageNet spurious ranking (Moayeri et al., 2022). Additionally, we compare it against a random baseline constructed as follows. For each (class, model), we take 16 random rankings, compute the PA Gap, and take the maximum; this procedure is then performed 16 times, we take the average as the baseline PA Gap for that (class, model) pair. Comparisons against these baselines provide evidence that we are capturing a statistically significant signal.

Next, we apply SpurLens to 79 COCO classes. (We exclude the "person" class because (1) it is extremely generic and broad, making it outstandingly difficult to find strong spurious features for, and (2) it is several times larger than the second-largest class, making it very computationally expensive.) We use GPT-4 to generate 32 potential spurious features, and go through the same SpurLens process to compute PA Gaps with $K = 100$. Likewise, we compute random baselines for each class in the same manner.

(We emphasize that the Random Baseline is not a useful ranking method, but is provided as statistical evidence that the signal we measure cannot be explained as the result of random variations. The rankings produced by SpurLens are through object detection scores of spurious features and are thus interpretable, while the random rankings are not.)

Finally, we apply SpurLens to the 100 Imagenet classes from SpuriousImagenet (Neuhaus et al., 2023). PA Gaps are computed with $K = 50$, but otherwise the same procedure is used as for COCO.

The class-wise averaged results of these experiments are presented in Table 11. We note that, while strong spurious features are not present in every class, SpurLens does tend to find significant spurious cues for most classes. The spurious gaps found by SpurLens for each class can be found in Appendix S.1 for HardImageNet, Appendix S.2 for COCO, and Appendix S.3 for the ImageNet

Table 11: Comparison of class-wise averaged PA Gaps (as %) measured by SpurLens versus baselines. $K = 50$ for HardImageNet and Imagenet, and $K = 100$ for COCO.

| Model | HardImageNet | | | COCO | | ImageNet Subset | |
|---|---|---|---|---|---|---|---|
| | SpurLens Ranking | Original Ranking | Random Baseline | SpurLens Ranking | Random Baseline | SpurLens Ranking | Random Baseline |
| Qwen2-VL | 5.9 | 5.8 | 5.3 | 15.2 | 8.7 | 6.7 | 5.0 |
| Llama-3.2 | 15.0 | 12.1 | 8.5 | 14.8 | 8.6 | 17.7 | 9.4 |
| LLaVA-v1.6 | 13.2 | 7.5 | 7.4 | 16.0 | 8.4 | 15.7 | 8.4 |
| GPT-4o-mini | 12.6 | 6.3 | 9.1 | 20.4 | 10.6 | 11.9 | 8.3 |

subset. These results provide evidence that our methodology is both statistically significant and more effective than past work.

## O  ALTERNATIVE OBJECT DETECTORS

In the SpurLens pipeline, we use OWLv2 (Minderer et al., 2024) for open-set object detection to identify spurious features in a given image dataset. The reliability of OWLv2 is established in Appendix F. Here, we investigate using other open-set object detectors: GroundingDINO (Liu et al., 2024b) and YOLO World (Cheng et al., 2024). (Note that closed-vocabulary object detectors, such as DETR and many popular R-CNN variants, are not applicable to our methodology as we need to identify arbitrary spurious features/objects.)

Empirically, we observe that different object detectors have have different confidence behaviors and characteristics. GroundingDINO object detection results tend to have many false positives (incorrect labeled bounding boxes with fairly high confidence). Some qualitative examples of this are provided in Figure 19. This introduces significant noise, which leads to ranking errors and thus errors in Spurious Gap computation, especially for relatively small datasets such as HardImageNet classes ($\sim$ 1000 images per class). In contrast, YOLO World tends to be very conservative in its confidence score estimates, and often misses instances of spurious objects; OWLv2 lies in the middle of these extremes, with the best overall performance that we observe empirically.

Nevertheless, we do find some agreement between $f_i$-scores computed with these different object detection systems. Figure 20 provides examples of correlations between OWLv2-based $f_i$-scores and both GroundingDINO-based $f_i$-scores and YOLO World-based $f_i$-scores, for select (target object, spurious feature) pairs from HardImageNet. We observe that the correlations are generally strong but varied and class/cue dependent. Nevertheless, they provide us with confidence that our pipeline is sound, and that OWLv2 is a strong choice of object detector for SpurLens.

## P  DATASET DIVERSITY EVALUATION

In Section 4, SpurLens assumes that we have access to a "large dataset of images", and that the dataset is sufficiently diverse to support object detection for spurious feature identification. Here, we clarify these heuristics and outline a simple procedure to examine a dataset's adequacy for use with SpurLens.

A simple minimum requirement for a spurious feature to be "sufficiently represented" in a dataset is for there to be at least $K$ instances where it is present, and at least $K$ instances where it is not, where $K$ is the number of high-spurious and low-spurious images used for the MLLM evaluation. When analyzing GPT4-suggested spurious features, likely some will be sufficiently represented in a dataset, and some will not; a user would likely want at least $\widetilde{N}$ of the features to be sufficiently represented, for some reasonable value of $\widetilde{N}$. We would then run SpurLens on these features, and take the feature with the highest Spurious Gap, which we hope is positive, indicating that we identified a spurious correlation.

Suppose a user has a large dataset, and would like to test its diversity and suitableness before running the MLLM evaluation (the most expensive part of SpurLens). The user could perform

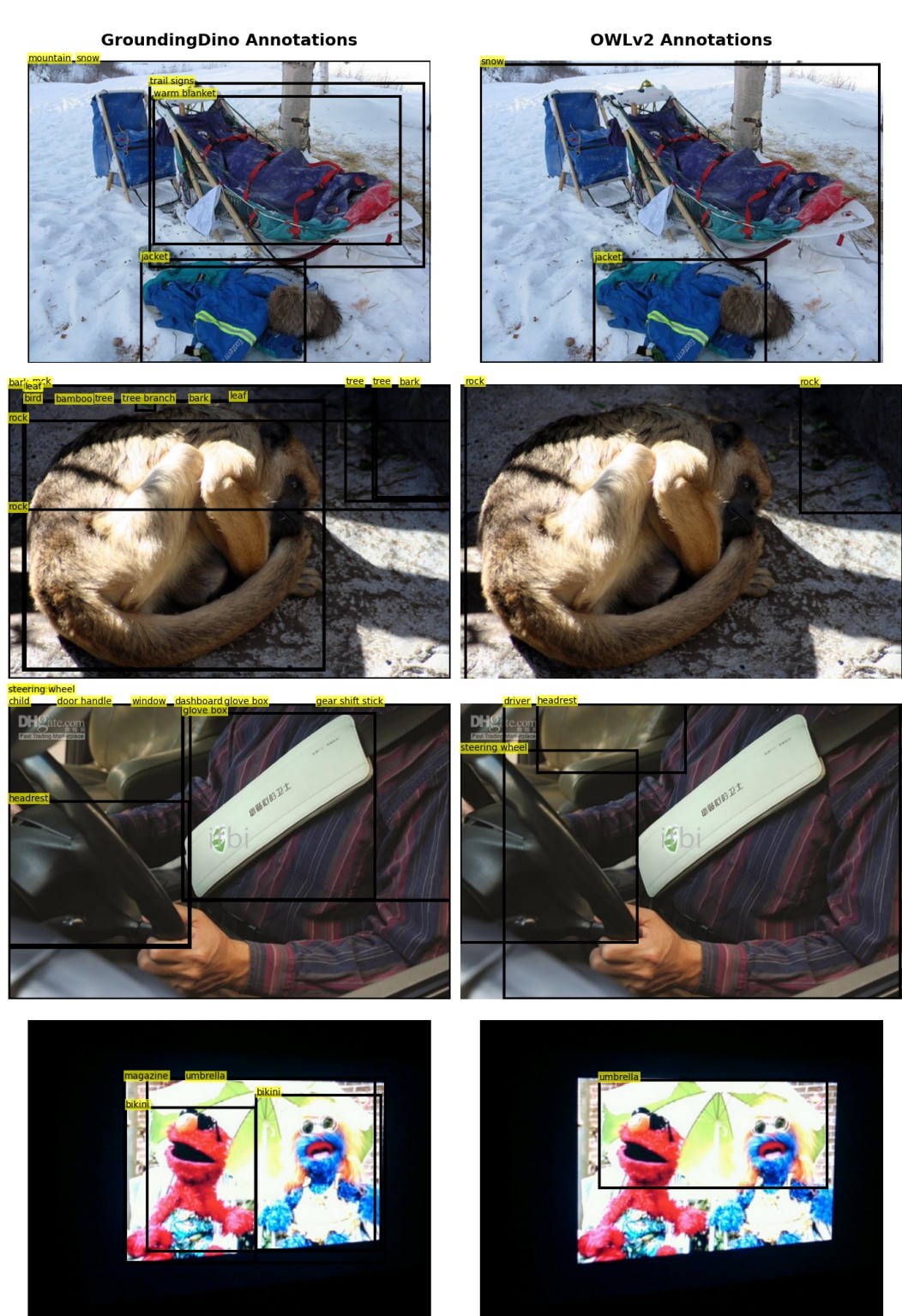

Figure 19: Qualitative examples comparing object detection with GroundingDino and OWLv2, taken from HardImageNet. OWLv2 is much more conservative with its bounding box proposals, while GroundingDino has many false positives before identifying the most pertinent spurious features.

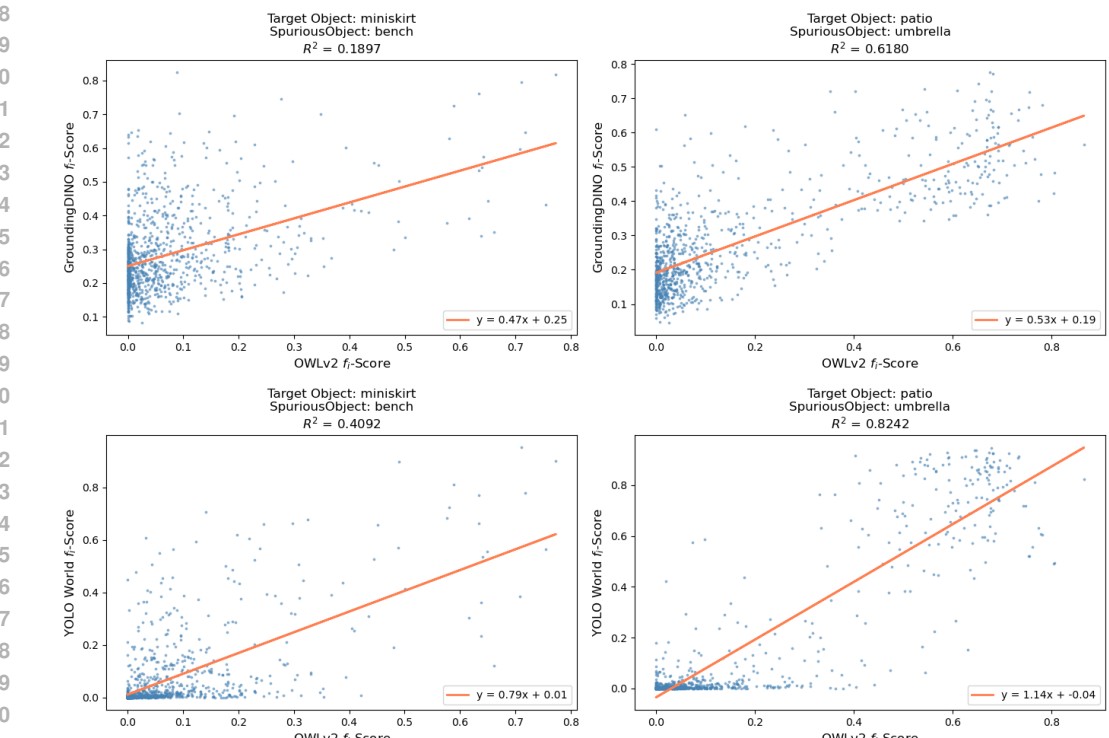

Figure 20: Plots comparing $f_i$-scores from OWLv2 to those from GroundingDINO and YOLO World object detection on two chosen (target object, spurious feature) pairs from HardImageNet. We observe that the correlation between the scores for the object detection models is object-dependent. We also observe the characteristics of these models: GroundingDINO tends to provide high detection scores to many images, while YOLO World is far far more conservative and gives near-0 scores to most images. OWLv2 lies between these extremes.

feature proposal with GPT-4 to obtain $N$ potential spurious cues (adjusting the prompt and providing additional domain-specific information, as needed, in order to obtain high-quality features for the downstream application). The user would choose $\widetilde{N} \le N$, the minimum number of sufficiently-represented spurious cues they would like to evaluate on (using their heuristics and domain-specific knowledge to estimate how many spurious cues would likely be needed before getting at least one positive result). The user would perform the object detection step with OWLv2 on all $N$ potential spurious cues.

For a given threshold $\tau \in [0, 1]$, a simple criteria is to believe that potential spurious cues with $f_i$-scores $> \tau$ have the image, and cues with $f_i$-score $< \tau$ do not have the image. Given $\tau$, a user can compute the maximum value of $K$ for each spurious cue such that it is sufficiently represented in the dataset. Finally, the user can compute the maximum $K$ such that at least $\widetilde{N}$ of the spurious cues are sufficiently represented; call denote value $K_{\tau,\widetilde{N}}$. The user perform this calculation for various values of $\tau$, and choose $\tau^\star$ with the maximal $K_{\tau^\star,\widetilde{N}}$. At this stage, the user can decide if the dataset is sufficiently diverse: if $K_{\tau^\star,\widetilde{N}}$ is sufficiently large, then they may proceed with the MLLM evaluation; otherwise, they must obtain a new dataset. The chosen value of $K$ determines the accuracy and precision of SpurLens' PA Gap estimates; the user must determine what value of $K$ is sufficiently large, based on their domain understanding and downstream objective.

As a case study, we apply this method to HardImageNet. In Figure 21, for various values of $\widetilde{N}$, we plot the minimum value of $K_{\tau^\star,\widetilde{N}}$ over all 15 HardImageNet classes against a range of $\tau$ values. We observe that $K_{\tau^\star,\widetilde{N}}$ is fairly high (over 100 in all cases). Additionally, if place harse restrictions on object detection confidence (such as $\tau > 0.3$), we are still able to achieve large $K$ values ($K > 50$).

This gives us confidence that HardImageNet is sufficiently diverse, and can support spurious feature detection and evaluation according to the standard SpurLens pipeline.

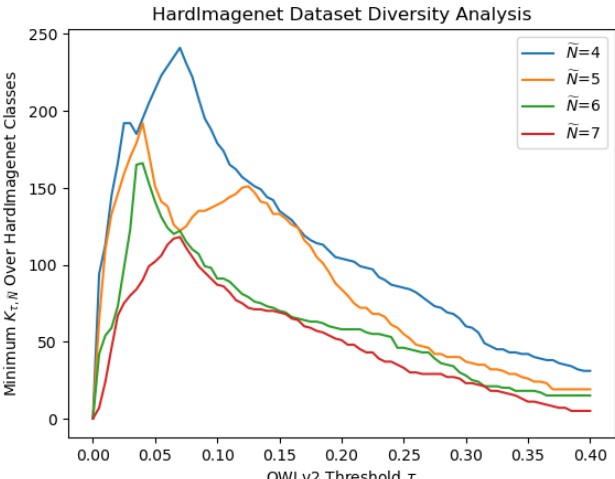

Figure 21: For various $\widetilde{N}$, we plot the minimum value of $K_{\tau,\widetilde{N}}$ over all 15 HardImageNet class for $\tau \in [0, 0.4]$. We observe that peak of each curve is above $K = 100$, supporting the fact that HardImageNet classes are diverse datasets and can support spurious object evaluation with (at least) $\widetilde{N} = 7$.

## Q    CLASS-WISE PLOTS

We provided plots for other models in Figure 22.

## R    QUALITATIVE EXAMPLES OF SPURIOUS GAP

We highlighted some failures identified by *SpurLens* in Figure 23.

## S    CLASS-WISE SPURLENS RESULTS

### S.1    HARDIMAGENET

As described in Section4, for each class, GPT-4 is used to generate 32 potential spurious features. We compute the PA Gap with $K = 50$ for these features, and choose feature with the maximum Gap. The results are in Table 12.

Additionally, we use token-dropping to turn each original image into an artificial negative example by dropping the tokens containing the target object. We compute the HR Gap with $K = 50$ for the same potential spurious features, and choose the feature with the maximum Gap. The results are in Table 13

### S.2    COCO

As described in Section 4, for 79 COCO classes, we use GPT-4 to generate 32 potential spurious features. We exclude COCO class 1 "person" from our analysis because (1) it is extremely generic and broad, making it outstandingly difficult to find good spurious features for, and (2) it is several times larger than the second-largest class, making it computationally expensive to analyze.

PA Gaps are computed with $K = 100$, and the feature with the largest Gap for each class is presented in Table 14.

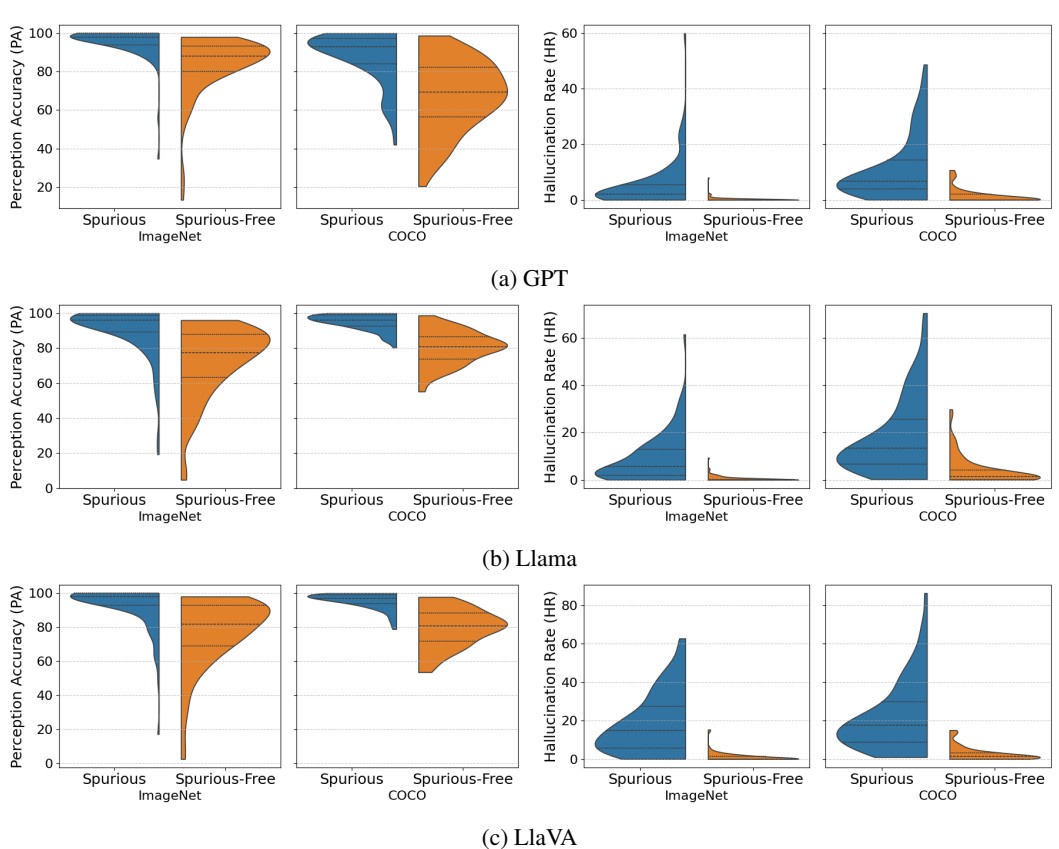

Figure 22: Class-wise distribution of PA and HR for experiments in Section 5.1 and Section 5.2 respectively. These highlight the class-dependent nature of spurious biases.

Table 12: Results of SpurLens applied to all 15 HardImageNet classes. For each model and for each class, we present the largest PA Gap (as %, with $K = 50$) and corresponding spurious feature found by SpurLens.

| HardImageNet | | Qwen2-VL | | Llama 3.2 | | LLaVa 1.6 | | GPT-4o-mini | |
|---|---|---|---|---|---|---|---|---|---|
| Index | Name | Spur. Feat. | PA Gap | Spur. Feat. | PA Gap | Spur. Feat. | PA Gap | Spur. Feat. | PA Gap |
| 0 | dog sled | goggles | 2.0 | snow | 22.0 | snow | 16.7 | goggles | 14.7 |
| 1 | howler monkey | tree branch | 2.0 | tree branch | 23.3 | tree branch | 6.7 | tree branch | 12.0 |
| 2 | seat belt | rearview mirror | 2.0 | rearview mirror | 6.7 | headrest | 8.0 | driver | 2.0 |
| 3 | ski | helmet | 2.0 | chairlift | 4.0 | tree | 4.0 | goggles | 6.0 |
| 4 | sunglasses | water bottle | 2.0 | water bottle | 5.3 | camera | 2.0 | camera | 6.0 |
| 5 | swimming cap | floatation device | 5.3 | pool noodles | 12.0 | goggles | 11.3 | floatation device | 8.0 |
| 6 | balance beam | leotard | 6.0 | leotard | 14.0 | tumbling mats | 15.3 | playground | 16.7 |
| 7 | gymnastic horizontal bar | scoreboard | 10.0 | foam pit | 17.3 | foam pit | 8.7 | rope | 10.7 |
| 8 | patio | table | 4.7 | umbrella | 9.3 | rug | 5.3 | table | 8.0 |
| 9 | hockey puck | goalie stick | 8.7 | goalie stick | 12.0 | goalie stick | 25.3 | ice surface | 9.3 |
| 10 | miniskirt | grass | 10.7 | grass | 28.7 | bench | 27.3 | crop top | 10.0 |
| 11 | keyboard space bar | monitor stand | 4.0 | printer | 29.3 | monitor stand | 14.0 | screen | 42.0 |
| 12 | volleyball | sky | 6.0 | scoreboard | 9.3 | ankle braces | 16.7 | scoreboard | 13.3 |
| 13 | baseball player | spectator | 6.7 | batting gloves | 18.0 | spectator | 20.0 | spectator | 8.7 |
| 14 | snorkel | sand | 16.7 | sand | 13.3 | goggles | 16.0 | sand | 22.0 |

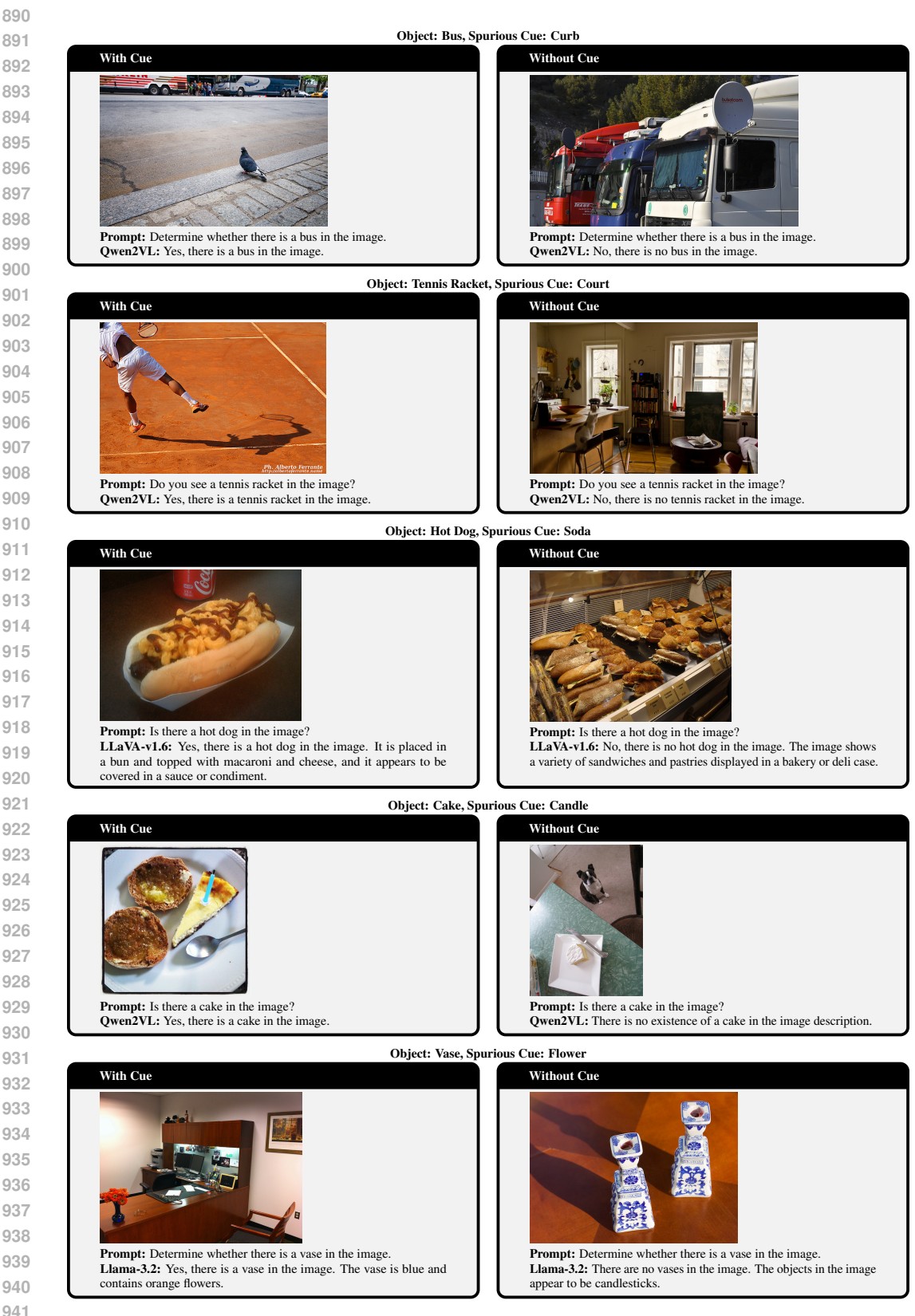

Figure 23: Visual examples from the COCO dataset. Object detection becomes more challenging for models when spurious cues are absent.

Table 13: Results of SpurLens applied to all 15 HardImageNet classes with artificial negative examples through token-dropping. For each model and for each class, we present the largest HR Gap (as %, with $K = 50$) and corresponding spurious feature found by SpurLens.

| HardImageNet | | Qwen2-VL | | Llama 3.2 | | LLaVa 1.6 | |
|---|---|---|---|---|---|---|---|
| Index | Name | Spur. Feat. | HR Gap | Spur. Feat. | HR Gap | Spur. Feat. | HR Gap |
| 0 | dog sled | collar | 38.0 | collar | 32.7 | collar | 44.7 |
| 1 | howler monkey | fallen logs | 8.7 | banana | 2.7 | fallen leaves | 9.3 |
| 2 | seat belt | cup holder | 12.7 | headrest | 27.3 | headrest | 38.0 |
| 3 | ski | goggles | 56.7 | goggles | 62.7 | slope | 66.0 |
| 4 | sunglasses | cap | 10.7 | cap | 17.3 | cap | 17.3 |
| 5 | swimming cap | floatation device | 23.3 | floatation device | 18.7 | starting block | 21.3 |
| 6 | balance beam | tumbling mats | 28.0 | playground | 38.0 | tumbling mats | 52.0 |
| 7 | gymnastic horizontal bar | scoreboard | 31.3 | scoreboard | 41.3 | scoreboard | 36.7 |
| 8 | patio | table | 28.0 | cushion | 30.0 | table | 18.7 |
| 9 | hockey puck | goalie stick | 18.0 | goalie stick | 18.0 | goal net | 16.7 |
| 10 | miniskirt | sidewalk | 41.3 | handbag | 8.0 | sidewalk | 36.7 |
| 11 | keyboard space bar | cable | 5.3 | mouse pad | 40.7 | cable | 11.3 |
| 12 | volleyball | banner | 37.3 | referee stand | 31.3 | gymnasium | 34.7 |
| 13 | baseball player | umpire mask | 51.3 | umpire mask | 38.0 | umpire mask | 50.0 |
| 14 | snorkel | wet suit | 15.3 | wetsuit | 20.7 | flipper | 36.9 |

To estimate HR Gaps, for each class, we take the images in the COCO superclass excluding the target class as the negative images dataset. For this experiment, we remove three additional classes ("sports ball", "dining table", and "keyboard") due to significant dataset error. We compute HR Gaps with the same spurious features and present the results in Table 15; we use $K = 50$ for GPT-4o-mini due to API costs, and $K = 100$ for all other models.

Finally, we also use token-dropping to turn images of each target object into artificial negative examples. HR Gaps and spurious features for these experiments are in Table 16.

Table 14: Results of SpurLens applied to almost all COCO classes. For each model and for each class, we present the largest PA Gap (as %, with $K = 100$) and corresponding spurious feature found by SpurLens.

| COCO | | Qwen2-VL | | Llama 3.2 | | LLaVa 1.6 | | GPT-4o-mini | |
|---|---|---|---|---|---|---|---|---|---|
| Index | Name | Spur. Feat. | PA Gap | Spur. Feat. | PA Gap | Spur. Feat. | PA Gap | Spur. Feat. | PA Gap |
| 2 | bicycle | basket | 16.5 | basket | 13.7 | basket | 10.0 | water bottle | 27.3 |
| 3 | car | road | 26.0 | road | 17.0 | pavement markings | 9.3 | road | 42.0 |
| 4 | motorcycle | mirror | 12.2 | glove | 13.3 | glove | 14.0 | glove | 22.3 |
| 5 | airplane | control tower | 3.0 | turbine | 7.9 | turbine | 5.0 | turbine | 8.4 |
| 6 | bus | curb | 18.0 | signage | 15.7 | curb | 18.7 | curb | 25.3 |
| 7 | train | fence | 1.7 | fence | 4.3 | fence | 4.0 | fence | 6.7 |
| 8 | truck | hitch | 43.3 | roof rack | 23.5 | hitch | 33.9 | hitch | 52.4 |
| 9 | boat | oar | 19.5 | buoy | 12.7 | dock | 18.7 | fishing gear | 27.0 |
| 10 | traffic light | street sign | 21.0 | street sign | 24.3 | street sign | 21.0 | street sign | 35.3 |
| 11 | fire hydrant | storm drain | 14.3 | storm drain | 12.7 | storm drain | 18.0 | storm drain | 21.3 |
| 13 | stop sign | tree | 10.0 | tree | 14.3 | landscape | 12.7 | tree | 18.0 |
| 14 | parking meter | ticket | 31.0 | ticket | 23.7 | ticket | 20.3 | ticket | 28.7 |
| 15 | bench | bush | 18.7 | flowerbed | 16.0 | book | 19.4 | tree | 25.3 |
| 16 | bird | leaf | 17.3 | branch | 16.7 | branch | 23.7 | branch | 18.7 |
| 17 | cat | collar | 4.0 | collar | 5.0 | scratching post | 4.1 | collar | 5.7 |
| 18 | dog | collar | 21.8 | collar | 30.2 | collar | 14.3 | collar | 28.1 |
| 19 | horse | saddle | 9.7 | saddle | 7.7 | saddle | 10.0 | saddle | 10.3 |
| 20 | sheep | path | 3.0 | pasture | 15.0 | pasture | 9.0 | pasture | 12.0 |
| 21 | cow | farmer | 9.1 | barn | 13.2 | barn | 13.3 | barn | 11.5 |
| 22 | elephant | mud | 5.0 | mud | 7.3 | mud | 9.3 | mud | 6.3 |
| 23 | bear | rock | 3.7 | grass | 7.7 | grass | 3.3 | grass | 5.0 |
| 24 | zebra | trail | 1.0 | dirt | 1.3 | grass | 3.0 | dirt | 3.0 |
| 25 | giraffe | fence | 0.7 | barbwire | 2.0 | savannah | 2.3 | savannah | 1.3 |
| 27 | backpack | pen | 16.7 | laptop | 16.0 | key | 19.7 | pen | 24.3 |
| 28 | umbrella | raincoat | 10.0 | bag | 10.7 | bag | 9.0 | raincoat | 22.7 |
| 31 | handbag | bag stand | 20.0 | lipstick | 17.2 | key | 21.0 | wallet | 39.8 |
| 32 | tie | lapel pin | 33.3 | suit | 35.3 | lapel pin | 17.7 | pocket square | 30.7 |
| 33 | suitcase | baggage cart | 36.5 | baggage cart | 15.7 | travel brochure | 17.3 | baggage cart | 28.3 |
| 34 | frisbee | park | 4.0 | tree | 7.0 | park | 8.3 | tree | 4.7 |
| 35 | skis | pole | 7.7 | glove | 14.3 | snowboard | 13.4 | glove | 14.3 |
| 36 | snowboard | pant | 19.2 | pant | 18.4 | terrain park | 23.0 | pant | 23.8 |
| 37 | sports ball | turf | 11.3 | jersey | 10.7 | referee | 10.3 | referee | 15.0 |
| 38 | kite | child | 2.0 | fence | 5.7 | field | 3.0 | fence | 7.7 |
| 39 | baseball bat | uniform | 5.5 | uniform | 8.8 | uniform | 27.1 | uniform | 8.2 |

*Continued on the next page*

| COCO | | Qwen2-VL | | Llama 3.2 | | LLaVa 1.6 | | GPT-4o-mini | |
|---|---|---|---|---|---|---|---|---|---|
| Index | Name | Spur. Feat. | PA Gap | Spur. Feat. | PA Gap | Spur. Feat. | PA Gap | Spur. Feat. | PA Gap |
| 40 | baseball glove | cap | 12.0 | cleat | 27.7 | cleat | 21.0 | grass | 22.3 |
| 41 | skateboard | wall | 2.0 | shoe | 5.0 | shoe | 4.0 | shoe | 4.3 |
| 42 | surfboard | bikini | 5.7 | wave | 6.7 | bikini | 5.7 | bikini | 8.3 |
| 43 | tennis racket | court | 12.0 | court | 18.0 | court | 19.3 | court | 14.3 |
| 44 | bottle | ice | 13.4 | cup | 8.7 | ice | 12.8 | table | 24.0 |
| 46 | wine glass | candle | 13.3 | cutlery | 11.7 | cutlery | 12.7 | cheese | 18.5 |
| 47 | cup | spoon | 24.3 | table | 18.7 | coaster | 24.7 | biscuit | 32.0 |
| 48 | fork | sauce | 20.7 | sauce | 20.3 | plate | 14.0 | plate | 21.0 |
| 49 | knife | chopping block | 21.0 | cutlery holder | 15.0 | chopping block | 14.7 | chopping block | 40.0 |
| 50 | spoon | bowl | 28.5 | food | 35.4 | sugar | 25.3 | sugar | 20.8 |
| 51 | bowl | granola | 27.3 | granola | 15.3 | granola | 19.7 | fruit | 34.7 |
| 52 | banana | coconut | 4.7 | fruit label | 9.6 | coconut | 9.0 | coconut | 12.0 |
| 53 | apple | table | 7.7 | knife | 5.3 | table | 10.7 | tree | 15.3 |
| 54 | sandwich | mayonnaise | 18.9 | pickle | 21.0 | pickle | 18.8 | mayonnaise | 23.8 |
| 55 | orange | fruit basket | 15.3 | fruit basket | 10.7 | fruit basket | 14.7 | fruit basket | 14.5 |
| 56 | broccoli | chicken | 14.0 | chicken | 13.0 | chicken | 11.3 | quinoa | 18.8 |
| 57 | carrot | vegetable | 30.5 | vegetable | 29.0 | vegetable | 33.6 | vegetable | 31.5 |
| 58 | hot dog | ketchup | 12.2 | ketchup | 17.7 | ketchup | 24.4 | ketchup | 23.7 |
| 59 | pizza | olive | 8.8 | onion | 10.1 | olive | 14.0 | mushroom | 12.0 |
| 60 | donut | box | 7.7 | box | 6.7 | box | 3.7 | box | 13.7 |
| 61 | cake | candle | 24.0 | candle | 23.3 | candle | 24.0 | candle | 31.0 |
| 62 | chair | cushion | 26.3 | lamp | 14.0 | table | 29.3 | table | 32.3 |
| 63 | couch | throw pillows | 14.7 | throw pillows | 14.5 | throw pillows | 36.4 | throw pillows | 20.1 |
| 64 | potted plant | tag | 14.8 | floor | 15.7 | vase | 12.6 | curtain | 21.7 |
| 65 | bed | sheet | 23.1 | sheet | 21.9 | sheet | 29.4 | sheet | 34.7 |
| 67 | dining table | napkin | 23.0 | plate | 34.7 | centerpiece | 21.7 | chair | 53.0 |
| 70 | toilet | sanitary bin | 10.0 | shower curtain | 9.3 | sanitary bin | 16.3 | sanitary bin | 12.0 |
| 72 | tv | dvd player | 39.7 | coffee table | 31.7 | sound system | 40.0 | dvd player | 42.7 |
| 73 | laptop | mouse | 10.3 | mouse | 11.3 | headphone | 9.7 | mouse | 15.3 |
| 74 | mouse | paper | 4.3 | computer | 11.0 | computer | 12.7 | computer | 13.3 |
| 75 | remote | blanket | 8.0 | carpet | 6.3 | couch | 2.3 | blanket | 1.3 |
| 76 | keyboard | wrist rest | 42.2 | wrist rest | 19.1 | wrist rest | 45.7 | wrist rest | 67.2 |
| 77 | cell phone | sunglass | 9.3 | watch | 9.7 | sunglass | 10.0 | sunglass | 7.7 |
| 78 | microwave | coffee maker | 8.7 | cabinet | 12.7 | oven | 23.3 | countertop | 9.7 |
| 79 | oven | cooling rack | 23.3 | wall | 17.7 | pan | 21.0 | microwave | 22.0 |
| 80 | toaster | cutting board | 9.7 | kitchen counter | 10.7 | microwave | 12.3 | kitchen counter | 9.3 |
| 81 | sink | sponge holder | 18.4 | towel | 11.3 | sponge holder | 13.4 | sponge holder | 23.6 |
| 82 | refrigerator | magnet | 18.7 | magnet | 20.0 | magnet | 37.7 | magnet | 19.3 |
| 84 | book | blanket | 14.7 | coaster | 10.7 | lamp | 16.7 | clock | 13.3 |
| 85 | clock | wall | 23.7 | sign | 17.3 | wall | 17.7 | wall | 25.0 |
| 86 | vase | leaf | 27.3 | flower | 30.0 | leaf | 19.7 | leaf | 33.0 |
| 87 | scissors | marker | 9.3 | felt | 20.6 | glue | 11.2 | felt | 10.3 |
| 88 | teddy bear | cuddly blanket | 6.0 | crib | 3.0 | cuddly blanket | 3.0 | cuddly blanket | 9.0 |
| 89 | hair drier | conditioner | 6.0 | conditioner | 12.0 | conditioner | 5.0 | basket | 10.7 |
| 90 | toothbrush | comb | 10.1 | floss | 9.3 | comb | 12.0 | comb | 6.0 |

Table 15: For almost all COCO classes, we apply SpurLens to other images in the same superclass. For each model and class, we present the largest HR Gap (as %), and corresponding spurious features. We use $K = 50$ for GPT-4o-mini and $K = 100$ for the other models.

| COCO | | Qwen2-VL | | Llama 3.2 | | LLaVa 1.6 | | GPT-4o-mini | |
|---|---|---|---|---|---|---|---|---|---|
| Index | Name | Spur. Feat. | HR Gap | Spur. Feat. | HR Gap | Spur. Feat. | HR Gap | Spur. Feat. | HR Gap |
| 2 | bicycle | basket | 24.0 | lamp post | 11.3 | basket | 24.3 | sidewalk | 8.0 |
| 3 | car | tree | 17.3 | traffic signs | 17.7 | tree | 17.3 | tree | 17.3 |
| 4 | motorcycle | mirror | 10.0 | billboard | 9.8 | billboard | 6.7 | barrier | 1.3 |
| 5 | airplane | baggage cart | 1.3 | sky | 2.7 | luggage | 1.3 | car | 0.7 |
| 6 | bus | traffic light | 1.7 | traffic cone | 12.7 | post box | 6.3 | tree | 3.3 |
| 7 | train | graffiti | 2.3 | bridge | 3.3 | bridge | 12.3 | bridge | 4.0 |
| 8 | truck | hitch | 8.3 | roof rack | 22.5 | parking lot | 24.0 | sign | 9.3 |
| 9 | boat | sail | 15.0 | dock | 28.8 | dock | 25.0 | dock | 18.7 |
| 10 | traffic light | bus stop | 7.7 | crosswalk | 10.7 | bus stop | 7.7 | crosswalk | 8.7 |
| 11 | fire hydrant | car | 9.0 | street | 10.2 | storm drain | 18.7 | storm drain | 9.3 |
| 13 | stop sign | crosswalk | 9.7 | side street | 7.0 | utility pole | 15.3 | side street | 4.7 |
| 14 | parking meter | car | 2.3 | bus stop | 8.7 | bus stop | 18.0 | curb | 5.3 |
| 15 | bench | trashcan | 5.0 | umbrella | 12.7 | coffee cup | 6.0 | coffee cup | 8.0 |
| 16 | bird | pond | 3.0 | sky | 7.8 | feeder | 18.3 | bush | 4.0 |
| 17 | cat | tv | 5.3 | furniture | 5.2 | shelf | 8.3 | door | 5.3 |
| 18 | dog | collar | 3.7 | bed | 7.5 | toy | 8.0 | bed | 8.7 |
| 19 | horse | saddle | 6.0 | barn | 4.0 | saddle | 19.3 | trailer | 6.0 |
| 20 | sheep | hill | 6.0 | hill | 4.7 | shepherd | 6.3 | hill | 5.3 |
| 21 | cow | farmer | 6.7 | road | 7.0 | road | 17.3 | hill | 7.3 |
| 22 | elephant | sky | 0.7 | bamboo | 0.7 | path | 1.0 | mud | 0.0 |
| 23 | bear | rock | 0.0 | cave | 0.0 | river | 1.3 | log | 0.0 |
| 24 | zebra | bush | 1.0 | antelope | 2.0 | antelope | 1.0 | bird | 0.0 |

*Continued on the next page*

| COCO | | Qwen2-VL | | Llama 3.2 | | LLaVa 1.6 | | GPT-4o-mini | |
|---|---|---|---|---|---|---|---|---|---|
| Index | Name | Spur. Feat. | HR Gap | Spur. Feat. | HR Gap | Spur. Feat. | HR Gap | Spur. Feat. | HR Gap |
| 25 | giraffe | safari vehicle | 1.0 | cloud | 0.5 | horizon | 2.0 | safari vehicle | 1.3 |
| 27 | backpack | charger | 23.0 | laptop | 20.3 | first aid kit | 26.7 | trail | 16.0 |
| 28 | umbrella | raincoat | 9.3 | raincoat | 6.2 | flower pot | 6.0 | dog | 2.0 |
| 31 | handbag | notebook | 43.3 | notebook | 25.2 | notebook | 35.0 | sofa | 10.7 |
| 32 | tie | suit | 14.7 | suit | 10.0 | suit | 40.3 | suit | 15.3 |
| 33 | suitcase | baggage cart | 7.3 | concrete floor | 13.8 | travel pillow | 22.7 | baggage cart | 6.7 |
| 34 | frisbee | dog | 5.3 | dog | 5.8 | dog | 18.7 | people | 2.0 |
| 35 | skis | snow shovel | 10.7 | snow shovel | 13.3 | snow pants | 65.0 | snow shovel | 12.0 |
| 36 | snowboard | goggle | 6.7 | snow | 4.2 | goggles | 17.7 | terrain park | 4.0 |
| 38 | kite | field | 3.3 | horizon | 2.7 | horizon | 7.0 | horizon | 4.7 |
| 39 | baseball bat | home plate | 9.0 | ballpark | 14.8 | home plate | 6.0 | ballpark | 6.0 |
| 40 | baseball glove | uniform | 15.3 | uniform | 16.0 | bat | 21.0 | uniform | 9.3 |
| 41 | skateboard | sticker | 7.7 | ramp | 4.0 | ramp | 13.3 | street | 4.0 |
| 42 | surfboard | paddle | 8.0 | wetsuit | 9.2 | wetsuit | 16.7 | fishing boat | 4.7 |
| 43 | tennis racket | court | 2.0 | court | 2.0 | court | 3.0 | court | 4.0 |
| 44 | bottle | counter | 6.0 | wall | 28.7 | box | 8.3 | cup | 22.7 |
| 46 | wine glass | candle | 6.3 | cork | 12.5 | coaster | 19.0 | cork | 5.3 |
| 47 | cup | tea | 35.3 | tea | 33.7 | tea | 25.0 | tea | 22.7 |
| 48 | fork | plate | 8.0 | food | 28.5 | food | 45.3 | cheese | 20.0 |
| 49 | knife | cutlery holder | 21.3 | utensil | 32.0 | utensil | 40.0 | utensil | 12.0 |
| 50 | spoon | whisk | 20.0 | chopstick | 30.2 | whisk | 36.0 | whisk | 21.3 |
| 51 | bowl | pudding | 17.3 | pudding | 27.7 | vegetable | 26.3 | ice cream | 26.0 |
| 52 | banana | coconut | 17.3 | paper towel | 5.7 | coconut | 6.3 | wall | 10.0 |
| 53 | apple | basket of fruit | 6.7 | picnic blanket | 6.7 | basket of fruit | 12.3 | table | 2.0 |
| 54 | sandwich | mustard | 31.3 | mustard | 67.7 | mustard | 66.7 | mustard | 40.7 |
| 55 | orange | wall | 11.3 | wall | 4.7 | wall | 20.7 | cutting board | 9.0 |
| 56 | broccoli | lemon | 7.3 | lemon | 8.0 | carrot | 11.7 | napkin | 4.0 |
| 57 | carrot | vegetable | 15.7 | soup | 11.2 | vegetable | 37.0 | bowl | 3.3 |
| 58 | hot dog | soda | 2.3 | cheese | 2.8 | relish | 2.0 | cheese | 6.7 |
| 59 | pizza | basil | 2.7 | olive | 4.3 | knife | 2.0 | basil | 2.0 |
| 60 | donut | coffee | 6.0 | coffee | 2.2 | coffee | 5.7 | table | 2.0 |
| 61 | cake | chocolate | 9.0 | chocolate | 12.7 | candle | 12.0 | candle | 4.0 |
| 62 | chair | cushion | 35.7 | footrest | 44.7 | footrest | 52.3 | cushion | 46.7 |
| 63 | couch | blanket | 27.3 | television | 17.3 | coffee table | 14.3 | coffee table | 21.3 |
| 64 | potted plant | vase | 51.7 | vase | 24.3 | vase | 83.0 | vase | 36.7 |
| 65 | bed | sheet | 19.0 | sheet | 21.7 | sheet | 13.0 | nightstand | 14.7 |
| 70 | toilet | diaper pail | 3.3 | sink | 9.8 | sink | 2.3 | sink | 8.7 |
| 72 | tv | game console | 4.7 | couch | 27.7 | coffee table | 11.7 | rug | 4.0 |
| 73 | laptop | mouse | 13.3 | webcam | 13.3 | chair | 13.0 | mouse | 10.0 |
| 74 | mouse | computer | 9.3 | computer | 13.0 | paper | 10.0 | lab equipment | 0.7 |
| 75 | remote | floor | 9.3 | coaster | 28.7 | window | 22.7 | blanket | 4.0 |
| 77 | cell phone | charger | 7.7 | key | 10.0 | bag | 11.7 | book | 8.7 |
| 78 | microwave | cooking tray | 4.0 | oven | 22.7 | oven | 9.3 | plastic wrap | 6.7 |
| 79 | oven | pan | 19.7 | microwave | 44.5 | pan | 21.7 | microwave | 28.7 |
| 80 | toaster | knife | 23.7 | coffee maker | 19.7 | microwave | 50.3 | knife | 20.0 |
| 81 | sink | trash can | 8.7 | trash can | 11.0 | knife | 23.0 | trash can | 18.0 |
| 82 | refrigerator | magnet | 13.7 | magnet | 14.3 | magnet | 14.3 | magnet | 25.3 |
| 84 | book | lamp | 9.3 | note | 24.3 | pencil | 12.7 | pencil | 4.7 |
| 85 | clock | lamp | 10.0 | alarm button | 9.0 | alarm button | 7.3 | alarm button | 6.7 |
| 86 | vase | flower | 18.0 | flower | 38.8 | plant | 30.3 | flower | 17.3 |
| 87 | scissors | marker | 28.3 | cardboard | 6.7 | glue | 35.0 | glue | 2.0 |
| 88 | teddy bear | cuddly blanket | 13.0 | bed | 6.5 | cuddly blanket | 14.3 | book | 4.0 |
| 89 | hair drier | conditioner | 10.0 | conditioner | 5.8 | shampoo | 40.7 | makeup remover | 4.0 |
| 90 | toothbrush | toothpaste | 11.3 | comb | 13.8 | comb | 24.0 | shaving cream | 4.0 |

Table 16: Results of SpurLens applied to almost all COCO classes with token-dropping to artificially create negative samples. For each model and for each class, we present the largest HR Gap (as %, with $K = 100$) and corresponding spurious feature found by SpurLens.

| COCO | | Qwen2-VL | | Llama 3.2 | | LLaVa 1.6 | |
|---|---|---|---|---|---|---|---|
| Index | Name | Spur. Feat. | HR Gap | Spur. Feat. | HR Gap | Spur. Feat. | HR Gap |
| 2 | bicycle | helmet | 22.0 | helmet | 16.3 | helmet | 32.7 |
| 3 | car | billboard | 16.7 | mountain | 15.6 | billboard | 8.7 |
| 4 | motorcycle | helmet | 66.7 | helmet | 21.8 | helmet | 61.3 |
| 5 | airplane | turbine | 19.6 | sky | 13.9 | turbine | 22.3 |
| 6 | bus | parking meter | 5.9 | crosswalk | 11.8 | signage | 7.0 |
| 7 | train | graffiti | 10.1 | graffiti | 11.1 | tunnel | 8.0 |
| 8 | truck | hitch | 12.6 | roof rack | 16.2 | parking lot | 36.5 |
| 9 | boat | fishing gear | 26.7 | buoy | 21.2 | buoy | 34.4 |
| 10 | traffic light | pedestrian | 18.3 | crosswalk | 21.1 | curb | 23.3 |

| COCO | | Qwen2-VL | | Llama 3.2 | | LLaVa 1.6 | |
|---|---|---|---|---|---|---|---|
| Index | Name | Spur. Feat. | HR Gap | Spur. Feat. | HR Gap | Spur. Feat. | HR Gap |
| 11 | fire hydrant | sidewalk | 12.0 | street | 12.2 | storm drain | 11.3 |
| 13 | stop sign | side street | 18.6 | side street | 7.8 | traffic light | 10.3 |
| 14 | parking meter | pavement | 14.5 | curb | 10.9 | bus stop | 20.7 |
| 15 | bench | sunglass | 25.0 | laptop | 23.6 | book | 28.1 |
| 16 | bird | branch | 12.7 | mountain | 17.9 | water | 12.0 |
| 17 | cat | scratching post | 14.3 | scratching post | 5.7 | scratching post | 14.0 |
| 18 | dog | leash | 7.0 | leash | 4.5 | leash | 12.7 |
| 19 | horse | riding boots | 38.3 | jumping poles | 11.1 | stirrup | 14.7 |
| 20 | sheep | barn | 11.3 | barn | 9.4 | hill | 10.7 |
| 21 | cow | animal pen | 10.3 | pasture gate | 12.1 | bale | 13.9 |
| 22 | elephant | sand | 5.3 | bush | 0.0 | waterhole | 7.0 |
| 23 | bear | tree | 5.0 | river | 4.6 | tree | 2.3 |
| 24 | zebra | waterhole | 12.0 | waterhole | 16.8 | waterhole | 5.3 |
| 25 | giraffe | waterhole | 7.0 | savannah | 6.0 | waterhole | 5.3 |
| 27 | backpack | first aid kit | 26.6 | water bottle | 12.6 | first aid kit | 24.0 |
| 28 | umbrella | lamp post | 8.7 | bag | 18.6 | people | 27.1 |
| 31 | handbag | bag stand | 32.9 | key | 16.2 | bag stand | 27.5 |
| 32 | tie | suit | 22.7 | pocket square | 14.0 | suit | 56.3 |
| 33 | suitcase | baggage cart | 38.3 | baggage cart | 32.0 | waiting area | 43.0 |
| 34 | frisbee | people | 28.2 | bicycle | 14.0 | tree | 28.0 |
| 35 | skis | glove | 43.0 | glove | 55.2 | snow pants | 42.6 |
| 36 | snowboard | glove | 51.0 | glove | 52.0 | terrain park | 31.7 |
| 37 | sports ball | jersey | 11.0 | jersey | 38.4 | referee | 21.3 |
| 38 | kite | backpack | 18.0 | mountain | 16.7 | camera | 7.3 |
| 39 | baseball bat | helmet | 55.3 | umpire's gear | 59.6 | helmet | 22.3 |
| 40 | baseball glove | bat | 17.7 | uniform | 29.5 | bat | 41.3 |
| 41 | skateboard | ramp | 50.0 | ramp | 29.1 | ramp | 27.7 |
| 42 | surfboard | wetsuit | 36.0 | wave | 32.1 | wetsuit | 30.0 |
| 43 | tennis racket | line | 16.7 | line | 21.1 | line | 20.0 |
| 44 | bottle | wall | 9.9 | cabinet | 30.6 | box | 13.3 |
| 46 | wine glass | candle | 14.0 | ice bucket | 16.5 | ice bucket | 18.0 |
| 47 | cup | window | 8.3 | tea | 23.8 | tea | 23.7 |
| 48 | fork | chair | 16.0 | chair | 18.0 | chair | 33.0 |
| 49 | knife | chef hat | 13.4 | cutlery holder | 30.0 | pan | 20.0 |
| 50 | spoon | whisk | 21.3 | food | 26.8 | chopstick | 26.3 |
| 51 | bowl | rice | 14.3 | yogurt | 22.8 | dishcloth | 26.7 |
| 52 | banana | coconut | 41.3 | coconut | 21.8 | coconut | 40.7 |
| 53 | apple | crate | 26.6 | crate | 14.6 | crate | 34.7 |
| 54 | sandwich | basket | 6.3 | chip | 19.6 | cheese | 5.9 |
| 55 | orange | wooden crate | 38.2 | wooden crate | 14.8 | wooden crate | 43.0 |
| 56 | broccoli | pan | 17.7 | pasta | 20.9 | quinoa | 18.4 |
| 57 | carrot | vegetable | 26.5 | fruit | 0.0 | vegetable | 55.5 |
| 58 | hot dog | grill | 16.7 | soda | 10.2 | grill | 8.3 |
| 59 | pizza | parchment paper | 22.2 | light fixture | 10.6 | light fixture | 16.7 |
| 60 | donut | display stand | 37.7 | display stand | 26.5 | display stand | 33.0 |
| 61 | cake | banner | 22.3 | balloon | 17.6 | banner | 18.3 |
| 62 | chair | footrest | 20.7 | footrest | 16.3 | footrest | 38.6 |
| 63 | couch | pillow | 13.8 | coffee cup | 18.6 | throw pillows | 18.1 |
| 64 | potted plant | tag | 16.9 | tag | 20.1 | pebble | 21.0 |
| 65 | bed | throw blanket | 18.4 | book | 17.0 | teddy bear | 16.9 |
| 67 | dining table | napkin | 30.8 | chair | 26.9 | silverware | 33.4 |
| 70 | toilet | sanitary bin | 13.7 | wall art | 18.6 | sanitary bin | 13.0 |
| 72 | tv | sound system | 16.6 | dvd player | 41.6 | dvd player | 42.7 |
| 73 | laptop | desk | 15.3 | desk | 21.3 | sticky notes | 13.7 |
| 74 | mouse | lab equipment | 5.3 | computer | 10.7 | lab equipment | 16.3 |
| 75 | remote | charger | 32.3 | wall | 28.0 | speaker | 25.3 |
| 76 | keyboard | wrist rest | 11.9 | chair | 21.7 | sticky notes | 11.3 |
| 77 | cell phone | sunglass | 17.7 | sunglass | 28.9 | sunglass | 21.0 |
| 78 | microwave | trash bin | 9.7 | fruit bowl | 24.9 | oven | 11.7 |
| 79 | oven | apron | 31.7 | baking sheet | 12.8 | pan | 20.9 |
| 80 | toaster | cutting board | 8.0 | wall cabinets | 9.7 | microwave | 28.0 |
| 81 | sink | sponge holder | 24.4 | towel | 30.4 | sponge | 26.7 |

| COCO | | Qwen2-VL | | Llama 3.2 | | LLaVa 1.6 | |
|---|---|---|---|---|---|---|---|
| Index | Name | Spur. Feat. | HR Gap | Spur. Feat. | HR Gap | Spur. Feat. | HR Gap |
| 82 | refrigerator | milk | 11.9 | milk | 17.7 | yogurt | 7.8 |
| 84 | book | pencil | 7.7 | clock | 17.1 | note | 9.2 |
| 85 | clock | lamp | 13.7 | table | 3.1 | chair | 2.3 |
| 86 | vase | decorative balls | 15.6 | flower | 51.6 | flower | 40.7 |
| 87 | scissors | marker | 16.7 | craft box | 8.7 | craft box | 47.1 |
| 88 | teddy bear | toy box | 20.3 | toy box | 16.3 | toy box | 32.0 |
| 89 | hair drier | toothbrush | 10.7 | brush | 13.7 | shampoo | 34.7 |
| 90 | toothbrush | comb | 14.3 | toothpaste | 22.6 | toothpaste | 44.7 |

## S.3   IMAGENET SUBSET

As described in Section 4, the 100 Imagenet classes considered in Spurious Imagenet (Neuhaus et al., 2023), we use GPT-4 to generate 32 potential spurious features.

PA Gaps are computed with $K = 50$, and the feature with the largest Gap for each class is presented in Table 17.

To estimate HR Gaps, for each class, we randomly sample 5000 images from the other 99 classes. We compute HR Gaps (with $K = 50$) with the same spurious features, and present the results in Table 18.

Table 17: Results of SpurLens applied to 100 Imagenet classes. For each model and for each class, we present the largest PA Gap (as %, with $K = 50$) and corresponding spurious feature found by SpurLens.

| Imagenet | | Qwen2-VL | | Llama 3.2 | | LLaVa 1.6 | | GPT-4o-mini | |
|---|---|---|---|---|---|---|---|---|---|
| Index | Name | Spur. Feat. | PA Gap | Spur. Feat. | PA Gap | Spur. Feat. | PA Gap | Spur. Feat. | PA Gap |
| 0 | tench | net | 10.0 | reed | 22.0 | fishing rod | 18.0 | reed | 5.3 |
| 2 | great white shark | boat | 7.3 | fish | 18.7 | fish | 20.7 | fish | 14.0 |
| 14 | indigo bunting | branch | 0.0 | branch | 6.7 | stone | 2.0 | branch | 4.7 |
| 42 | agama | rock | 2.7 | rock | 19.3 | rock pile | 18.7 | stone | 9.3 |
| 50 | American alligator | turtle | 4.0 | lily pads | 24.7 | water | 21.3 | water | 17.3 |
| 58 | water snake | cattail | 3.3 | dragonfly | 6.7 | pond | 30.7 | reed | 6.0 |
| 80 | black grouse | grass | 16.0 | grass | 18.0 | grass | 14.7 | grass | 14.7 |
| 81 | ptarmigan | sky | 2.0 | boulder | 4.0 | mountain | 25.3 | sky | 2.0 |
| 82 | ruffed grouse | bush | 12.0 | branch | 34.7 | grass | 14.0 | branch | 26.7 |
| 89 | sulphur-crested cockatoo | feeder | 0.0 | tree | 29.3 | feeder | 8.0 | bush | 4.0 |
| 91 | coucal | nest | 10.0 | bark | 18.0 | tree | 28.7 | nest | 12.0 |
| 94 | hummingbird | sky | 2.0 | sky | 4.0 | feeder | 6.7 | sky | 2.0 |
| 103 | platypus | bubble | 4.7 | bubble | 23.3 | branch | 18.0 | bubble | 9.3 |
| 105 | koala | eucalyptus leaves | 1.3 | foliage | 6.0 | foliage | 2.0 | eucalyptus leaves | 4.0 |
| 147 | grey whale | dolphin | 4.0 | fisherman | 36.0 | dolphin | 29.3 | beach | 16.7 |
| 288 | leopard | fence | 0.0 | tree | 16.0 | waterhole | 8.0 | tree | 18.7 |
| 297 | sloth bear | rock | 4.0 | waterhole | 20.0 | tree | 14.7 | rock | 12.7 |
| 309 | bee | grass | 0.0 | garden | 6.0 | garden | 0.0 | garden | 0.7 |
| 313 | stick insect | stone | 1.3 | water droplet | -1.3 | dirt mound | 1.3 | dirt mound | 3.3 |
| 324 | small white butterfly | grass | 3.3 | garden | 19.3 | garden | 14.7 | wildflower | 3.3 |
| 325 | sulphur butterfly | stone | 6.0 | twig | 6.7 | bee | 8.0 | grass | 5.3 |
| 335 | fox squirrel | tree | 16.0 | tree | 30.0 | tree | 22.7 | tree | 16.0 |
| 351 | hartebeest | muddy ground | 0.0 | rock | 15.3 | waterhole | 18.7 | horizon | 5.3 |
| 364 | three-toed sloth | bird | 2.0 | rock | 13.3 | tree | 12.0 | bird | 2.0 |
| 379 | howler monkey | branch | 4.0 | branch | 37.3 | branch | 5.3 | branch | 10.7 |
| 384 | indri | tree | 12.0 | tree | 20.0 | branch | 21.3 | tree | 60.7 |
| 389 | snoek fish | bait | 7.3 | cooler | 14.7 | bait | 24.0 | tide pool | 21.3 |
| 394 | sturgeon | camera | 1.7 | glove | 5.3 | riverbed | 20.7 | net | 5.3 |
| 395 | gar fish | bait | 12.0 | bait | 27.3 | bait | 13.3 | turtle | 29.3 |
| 400 | academic gown | desk | 2.0 | diploma | 5.3 | diploma | 4.0 | flower | 6.0 |
| 415 | bakery | signpost | 24.0 | car | 38.7 | signpost | 28.0 | signpost | 36.0 |
| 416 | balance beam | camera | 6.0 | camera | 14.0 | rope | 24.7 | camera | 17.3 |
| 419 | Band-Aid | alcohol wipes | 0.7 | alcohol wipes | 18.0 | skin | 12.0 | skin | 12.7 |
| 424 | barbershop | pavement | 2.0 | pavement | 18.0 | tree | 8.0 | flowerpot | 4.0 |
| 425 | barn | bucket | 2.0 | haystack | 6.0 | windmill | 6.0 | windmill | 9.3 |
| 433 | swimming cap | bleacher | 4.7 | paddle | 13.3 | lifeguard | 17.3 | water bottle | 8.0 |
| 434 | bath towel | sink | 2.7 | mirror | 33.3 | shelf | 2.0 | showerhead | 5.3 |
| 435 | bathtub | brush | 4.0 | brush | 14.0 | soap | 48.0 | wall tiles | 10.0 |
| 440 | beer bottle | chip | 2.0 | table | 12.0 | chair | 6.0 | table | 5.3 |
| 445 | bikini | cooler | 5.3 | beach umbrella | 10.7 | cooler | 4.7 | cooler | 8.7 |
| 454 | bookstore | picture frame | 5.3 | picture frame | 14.0 | laptop | 3.3 | laptop | 7.3 |
| 465 | bulletproof vest | picture | 0.0 | tactical belt | 33.3 | tactical belt | 23.3 | tactical belt | 7.3 |
| 466 | high-speed train | overpass | 9.3 | overpass | 14.0 | overpass | 22.7 | overpass | 12.0 |
| 490 | chain mail | tent | 4.7 | stone wall | 14.0 | sword stand | 7.3 | forest | 15.3 |
| 491 | chainsaw | work boots | 4.0 | work boots | 8.7 | trailer | 5.3 | work boots | 9.3 |

*Continued on the next page*

| Imagenet | | Qwen2-VL | | Llama 3.2 | | LLaVa 1.6 | | GPT-4o-mini | |
|---|---|---|---|---|---|---|---|---|---|
| Index | Name | Spur. Feat. | PA Gap | Spur. Feat. | PA Gap | Spur. Feat. | PA Gap | Spur. Feat. | PA Gap |
| 515 | cowboy hat | saddle | 3.3 | guitar | 10.7 | plain | 4.0 | fence | 8.0 |
| 516 | cradle | changing table | 6.0 | nursery rug | 24.0 | nursery rug | 11.3 | photo frame | 7.3 |
| 525 | dam | bridge | 13.3 | bridge | 36.0 | bridge | 14.0 | bridge | 14.7 |
| 537 | dog sled | ski tracks | 2.7 | ski tracks | 28.0 | ski tracks | 21.3 | ski tracks | 23.3 |
| 543 | dumbbell | kettlebell | 6.0 | fitness tracker | 12.7 | floor tiles | 8.0 | yoga block | 6.0 |
| 554 | fireboat | tugboat | 24.7 | coast guard | 29.3 | tugboat | 46.0 | tugboat | 35.7 |
| 557 | flagpole | cloud | 4.0 | bench | 6.0 | cloud | 8.0 | sign | 6.7 |
| 563 | fountain pen | stapler | 2.0 | ink bottle | 8.0 | ink bottle | 8.7 | stamp | 10.0 |
| 565 | freight car | crane | 0.7 | warehouse | 7.3 | crane | 4.0 | warehouse | 3.3 |
| 576 | gondola | flag | 2.0 | canal | 12.7 | paddle | 5.3 | canal | 7.3 |
| 585 | hair spray | bobby pins | 14.3 | bobby pins | 38.0 | bobby pins | 16.7 | conditioner | 18.7 |
| 588 | hamper | wall | 24.7 | cup | 27.3 | curtain | 16.0 | image | 10.7 |
| 592 | hard disk drive | desk | -1.3 | keyboard | 4.0 | printer | 4.7 | wall | -2.7 |
| 602 | gymnastic horizontal bar | exercise balls | 8.7 | ring | 9.3 | jump rope | 7.3 | ring | 7.3 |
| 626 | lighter | cell phone | 0.7 | cell phone | 7.3 | keychain | 16.7 | matchstick | 6.7 |
| 655 | miniskirt | jacket | 6.7 | belt | 14.0 | streetlamp | 32.7 | grass | 4.0 |
| 667 | graduation cap | flag | 4.0 | diploma | 5.3 | gown | 4.7 | diploma | 4.0 |
| 671 | mountain bike | backpack | 4.7 | rock | 12.7 | bush | 8.0 | backpack | 15.3 |
| 678 | neck brace | lamp | 11.3 | wall | 26.7 | magazine | 8.0 | lamp | 6.0 |
| 680 | baby pacifier | teething toy | 72.0 | teething toy | 75.3 | teething toy | 40.0 | teething toy | 72.7 |
| 682 | obelisk | sidewalk | 2.0 | mountain | 4.0 | sidewalk | 2.0 | cloud | 3.3 |
| 684 | ocarina | headphone | 2.7 | photo | -1.3 | table | 0.0 | headphone | 5.3 |
| 695 | padlock | bag | 2.0 | post | 6.0 | box | 4.7 | gate | 2.0 |
| 702 | parallel bars | pull-up bar | 9.3 | wrist tape | 15.3 | pull-up bar | 20.0 | pull-up bar | 10.7 |
| 709 | pencil case | clip | 12.0 | clip | 36.0 | clip | 58.7 | clip | 8.7 |
| 720 | pill bottle | first aid kit | 4.7 | inhaler | 19.3 | glass of water | 11.3 | inhaler | 6.7 |
| 722 | ping-pong ball | referee | 7.3 | racket | 25.3 | racket | 36.0 | floor | 8.0 |
| 728 | plastic bag | tissue | 2.0 | water bottle | 8.7 | tissue | 3.3 | tissue | 2.7 |
| 731 | plunger | soap | 12.7 | faucet | 22.0 | tile | 42.0 | trashcan | 12.7 |
| 733 | pole | road | 3.3 | cloud | 12.0 | cloud | 9.3 | cloud | 8.0 |
| 737 | soda bottle | table | 2.7 | chair | 20.0 | chair | 14.0 | straw | 4.7 |
| 738 | plant pot | vase | 16.7 | vase | 15.3 | vase | 19.3 | vase | 12.0 |
| 739 | potter's wheel | display | 2.7 | brush | 21.3 | clay | 12.6 | water | 13.3 |
| 746 | hockey puck | helmet | 9.3 | skate | 13.3 | skate | 18.7 | game clock | 16.7 |
| 749 | quill | book | 21.3 | paper | 53.3 | paper | 22.7 | paper | 32.0 |
| 752 | racket | fence | 2.7 | fence | 8.0 | bench | 7.3 | court | 4.7 |
| 755 | radio telescope | sky | 6.0 | sky | 31.3 | sky | 20.0 | sky | 13.3 |
| 756 | rain barrel | siding | 4.7 | siding | 30.7 | siding | 12.7 | siding | 18.0 |
| 766 | rotisserie | bread rolls | 11.3 | food baskets | 19.3 | window | 14.7 | bread rolls | 18.0 |
| 767 | eraser | marker | 9.3 | highlighter | 16.7 | sticky note | 24.0 | marker | 6.0 |
| 771 | safe | electronic devices | 12.0 | electronic devices | 24.0 | electronic devices | 11.3 | electronic devices | 16.0 |
| 776 | saxophone | bass guitar | 4.0 | music stand | 4.0 | photo | 20.7 | microphone | 14.0 |
| 785 | seat belt | driver | 2.0 | headrest | 7.3 | rearview mirror | 6.0 | driver | 2.7 |
| 788 | shoe store | mirror | 10.0 | poster | 37.3 | mirror | 19.3 | poster | 22.0 |
| 792 | shovel | soil | 4.7 | wheelbarrow | 6.7 | rock | 8.0 | garden tool | 6.7 |
| 793 | shower cap | comb | 6.0 | towel | 19.3 | mirror | 6.7 | comb | 8.7 |
| 797 | sleeping bag | tent | 7.3 | hiking boots | 18.7 | campsite | 17.3 | ground mat | 14.7 |
| 801 | snorkel | sand | 9.3 | beach | 14.0 | goggles | 19.3 | beach | 23.3 |
| 802 | snowmobile | trail markers | 2.0 | snow shovel | 16.0 | snow | 68.0 | snow shovel | 8.0 |
| 803 | snowplow | utility pole | 2.0 | road | 8.0 | snowdrift | 21.3 | snowdrift | 6.0 |
| 822 | steel drum | lamp | 2.0 | sunglass | 10.0 | guitar | 5.3 | guitar | 8.0 |
| 827 | stove | spoon | 9.3 | pan | 20.0 | sink | 19.3 | pan | 16.7 |
| 867 | semi-trailer truck | traffic light | 4.0 | pedestrian | 12.7 | pedestrian | 6.0 | concrete barrier | 14.0 |
| 908 | airplane wing | propeller | 3.3 | propeller | 6.0 | sea | 2.0 | sky | 7.3 |
| 933 | cheeseburger | burger wrapper | 10.7 | burger wrapper | 25.3 | burger wrapper | 38.0 | burger wrapper | 36.0 |

Table 18: For each of the 100 selected Imagenet classes, we randomly sample 5000 images outside the class and apply SpurLens. We report the largest HR Gaps (as %, with $K = 50$) found and the associated spurious features.

| Imagenet | | Qwen2-VL | | Llama 3.2 | | LLaVa 1.6 | | GPT-4o-mini | |
|---|---|---|---|---|---|---|---|---|---|
| Index | Name | Spur. Feat. | PA Gap | Spur. Feat. | PA Gap | Spur. Feat. | PA Gap | Spur. Feat. | PA Gap |
| 0 | tench | net | 6.0 | lure | 2.7 | lure | 20.7 | net | 2.0 |
| 2 | great white shark | swimmer | 3.3 | net | 0.0 | net | 0.0 | net | 0.0 |
| 14 | indigo bunting | feeder | 7.3 | branch | 0.0 | feeder | 11.3 | branch | 0.0 |
| 42 | agama | mud | 0.7 | tree | 0.7 | grass | 31.3 | stone | 0.0 |
| 50 | American alligator | water | 1.3 | tree | 0.0 | water | 0.7 | water | 4.0 |
| 58 | water snake | reed | 4.0 | reed | 2.0 | shell | 3.3 | shell | 1.3 |
| 80 | black grouse | grass | 8.7 | grass | 2.7 | tree | 18.0 | grass | 3.3 |
| 81 | ptarmigan | grass | 24.0 | snowdrift | 6.0 | grass | 14.7 | grass | 9.3 |
| 82 | ruffed grouse | moss | 16.0 | moss | 3.3 | moss | 13.3 | moss | 8.7 |
| 89 | sulphur-crested cockatoo | tree | 2.0 | feeder | 0.0 | tree | 22.7 | feeder | 0.0 |
| 91 | coucal | rock | 1.3 | grass | 1.3 | tree | 30.7 | grass | 5.3 |
| 94 | hummingbird | garden | 0.7 | sky | 0.7 | petal | 32.7 | tree | 0.0 |

| Imagenet | | Qwen2-VL | | Llama 3.2 | | LLaVa 1.6 | | GPT-4o-mini | |
|---|---|---|---|---|---|---|---|---|---|
| Index | Name | Spur. Feat. | PA Gap | Spur. Feat. | PA Gap | Spur. Feat. | PA Gap | Spur. Feat. | PA Gap |
| 103 | platypus | frog | 2.0 | sand | 0.7 | lily pads | 1.3 | lily pads | 0.7 |
| 105 | koala | branch | 0.0 | branch | 0.0 | foliage | 6.0 | foliage | 2.0 |
| 147 | grey whale | fisherman | 14.7 | surfboard | 0.7 | fisherman | 4.0 | surfboard | 2.7 |
| 288 | leopard | fence | 0.0 | fence | 0.0 | tree | 6.0 | fence | 0.0 |
| 297 | sloth bear | tree | 14.0 | tree | 0.7 | tree | 45.3 | waterhole | 0.0 |
| 309 | bee | leaf | 2.0 | flower | 0.7 | flower | 5.3 | leaf | 2.0 |
| 313 | stick insect | water droplet | 6.0 | dirt mound | 0.0 | dirt mound | 0.0 | dirt mound | 0.0 |
| 324 | small white butterfly | wildflower | 11.3 | tree | 1.3 | wildflower | 14.7 | leaf | 2.0 |
| 325 | sulphur butterfly | petal | 22.0 | petal | 6.0 | petal | 36.7 | garden | 2.0 |
| 335 | fox squirrel | acorn | 2.7 | acorn | 0.7 | tree | 16.0 | tree | 0.0 |
| 351 | hartebeest | muddy ground | 0.0 | tree | 2.0 | grass | 6.7 | muddy ground | 0.0 |
| 364 | three-toed sloth | tree | 4.0 | tree | 0.7 | tree | 25.3 | tree | 0.0 |
| 379 | howler monkey | tree | 28.0 | tree | 0.7 | tree | 41.3 | tree | 4.0 |
| 384 | indri | tree | 2.0 | rock | 0.0 | tree | 41.3 | tree | 8.0 |
| 389 | snoek fish | seaweed | 24.0 | sea surface | 2.0 | seaweed | 30.0 | rod | 4.7 |
| 394 | sturgeon | fish | 32.0 | fish | 6.7 | fish | 48.7 | fish | 12.0 |
| 395 | gar fish | turtle | 26.0 | turtle | 11.3 | turtle | 50.0 | bird | 0.0 |
| 400 | academic gown | diploma | 27.3 | diploma | 30.7 | diploma | 32.0 | diploma | 25.3 |
| 415 | bakery | car | 2.7 | car | 5.3 | car | 3.3 | car | 1.3 |
| 416 | balance beam | gymnast | 73.3 | gymnast | 36.7 | gymnast | 44.7 | rope | 4.7 |
| 419 | Band-Aid | blood | 9.3 | alcohol wipes | 6.7 | skin | 6.7 | ruler | 4.0 |
| 424 | barbershop | bus stop | 4.0 | bus stop | 4.0 | flowerpot | 0.0 | fire hydrant | 0.7 |
| 425 | barn | fence | 2.0 | birdhouse | 11.3 | fence | 9.3 | fence | 2.7 |
| 433 | swimming cap | towel | 10.0 | pool | 2.0 | snorkel | 32.0 | water bottle | 2.7 |
| 434 | bath towel | soap | 30.0 | soap | 33.3 | soap | 38.7 | soap | 20.0 |
| 435 | bathtub | shower cap | 11.3 | toilet | 11.3 | showerhead | 2.7 | shampoo | 6.0 |
| 440 | beer bottle | table | 20.7 | table | 9.3 | glass | 14.0 | cup | 8.7 |
| 445 | bikini | snorkel | 28.0 | pool | 23.3 | pool | 24.0 | snorkel | 20.0 |
| 454 | bookstore | clock | 2.0 | clock | 6.7 | picture frame | 2.7 | picture frame | 1.3 |
| 465 | bulletproof vest | gun | 4.7 | gun | 2.0 | helmet | 22.7 | floor | 0.0 |
| 466 | high-speed train | construction barriers | 0.0 | construction barriers | 0.0 | construction barriers | 0.0 | construction barriers | 0.0 |
| 490 | chain mail | sword stand | 7.3 | torch | 4.7 | sword stand | 12.7 | shield | 2.0 |
| 491 | chainsaw | work boots | 4.7 | woodpile | 1.3 | workbench | 12.0 | work boots | 2.0 |
| 515 | cowboy hat | horse | 10.7 | whip | 7.3 | guitar | 16.7 | whip | 4.0 |
| 516 | cradle | bedside table | 5.3 | stroller | 10.0 | infant toys | 6.7 | baby blanket | 4.0 |
| 525 | dam | waterfall | 5.3 | waterfall | 6.7 | waterfall | 10.7 | waterfall | 5.3 |
| 537 | dog sled | ski tracks | 13.3 | pine trees | 0.7 | pine trees | 2.7 | canoe | 0.0 |
| 543 | dumbbell | weight plate | 2.7 | weight plate | 2.0 | weight plate | 3.3 | weight plate | 2.0 |
| 554 | fireboat | paddle | 2.0 | pier | 0.7 | ramp | 2.0 | paddle | 0.7 |
| 557 | flagpole | building | 12.7 | building | 20.0 | building | 13.3 | building | 10.7 |
| 563 | fountain pen | paperclip | 16.7 | paper | 16.0 | paper | 34.7 | paper | 8.7 |
| 565 | freight car | road | 6.0 | road | 8.7 | road | 22.0 | grass | 0.0 |
| 576 | gondola | bridge | 8.0 | bridge | 3.3 | poster | 12.7 | tree | 0.0 |
| 585 | hair spray | shampoo | 31.3 | shampoo | 17.3 | shampoo | 42.0 | shampoo | 5.3 |
| 588 | hamper | wall | 1.3 | curtain | 15.3 | wall | 12.0 | curtain | 6.0 |
| 592 | hard disk drive | router | 6.7 | printer | 12.7 | router | 7.3 | desk | 2.0 |
| 602 | gymnastic horizontal bar | bench | 15.3 | bench | 10.7 | bench | 13.3 | banner | 7.3 |
| 626 | lighter | matchbox | 26.0 | cigarette | 18.0 | cigarette | 25.3 | keychain | 1.3 |
| 655 | miniskirt | tights | 12.0 | earring | 6.0 | tights | 10.0 | tights | 10.7 |
| 667 | graduation cap | diploma | 31.3 | diploma | 30.0 | diploma | 36.0 | diploma | 30.0 |
| 671 | mountain bike | water bottle | 2.7 | backpack | 4.7 | backpack | 4.0 | glove | 2.0 |
| 678 | neck brace | cup | 2.7 | remote | 3.3 | cup | 2.0 | wall | 0.7 |
| 680 | baby pacifier | teddy bear | 16.0 | diaper | 4.0 | crib | 22.0 | crib | 2.0 |
| 682 | obelisk | lamp post | 11.3 | statue | 12.0 | statue | 20.0 | pathway | 2.0 |
| 684 | ocarina | photo | 1.3 | microphone | 2.7 | headphone | 2.7 | headphone | 0.0 |
| 695 | padlock | wall | 4.0 | door | 14.7 | chain | 25.3 | wall | 4.0 |
| 702 | parallel bars | leotard | 40.7 | exercise bands | 24.0 | exercise bands | 30.7 | exercise bands | 24.0 |
| 709 | pencil case | notebook | 19.3 | notebook | 12.0 | notebook | 14.7 | notebook | 6.0 |
| 720 | pill bottle | tissue | 14.7 | tissue | 11.3 | thermometer | 9.3 | coffee cup | 0.0 |
| 722 | ping-pong ball | referee | 4.7 | scoreboard | 2.7 | racket | 14.7 | scoreboard | 2.0 |
| 728 | plastic bag | tissue | 26.0 | tissue | 24.0 | trash can | 25.3 | trash can | 10.0 |
| 731 | plunger | cleaner | 11.3 | brush | 12.0 | pipe | 27.3 | brush | 1.3 |
| 733 | pole | street | 54.0 | building | 52.0 | car | 46.7 | street | 52.0 |
| 737 | soda bottle | table | 14.0 | chair | 18.0 | cup | 14.0 | cup | 13.3 |
| 738 | plant pot | watering can | 18.7 | window | 20.7 | watering can | 34.7 | watering can | 23.3 |
| 739 | potter's wheel | sculpture | 2.7 | shelf | 2.0 | jug | 6.7 | jug | 0.0 |
| 746 | hockey puck | jersey | 2.0 | penalty box | 2.7 | skate | 2.0 | penalty box | 1.3 |
| 749 | quill | notebook | 0.7 | candle | 2.0 | notebook | 15.3 | ink bottle | 4.7 |
| 752 | racket | ball | 21.3 | ball | 20.7 | ball | 22.0 | ball | 21.3 |
| 755 | radio telescope | satellite | 2.0 | satellite | 15.3 | power lines | 6.0 | bench | 0.0 |
| 756 | rain barrel | potted plants | 8.0 | shed roof | 13.3 | garden statue | 19.3 | potted plants | 4.7 |
| 766 | rotisserie | cutlery | 4.7 | serving trays | 4.7 | cutlery | 5.3 | plant | 0.0 |
| 767 | eraser | notebook | 14.7 | notebook | 12.7 | paper | 46.0 | notebook | 9.3 |
| 771 | safe | electronic devices | 1.3 | bookshelf | 14.7 | clock | 0.0 | window | 0.0 |
| 776 | saxophone | guitar | 6.0 | guitar | 2.7 | guitar | 4.7 | amplifier | 0.0 |
| 785 | seat belt | headrest | 15.3 | rearview mirror | 8.0 | driver | 8.7 | dashboard | 0.0 |
| 788 | shoe store | vendor cart | 7.3 | clock | 3.3 | poster | 2.0 | vendor cart | 2.7 |
| 792 | shovel | wheel | 16.0 | wheel | 11.3 | wheel | 23.3 | wheel | 14.0 |

*Continued on the next page*

| Imagenet | | Qwen2-VL | | Llama 3.2 | | LLaVa 1.6 | | GPT-4o-mini | |
|---|---|---|---|---|---|---|---|---|---|
| Index | Name | Spur. Feat. | PA Gap | Spur. Feat. | PA Gap | Spur. Feat. | PA Gap | Spur. Feat. | PA Gap |
| 793 | shower cap | bathrobe | 15.3 | mirror | 10.7 | towel | 62.7 | towel | 3.3 |
| 797 | sleeping bag | camp chair | 14.7 | camp chair | 11.3 | camp chair | 16.0 | backpack | 4.7 |
| 801 | snorkel | buoy | 8.7 | buoy | 12.7 | wetsuit | 13.3 | beach | 0.7 |
| 802 | snowmobile | winter jacket | 4.7 | trail markers | 4.7 | trail markers | 7.3 | winter jacket | 1.3 |
| 803 | snowplow | shovel | 8.7 | glove | 0.7 | shovel | 8.7 | shovel | 3.3 |
| 822 | steel drum | speaker | 10.7 | fence | 10.7 | fence | 11.3 | beachball | 0.7 |
| 827 | stove | pan | 18.0 | dishwasher | 17.3 | pan | 14.7 | cabinet | 12.7 |
| 867 | semi-trailer truck | construction site | 16.7 | construction site | 22.0 | construction site | 40.7 | construction site | 5.3 |
| 908 | airplane wing | propeller | 5.3 | propeller | 4.7 | propeller | 3.3 | propeller | 1.3 |
| 933 | cheeseburger | napkin | 0.0 | burger wrapper | 2.0 | burger wrapper | 2.0 | napkin | 0.0 |

