# OpenReview forum: "SpurLens: Automatic Detection of Spurious Cues in Multimodal LLMs"
_ICLR.cc/2026/Conference — Submitted to ICLR 2026_

### Official Review · Reviewer_g5KA · 2025-10-25

**Soundness:** 3
**Presentation:** 2
**Contribution:** 2
**Rating:** 4
**Confidence:** 5

**Summary:**

This paper extends the ICLR 2025 Workshop paper “SpurLens: Finding Spurious Correlations in Multimodal LLMs.” It presents a framework combining GPT-4 and open-set object detection to automatically discover and quantify spurious visual correlations that affect MLLMs. SpurLens identifies interpretable spurious cues and computes “Spurious Gaps” in perception accuracy and hallucination rate.

**Strengths:**

1. Spurious correlations remain a genuine and pressing issue in MLLMs, and this work contributes a practical diagnostic framework that could be broadly useful to the community.

2. The paper appropriately frames its findings as diagnostic rather than as mitigation breakthroughs; its conclusion, that prompting alone cannot effectively address spurious correlations in MLLMs, is meaningful.

**Weaknesses:**

1. The contribution is incremental. The main novelty is the inclusion of more base models and expanded analyses (vision-encoder ablation, prompt variants), but the core method, evaluation setup, and key conclusions are essentially unchanged from the ICLR 2025 workshop version. Many figures, metrics, and formulations (PA, HR, Spurious Gap) are nearly identical to the workshop paper, making this feel like a polished version rather than a new contribution.

2.  Since SpurLens depends heavily GPT-4 outputs, errors or biases in those models may propagate without strong theoretical justification for robustness.

3. The study concludes that CoT and prompting fail to fix spurious bias, but provides no new mitigation direction.

**Questions:**

1. The paper identifies spurious gaps, but what insights do the authors have into why certain spurious features dominate or how aspects of the training data lead to these biases?

2. Can SpurLens be extended beyond evaluation to support training-time mitigation or data curation, leveraging its diagnostic findings to improve model robustness?

---

> ### Author Response · Authors · 2025-11-23
> **Response to Reviewer g5KA**
>
> We appreciate the reviewer’s recognition that spurious correlations remain a pressing issue for MLLMs and that SpurLens provides a practical diagnostic framework that can be broadly useful to the community. Below, we address the points raised by the reviewer:
>
> ---
>
> ### **Weakness 1.**
> While this work uses the ICLR 2025 workshop paper as a foundation, it significantly expands on the main themes and introduces dozens of additional experiments and ablations, ensuring the reliability of our methodology and providing several new insights that advance our understanding of the causes of spurious bias in MLLMs.
>
> The ICLR 2025 workshop paper only presents a core pipeline methodology and object perception experiments on HardImageNet and COCO. We include a subset of 100 ImageNet classes in our evaluation of the PA Gap. Additionally, we examine how spurious correlations impact object hallucination through the HR Gap via experiments on COCO and an ImageNet subset (the workshop paper contained no discussion of hallucination). The pipeline in the workshop paper was notably not fully-automatic: it required manual inspection of object detection results before proceeding with MLLM evaluation; furthermore, the use of GPT-4 and OWLv2 for spurious cue proposal and object detection respectively are not validated. Our pipeline is fully-automatic, requiring no human supervision, and we validate the use of GPT-4 and OWlv2 in Appendices E and F respectively, and provide further validation of our results in Appendices I, L, and N.
>
> All of the ablations in Section 6 (and associated Appendices), including the token-dropping experiments to corroborate the HR Gaps, the unsuccessful attempts at language-based mitigation, and the vision encoder experiments, are all unique to our work. Combined, these results provide a better understanding of the source of spurious bias in MLLMs, while the workshop paper merely frames its computation of PA Gaps as a recognition of spurious bias in MLLMs.
>
> ---
>
> ### **Weakness 2.**
> We emphasize that SpurLens does not rely on GPT-4 being perfectly correct or complete; the LLM only proposes candidate spurious cues, and the subsequent stages of SpurLens automatically filter out proposals, as described in Section 4 and Appendix A.
>
> In Appendix E, we compare the spurious feature proposals of GPT-4 to feature proposals by Qwen 2.5 72B and Gemini 2.5 Flash. We find significant overlap between the spurious cues proposed; the consistency of the results despite the diversity of training data between these models suggests that the choice of general-purpose LLM does not significantly affect the spurious cues obtained or the Spurious Gaps measured.
>
> Additionally, as discussed at the end of Section 4, we emphasize that SpurLens prioritizes precision over completeness. Our goal is to consistently find strong spurious cues for (MLLM, target object) combinations, rather than find the absolute best spurious cue for each combination. While GPT-4 may not propose the best spurious feature for every target object, its general world knowledge is able to propose strong spurious features for common objects (as we empirically observe in our main results).
>
> ---
>
> ### **Question 1.**
>
> Our experiments suggest that certain spurious features lead to large spurious gaps because they correspond to ambiguous or entangled representations in the vision encoder.
>
> In Section 6.2, we experiment with various prompting techniques, including Chain-of-Thought, ensembling, and informing the MLLM of spurious features (and, in the “Spurious Top” prompt, even “cheating” by utilizing SpurLens’ analysis of the MLLM’s performance on the whole dataset). In all cases, we did not find a significant decrease in either the PA Gap or HR Gap, providing evidence that prompting techniques and text-based guidance alone are unable to alleviate spurious bias. In Section 6.3 and Appendix M, we isolate the MLLM vision encoder and perform experiments similar to our PA evaluation on binary classifiers. We are able to obtain similar results compared to our main MLLM experiments, and argue that spurious bias in MLLMs mainly arises due to biases in the vision embeddings, which are influenced by dataset size and composition. We have extended Appendix M with an additional experiment where we compare the PA Gaps on HardImageNet using SpurLens and using vision encoder classifiers; we find a correlation, suggesting that the vision encoder is partially responsible for the observed spurious bias.
>
> ---
>
> ### **Weakness 3 / Question 2.**
> Our insights indicate that simple SFT fine-tuning of the LLM would not address the underlying issue, and that effective mitigation will likely require modifying the vision encoder or multimodal fusion components rather than the language pathway. While we do not propose a mitigation algorithm in this work, SpurLens provides precisely the diagnostic signals: interpretable spurious cues and high/low-spurious image splits that such methods could leverage.

---

### Official Review · Reviewer_DJ5w · 2025-10-31

**Soundness:** 3
**Presentation:** 3
**Contribution:** 2
**Rating:** 6
**Confidence:** 4

**Summary:**

SpurLens introduces an automatic pipeline for detecting multi-modal spurious correlations in
multi-modal large language models (MLLMs) under object recognition tasks. The study primarily
uses GPT-4 and OWLv2 to extract and rank image features. Experiments on HardImageNet, an
ImageNet subset, and the COCO dataset demonstrate the effectiveness of the extracted
features in increasing perception accuracy (PA) and hallucination rates (HR), indicating
a strong (spurious) correlation between the extracted features and the corresponding object class. Three open-source models and GPT-4o are tested against extracted biases, and all demonstrate significant reliance on bias. Further testing of the proposed pipeline with different backbones
and human verification further supports the method's credibility. Ablation studies of hallucination
with dropped object patches, prompt-based mitigation, and visual spurious cues during
encoding extend the findings of this work.

**Strengths:**

1. SpurLens focuses on spurious bias in MLLMs —an underexplored field that could
enhance the robustness of MLLM applications.
2. SpurLens is a fully automated pipeline that extracts spurious features that significantly
degrade MLLMs’ detection performance. Such scalability indicates broad potential
applications.
3. Human verification and cross-backbone verification enhance the design's soundness.

**Weaknesses:**

1. The feature filter is required for the OWLv2 object detector to function. However, the
filtered-out features could be valuable for studying spurious bias, e.g., the backgrounds
in the WaterBirds dataset.
2. The extraction of spurious bias is limited to object detection, whereas MLLMs have a
broader capacity. Modern MLLMs need to handle tasks that require accurate processing
of information beyond objects’ presence or absence, e.g., time, actions, and scenes.
This work's scope is comparably limited in this sense, and would not help the tasks
above as a result.
3. The patch drop method is debatable, as missing patch tokens fail to pass positional
information to the transformer blocks. Zero-filling on the patches could have been a more
robust approach, as it reduces the domain shift that might have contributed to the observations.

**Questions:**

1. Does the PA gap remain high if the target object is cropped out with OWLv2 and used as input? I am
curious whether the images that contain spurious features depict intrinsically easier
class object instances, contributing to the PA gap.
2. Do HR and PA gaps correlate across target objects? What other insights can we draw from the split of PA and HR compared to their sum, which would also reveal the detection bias?
3. How does your ranking work when there are more than K images with zero scores?

---

> ### Author Response · Authors · 2025-11-23
> **Response to Reviewer DJ5w**
>
> We thank the reviewer for recognizing the importance of studying spurious bias in MLLMs and for noting the value of SpurLens as a fully automated, scalable diagnostic pipeline supported by strong human and cross-backbone verification.
>
> ---
>
> ### **Weakness 1.**
>
> The filtering of spurious cues does eliminate some specific classes of spurious cues. For example, cues such as “sunlight” and “walk” are excluded (not easily visualizeable), and spurious cues that are visually part of their target objects are excluded (such as cue “handlebar” for object “bicycle”). These limited and targeted exclusions (enumerated in Appendix A) are important for object detection robustness; the reliability of OWLv2 for object detection in our pipeline is validated under these exclusions in Appendix F.
>
> However, as discussed at the end of Section 4, SpurLens emphasizes precision, rather than completeness. Our goal is to consistently and efficiently find strong spurious cues for given MLLMs and target objects, rather than the absolute best cue for any given combination. Tables 1 and 2 and Figure 5 show that SpurLens is successful in this regard: with the given filtering restrictions, we are able to consistently identify high-spurious images for various MLLMs on a wide variety of common target objects.
>
> ---
>
> ### **Weakness 2.**
>
> Our aim is to isolate high-level visual spurious cues that affect an MLLM’s perceptual grounding. Object perception is a fundamental perceptual primitive that underlies many higher-level tasks, including relative positioning, actions, counting, temporal reasoning, and causal inference. If a model’s visual representation or its use of visual information are impaired by spurious cues, then that bias will naturally propagate into many downstream capabilities.
>
> The goal of our work is to provide a diagnostic foundation for studying spurious bias at-scale in MLLMs. Our experiments and ablations in Sections 6.2 and 6.3 argue that visual spurious bias is due to the vision encoder representation, and cannot be remedied through prompt structure or language-based reasoning, which is common when dealing with higher-level tasks. Recognizing these lower-level biases, and gaining a better understanding of where they originate, is essential for improving at higher-level understanding and more complex tasks.
>
> Finally, while our pipeline focuses on object-level cues, our pipeline is modular and flexible. Our spurious cue proposal and detection could likely be extended beyond associated objects and background elements (as we consider), to actions, scenes, and other simple relationships that are essential in reasoning tasks.
>
> ---
>
> ### **Weakness 3.**
>
> Our token dropping implementation does pass positional information to the MLLM. As described in Appendix K.1, when dropping image tokens, we take care to apply the same indexing to the rotary positional embeddings, so that the image tokens that remain have the same positional embeddings applied as normal. For Qwen2-VL, in addition to the tokens containing the target object, we drop all tokens that are in the same 2x2 patches because Qwen2-VL performs 2x2 patch compression using an MLP. (Similar corresponding changes are made to all models separately, as they have slightly different architectures; the exact changes are included in our supplementary code submission.)
>
> In Appendix K.2, we perform some simple tests to demonstrate the outcome of token-dropping. If we provide the MLLM with an image where the target object pixels have been zero-filled, the MLLM is able to recognize that the image has been altered. However, if we drop the tokens containing the target object, then the MLLM does not report anything wrong with the image, indicating that the remaining tokens (along with their positional information) are not affected. With this experiment, we observe that zero-filling the object mask also introduces a significant domain shift that the model can recognize, and may affect perception. [1] similarly argues that zero-filling pixels can introduce “missingness bias” in inference, and that simply removing image tokens in transformer-based models (as we do) can alleviate this.
>
> [1] Jain et al., “Missingness bias in model debugging”, ICLR 2022.

---

> ### Author Response · Authors · 2025-11-23
> **Response to Reviewer DJ5w**
>
> ### **Question 1.**
>
> We thank the reviewer for their suggestion of this experiment. We perform the PA evaluation on HardImageNet. Rather than use OWLv2 to extract bounding boxes for the target objects, we instead utilize the ground-truth masks provided by HardImageNet and crop to the smallest box that contains the target object mask for each image. We then perform MLLM evaluation on these images; the class-wise averaged results are presented in the table below. We note that the PA Gaps are all near-zero, but negative. This provides evidence that high-spurious images do not necessarily contain more difficult target object representations compared to low-spurious images.
>
> | Model | $\text{PA}_{\text{s}}$ | $\text{PA}_{\text{c}}$ | $\text{PA}$ Gap |
> |-------|------|------|--------|
> | Qwen2-VL  | 92.5 | 94.7 | -2.2   |
> | Llama 3.2 | 67.8 | 72.7 | -4.9   |
> | LLaVa 1.6 | 78.5 | 80.3 | -1.8   |
>
> ---
>
> ### **Question 2.**
>
> While our method of analysis is similar in both cases, PA and HR spurious gaps capture different MLLM failure modes (perception failure and hallucination), and as such need not be correlated across objects. In our class-wise results (presented in Appendix S), many classes exhibit a high PA Gap but small HR Gap (reliance on spurious cues in context for recognition, but not a tendency to hallucinate), while others show the reverse. Additionally, the highest spurious cues found for the same target object in both cases may be different. This asymmetry is visible in our classwise distribution of PA and HR Gaps (Figure 5), which supports our treatment of PA and HR as complementary, rather than redundant, metrics. In line with this, computing the correlation of PA and HR Gaps class-wise based on the results in Appendix S reveals weak, model-dependent linear correlations (under 0.15 for all four models).
>
> The reviewer’s suggestion of considering the sum (or average) of the PA and HR Gaps can be viewed as an overall detection bias and general quality metric for an MLLM. However, this conflation of the two failure modes does not capture the granularity of our analysis; our split metrics allows us to isolate the (potentially distinct) causes of failure in both cases. In practice, we observe that models can exhibit one bias more strongly than the other, and this separation is essential for interpreting and analyzing MLLM visual failures.
>
> ---
>
> ### **Question 3.**
>
> In our implementation, to avoid bias, we randomly shuffle the dataset order before sorting by $f_i\text{-score}$ and selecting the top-$K$ and bottom-$K$. If there are more than $K$ images with zero object-detection scores, then we are essentially selecting $K$ of them uniformly for MLLM evaluation. The hyperparameter $K$ controls the precision of our Spurious Gap estimates; utilizing a sample of all images not containing the particular spurious cue is necessary for efficient estimation of the Gap. In Appendix I, we examine the sensitivity of our results with respect to $K$, and find stability at the magnitudes of $K$ that we use for analysis, suggesting that the permutation of zero-score images is generally not a concern.

---

### Official Review · Reviewer_yEFM · 2025-10-31

**Soundness:** 3
**Presentation:** 3
**Contribution:** 2
**Rating:** 4
**Confidence:** 4

**Summary:**

This paper proposes SpurLens, an automatic pipeline to detect spurious correlations in multimodal large language models (MLLMs). The method uses GPT-4 to suggest potential spurious cues associated with object classes and OWLv2 as an open-set detector to quantify their co-occurrence with these objects. Based on this, the framework ranks images by spuriosity and evaluates the perception accuracy (PA) and hallucination rate (HR) differences across the ranking. Experiments on COCO, ImageNet, and HardImageNet show consistent spurious gaps across models including GPT-4o-mini, Qwen2-VL, LLaVA-1.6, and Llama-3.2-Vision. Additional analyses such as patch dropping and prompt-based mitigation highlight that removing spurious cues lowers performance but does not eliminate model dependence on them. The framework is timely, but its contribution is mainly an automation of existing ideas rather than a fundamentally new perspective.

**Strengths:**

1. The end-to-end framework is well constructed and reproducible, combining large language models with open-set detectors to extract spurious cues automatically.
2. Cross-model and cross-backbone verification strengthens the reliability of the findings.
3. The experiments consistently reveal strong spurious dependencies, especially in models that exhibit hallucination under complex visual contexts.
4. The scalability of the method makes it a useful diagnostic tool for future robustness studies.

**Weaknesses:**

1. The conceptual novelty is limited, as the paper builds directly on prior works in spurious bias analysis and multimodal robustness [1,2].
2. The OWLv2 detector may itself encode contextual/spurious bias, which could affect the measured spuriosity.
3. The study is limited to object-level recognition and does not address relational or compositional tasks that better reflect multimodal reasoning. There are more spurious correlation such as color or textual biases.
4. The patch-dropping experiment may introduce a domain shift rather than isolating causal visual features.
5. The analysis primarily confirms existing findings without offering a deeper explanation of why these biases emerge.

[1] Moayeri et al., “Spuriosity Rankings: Sorting Data to Measure and Mitigate Biases,” NeurIPS 2023. \
[2] Varma et al., “RAVL: Discovering and Mitigating Spurious Correlations in Fine-Tuned Vision-Language Models,” ICCV 2024.

**Questions:**

1. How consistent are the discovered spurious cues when using a different detector such as Grounding-DINO or DETR?
2. Do GPT-4-generated cues tend to favor specific linguistic patterns or cultural biases that could influence the detected correlations?
3. How much overlap exists between cues found by SpurLens and those manually identified in benchmarks such as MM-SpuBench or Waterbirds?
4. What happens if the cue filtering is relaxed, do we recover subtler correlations missed under stricter thresholds?
5. Could causal probing or concept activation vector [3] analysis complement the current ranking-based approach?

[3] B. Kim et al., “Interpretability Beyond Feature Attribution: Quantitative Testing with Concept Activation Vectors (TCAV),” ICML 2018.

---

> ### Author Response · Authors · 2025-11-23
> **Response to Reviewer yEFM**
>
> We appreciate the reviewer’s positive assessment of the SpurLens framework, its reproducibility, and the consistency of our findings across models, as well as the recognition of its scalability as a diagnostic tool.
>
> ---
>
> ### **Weakness 1.**
>
> Our work introduces the first automated pipeline for discovering interpretable spurious visual cues in MLLMs, which is different from both cited works. While we take inspiration from Spurious ImageNet regarding ranking images by spuriosity for evaluation, Spurious ImageNet only operates on vision classifiers, not on VLMs (contrastive or autoregressive). Additionally, their method is not fully automatic, requiring a human-in-the-loop to decide which cues are truly spurious; in contrast, our method is fully automatic and can efficiently and reliably identify high-level spurious cues without human involvement. RAVL analyzes contrastive (CLIP-like) VLMs, while our analysis centers on autoregressive MLLMs; this affords us the ability to separately consider object perception and object hallucination failures, rather than general misclassification errors. Additionally, RAVL utilizes clustering to identify spurious cues in representation space. These spurious features are not interpretable, while ours have natural-language descriptors; additionally, they require access to the vision encoder, while our method does not and can be used to analyze black-box MLLMs (such at GPT-4o-mini).
>
> ---
>
> ### **Weakness 2.**
>
> We validate the accuracy of OWLv2 object detection in Appendix F through two human validation studies. In Appendix F.1, we randomly choose (target object, spurious cue) pairs from our COCO and ImageNet results, and manually label the top and bottom-ranked images. We find that OWLv2 and the human annotators agree on the presence/absence of the spurious cue in 90% and 93% of images (for ImageNet and COCO respectively). In Appendix F.2, we replace OWLv2 object detections with human annotations when computing the Spurious Gap, and find that the results closely match SpurLens’ output. In addition, we introduce Appendix O, where we consider alternative open-vocabulary object detectors such as GroundingDINO and YOLO-World. We find that, while there is significant (though class/cue-dependent) correlation between the $f_i\text-{scores}$ produced by these object detectors, OWLv2 strikes the best balance in terms of confidence scores (GroundingDINO tends to have many false positives, while YOLO-World tends to give very conservative confidence scores). These results reaffirm that OWLv2 is a reliable object detector for our use case.
>
> ---
>
> ### **Weakness 3.**
>
> Our aim is to isolate high-level visual spurious cues that affect an MLLM’s perceptual grounding. Object perception is a fundamental perceptual primitive that underlies many higher-level tasks, including relative positioning, actions, counting, temporal reasoning, and causal inference. If a model’s visual representation or its use of visual information are impaired by spurious cues, then that bias will naturally propagate into many downstream capabilities.
>
> The goal of our work is to provide a diagnostic foundation for studying spurious bias at-scale in MLLMs. Our experiments and ablations in Sections 6.2 and 6.3 argue that visual spurious bias is due to the vision encoder representation, and cannot be remedied through prompt structure or language-based reasoning, which is common when dealing with higher-level tasks. Recognizing these lower-level biases, and gaining a better understanding of where they originate, is essential for improving at higher-level understanding and more complex tasks.
>
> Finally, while our pipeline focuses on object-level cues, our pipeline is modular and flexible. Our spurious cue proposal and detection could likely be extended beyond associated objects and background elements (as we consider), to colors, texture, or simple visual relationships that are essential in reasoning tasks.

---

> ### Author Response · Authors · 2025-11-23
> **Response to Reviewer yEFM**
>
> ### **Weakness 4.**
>
> We discussed the distribution-shift concern with token dropping in Section 6.1 and in Appendix K. Several recent works, such as [1, 2, 3] utilize visual token dropping for improved efficiency, visual attention, and task performance. While token-dropping can result in out-of-domain evaluations, that does not imply that the Spurious Gap should exist; the fact that it does is still of interest. These experiments are useful ablations and provide further evidence that MLLMs still suffer from hallucination due to visual spurious cues. They allow us to directly contrast the MLLMs performance on the same spurious cue tokens, with and without the target object tokens. These artificial negative examples are drawn from the distribution of images in the dataset containing the target object (which is excised), while the images for standard HR Gap evaluation are images from the dataset not containing the target object. The token-dropping experiment results thereby complement the standard HR results and provide a more holistic view of spurious cue impact over different types of images. Finally, in Appendix K, we perform some simple tests to validate that token-dropping works correctly during inference. For example, masking the object (by zeroing the object pixels, but not dropping them), the MLLM is able to notice that the image has been altered; however, when dropping the object tokens, the MLLM does not notice that the image is missing pieces.
>
> [1] Liu et al., “Multi-Stage Vision Token Dropping: Towards Efficient Multimodal Large Language Model”, arXiv:2411.10803v1
>
> [2] Li et al., “ToDRE: Effective Visual Token Pruning via Token Diversity and Task Relevance”, arXiv:2505.18757
>
> [3] Arif et al., “HiRED: Attention-Guided Token Dropping for Efficient Inference of High-Resolution Vision-Language Models”, AAAI 2025
>
> ---
>
> ### **Weakness 5.**
>
> Our work primarily aims to develop a fully-automated and efficient method of identifying interpretable spurious cues – a novel and significant improvement over previous work, as discussed in the response to Weakness 1. Nevertheless, our experiments, particularly in Section 6, reveal key insights regarding the source of spurious bias; in particular, we argue that it largely stems from the vision encoder image representations, and cannot be remedied through language-based methods. In Section 6.2, we experiment with various prompting techniques, including Chain-of-Thought, ensembling, and informing the MLLM of spurious features (and, in the “Spurious Top” prompt, even “cheating” by utilizing SpurLens’ analysis of the MLLM’s performance on the whole dataset). In all cases, we did not find a significant decrease in either the PA Gap or HR Gap, providing evidence that prompting techniques and text-based guidance alone are unable to alleviate spurious bias. In Section 6.3 and Appendix M, we isolate the MLLM vision encoder and perform experiments similar to our PA evaluation on binary classifiers. We are able to obtain similar results compared to our main MLLM experiments, and argue that spurious bias in MLLMs mainly arises due biases in the vision embeddings, which are influenced by dataset size and composition. Our main pipeline, in addition to these insights, provides a foundation and direction for further investigation leading towards a remedy.
>
> ---
>
> ### **Question 1.**
>
> We have added Appendix O, which discusses using alternative object detectors in our pipeline; we find relative consistency, and find that OWLv2 strikes the best balance between over- and under-confidence. As SpurLens must perform object detection on arbitrary spurious cues, the object detection model must be open-vocabulary (so standard DETR will not suffice); we elect to study GroundingDINO and YOLO World in comparison with OWLv2 (which was used for our main experiments). When applying these detectors to HardImageNet, we empirically observe that GroundingDINO tends to have many false positives (labeled bounding boxes with high confidence scores that are incorrect), while YOLO World tends to have very low confidence scores (mostly near-zero). Both of these can introduce noise and error in object detection, and thereby errors in Spurious Gap computation. OWLv2 tends to lie in-between these extremes, with many near-zero scores and several high-scores for each object/cue. We compute correlations between OWLv2 $f_i\text{-scores}$ and scores from GroundingDINO and YOLO World for many target object / cue pairs, and present correlation plots for two chosen examples. We observe that the correlation tends to be high, though it is class/cue dependent. Thus, our results do slightly vary based on the choice of object detectors, these experiments support our methodology’s soundness, and reaffirm our choice of OWLv2 as our object detection backbone (which is separately verified via human study in Appendix F).

---

> ### Author Response · Authors · 2025-11-23
> **Response to Reviewer yEFM**
>
> ### **Question 2.**
>
> We do not observe any significant cultural or linguistic biases in the generated or chosen spurious cues in our evaluation. In Appendix E, we compare the spurious feature proposal of GPT-4 with other LLMs (Gemini 2.5 Flash and Qwen 2.5 72B). We find significant agreement between the spurious features proposed by these models, suggesting that our pipeline is not sensitive to the choice of model for spurious cue proposal. The evaluations in our main results are performed on HardImageNet, COCO, and a large ImageNet subset, all of whose classes are relatively common objects pertinent to general MLLM use cases. The SpurLens top spurious features for all classes (in both PA and HR settings) are listed in Appendix S; we observe no cultural bias or linguistic trend in the provided cues. Finally, we note that SpurLens is modular: it is relatively easy for a user to modify the cue proposal instructions or filtering procedure to suit their needs.
>
> ---
>
> ### **Question 3.**
>
> We applied SpurLens evaluation on the Waterbirds dataset in Appendix H, and found that all of the open-source MLLMs we evaluated exhibited significant spurious bias. We refer further discussion of the experiments and results to that appendix.
> Regarding MM-SpuBench, their dataset of images is taken from ObjectNet, ImageNet, and several ImageNet variants. The authors do not specify which images are taken from which datasets, so it is difficult to construct a sufficiently large dataset of each target object from its corresponding dataset, which is necessary to apply SpurLens. However, we note that MM-Spubench obtains their provided spurious cues by prompting GPT-4V with the image, target class, and misclassification results of an auxiliary image classifier; yet, these spurious cues are not further checked for causality or robustness. In contrast, while SpurLens similarly uses GPT-4 for spurious cue proposal, we select only the spurious cue with the largest Gap, based on MLLM evaluation on images. Therefore, our final spurious cues have strong empirical evidence backing their causality.
>
> ---
>
> ### **Question 4.**
>
> The cue filtering does eliminate some classes of spurious cues. The types of relationships that are removed from consideration are highlighted through in-context examples in the filtering prompts in Appendix A. For example, cues such as “sunlight” and “walk” are excluded (not easily visualizeable), and spurious cues that are visually part of their target objects are excluded (such as cue “handlebar” for object “bicycle”). Relaxing the filtering may lead to additional, stronger spurious cues being uncovered. However, the filters are necessary in our pipeline to ensure the quality of object detection, as we empirically observe that object detectors struggle with difficult cases such as those previously mentioned. The current set of filters eliminates such errors, and the reliability of OWLv2 for object detection with the filtered set of cues is validated via human study in Appendix F. Finally, we reiterate that SpurLens emphasizes precision rather than recall; we aim to consistently and efficiently find strong spurious cues for (MLLM, target object) pairs, rather than the absolute best cue for any given combination.
>
> ---
>
> ### **Question 5.**
>
> We thank the reviewer for this suggestion. Causal probing and TCAV-style analyses require predefined concepts and measure their directional influence inside the network. However, SpurLens discovers the spurious concepts themselves without any human-defined attributes. TCAV cannot operate without concept sets, whereas SpurLens automatically proposes, filters, detects, and ranks candidate cues across classes and datasets. That said, once SpurLens identifies strong spurious cues, TCAV could be applied downstream to study how those discovered cues are encoded within the vision encoder.

---

### Official Review · Reviewer_wHCG · 2025-11-03

**Soundness:** 3
**Presentation:** 3
**Contribution:** 3
**Rating:** 6
**Confidence:** 4

**Summary:**

The paper proposes a pipeline designed to produce interpretable spuriosity rankings of images to study spurious bias in Multimodal Large Language Models (MLLMs). The pipeline leverages GPT-4 and open-set object detectors to automatically identify spurious visual cues without human supervision. By applying the pipeline to various MLLMs and datasets, the paper reveals that MLLMs often exhibit over-reliance on spurious cues for object recognition and object hallucination. These findings are validated with multiple robustness checks. The paper also explores potential mitigation strategies, such as prompt ensembling and reasoning-based prompting, and shows that spurious bias in MLLMs cannot be trivially overcome through language-based techniques, calling for more rigorous evaluation methods and mitigation strategies to enhance the reliability of MLLMs.

**Strengths:**

- The paper studies spurious bias, an important and fundamental problem in MLLMs, and provides detailed and robust analyses on how spurious correlations influence both object recognition and hallucination in MLLMs.

- The proposed Spurious Gap for object recognition and object hallucination is a useful metric to measure an MLLM's reliance on spurious correlations.

- The experiments are comprehensive, covering object recognition and object hallucination as well as ablation studies on visual cues of the target object, prompting strategies for mitigation, and spuriousity in the vision encoder, demonstrating that spurious bias is a fundamental issue in MLLMs.

**Weaknesses:**

- The paper only analyzes spurious visual cues. In a multimodal setting, spurious correlations may also exist in the text modality. For example, an MLLM may completely ignore the given visual content. I think it is important to emphasize that the paper focuses on spurious visual cues and to provide justifications on why only considering visual features in analyzing spurious bias in MLLMs.

- The spurious bias detection capability of SpurLens is limited to a language model such as GPT-4, as it relies on the language model to generate potential spurious attributes. It may work well for general image datasets where GPT-4 can generate spurious features that indeed commonly appear in images. However, the method may not work well if there is a mismatch between GPT-4 and the dataset being studied, such as a domain-specific dataset. On a similar vein, using an open-set object detector may further limit the spurious bias detection capability.

- The paper assumes that that the image dataset being studied is reasonably large and diverse but does not provide concrete criteria for selecting such datasets. It would be beneficial to provide some practical guidelines to select datasets for effectively analyzing spurious bias in MLLMs.

**Questions:**

- Why does the paper only study spurious visual cues in MLLMs?
- Can the method generalize to domain-specific datasets?
- What are the guidelines to select datasets for an effective analysis of spurious bias in an MLLM?

---

> ### Author Response · Authors · 2025-11-23
> **Response to Reviewer wHCG**
>
> We thank the reviewer for recognizing the importance of studying spurious bias in MLLMs and for highlighting the strength of our analyses, metrics, and experiments across recognition, hallucination, and ablation settings. Below, we address the reviewer’s comments:
>
> ---
>
> ### **Weakness 1 / Question 1.**
> Prior work, such as *Eyes Wide Shut* [1], show that many MLLM failures stem from biases internal to the vision encoder, or biases in vision-language fusion. In particular some works show that MLLMs do not adequately utilize visual information, or that text and vision embeddings are not well-aligned in some cases. Motivated by these works, our work aims to interpretably identify such visual failures in a scalable manner, enabling further study of visual understanding shortcomings of MLLMs. The insufficiency of our language-based mitigation strategies in Section 6.2, and the correlation between Spurious Gaps measured in MLLMs and vision encoder embeddings in Section 6.3, reaffirm our intuition that the vision encoder and vision-language fusion are a significant source of spurious bias in modern MLLMs.
>
> [1] Tong, Shengbang, et al. *"Eyes wide shut? exploring the visual shortcomings of multimodal llms."* Proceedings of the IEEE/CVF Conference on Computer Vision and Pattern Recognition. 2024.
>
> ---
>
> ### **Weakness 2 / Question 2.**
> As discussed at the end of Section 4, SpurLens emphasizes precision rather than completeness. Our goal is not to find the best spurious feature for any given MLLM and target object, but rather to consistently identify strong spurious features, which SpurLens is able to do as demonstrated in our main results. In Appendix E, we demonstrate that feature proposals are consistent across different LLMs, confirming that our method is not tied to the use of GPT-4 for spurious cue proposal.
>
> SpurLens is modular by design. For domain-specific datasets and applications, it is straightforward to leverage corresponding domain-matched models (domain-specific LLMs or fine-tuned vision backbones for the object detector). The spurious cue proposal LLM can also be provided special instructions or additional context to adjust for domain-specific understanding, per the user’s inclination. Our human validation studies in Appendix F examine the reliability of SpurLens (with GPT-4 and OWLv2) for a large class of common objects (classes available in COCO and ImageNet), which covers general-purpose use cases. While altering the models used significantly may require re-validation to ensure reliability, our experiments on large common-object datasets is strong evidence for SpurLens’ robustness in a variety of contexts.
>
> ---

---

> > ### Author Response · Authors · 2025-11-23
> > **Response to Reviewer wHCG**
> >
> > ### **Weakness 3 / Question 3.**
> > We thank the reviewer for their suggestion. Given the range of analyses and insights presented in our paper, it is indeed valuable to formalize what “sufficiently large and diverse” means as a prerequisite to effective spurious bias evaluation with SpurLens. We present a simple thresholding procedure as a heuristic measure of diversity, providing the user of SpurLens with a method to evaluate their dataset’s suitableness before computing PA/HR Gaps with the MLLM.
> >
> > For a spurious feature to be “sufficiently represented” to be evaluated, at minimum there must be $K$ instances of its presence and $K$ instances of its absence, where $K$ is the number of high-spurious and low-spurious instances used for the MLLM evaluation (which also must be determined suitably). While GPT-4 suggests $N$ potential spurious features, not all of these may be “sufficiently represented”, though we would like at least some fixed number $\widetilde N \leq N$ to be.
> >
> > To test this, we leverage the object detection portion of the SpurLens pipeline, which is fast and cheap to run compared to the MLLM evaluation. For a given object detection threshold $\tau$, we test all $N$ proposed spurious features $f_i$ and find the maximum $K$ such that there are at least $K$ images in the dataset with $f_i\text{-scores}$ above $\tau$, and at least $K$ with $f_i\text{-scores}$ below $\tau$. Then, we find the largest $K$ (denoted $K_{\tau, \widetilde N}$) such that at least $\widetilde N$ of the spurious features have $K$ high-spurious and low-spurious examples, according to the threshold $\tau$.
> >
> > The value of $\widetilde N$ must be chosen in advance by the user, according to their own heuristics, understanding of the domain, and computational resources (how many spurious cues they are able to run the MLLM evaluation on, in the hope that at least one is strong). With this fixed, the user can vary $\tau \in [0, 1]$, and find the value that maximizes $K_{\tau, \widetilde N}$. During MLLM evaluation, $K$ controls the precision of the Gap estimate; the user must decide if $\max_\tau K_{\tau, \widetilde N}$ is sufficiently high to evaluate on (given their computational resources). This criteria can be made stricter by, for instance, limiting the range of $\tau$ to a smaller interval (constraining the object detector to be confident in its assessment), or by taking the minimum of $K_{\tau, \widetilde N}$ for multiple target object classes (if the user is planning to run SpurLens for multiple target objects, as we did for COCO and ImageNet classes).
> >
> > This simple method provides the user with appropriate hyperparameter ranges for evaluation under specified compute and confidence thresholds, and enables them to decide if their dataset is sufficient to support MLLM evaluation at their desired accuracy for the proposed spurious features. As a case study, we apply this approach to HardImageNet; the results are available in Appendix P, which reaffirm our choice of hyperparameters used in our main experiments, and acts as a guide for future users of SpurLens.
> >
> > ---

---

### Author Response · Authors · 2025-11-27

We thank all of the reviewers for their insightful comments and constructive criticisms. We appreciate their recognition of the importance of spurious bias analysis in MLLMs, and the design of our fully-automated and scalable approach to spurious bias detection. Below, we highlight the changes made in the paper based on the comments:

**Appendix M (Vision Encoder Experiments):**
This appendix isolates the vision encoder to confirm the problem of spurious correlations. We trained binary linear classifiers on vision encoder features. The spurious biases measured from these linear classifiers are of similar magnitude to those measured from the Qwen2-VL experiments. Furthermore, we added a new experiment: when comparing these gaps class-wise against the PA Gaps obtained when evaluating Qwen2-VL on the HardImageNet rankings, we find a strong correlation. This evidence suggests that the vision encoder is largely responsible for the spurious bias effect.

**Appendix O (Alternative Object Detectors):**
We experimented with SpurLens using other open-set object detectors, namely GroundingDINO and YOLO World. Empirically, we observed that GroundingDINO tends to have many false positives, introducing significant noise, while YOLO World tends to be very conservative, often missing instances of spurious objects. OWLv2 lies in the middle of these extremes, with the best overall performance that we observe empirically. Nevertheless, we found some agreement between *fᵢ*-scores computed with these different systems, and observed that the correlations are generally strong. This provides us with confidence that our pipeline is sound, and that OWLv2 is a strong choice of object detector for SpurLens.

**Appendix P (Dataset Diversity Evaluation):**
This appendix outlines a procedure to examine whether an image dataset is sufficiently diverse to support object detection for spurious feature identification. The method helps determine if the dataset can provide enough instances where a spurious feature is present and absent (at least *K* instances of each). Applying this as a case study to HardImageNet, we find that the dataset is sufficiently diverse, with the minimum required *K* value being fairly high (over 100 in all cases).

These changes directly address the main concerns raised across the reviews, particularly regarding the source of spurious bias, the consistency of results across different object detectors, and the criteria for dataset suitability.

---

### Meta-Review · Area_Chair_tvj3 · 2026-01-05

**Summary:**

This paper proposes an automated pipeline (named SpurLens) utilizing GPT-4 and open-set object detectors (OWLv2) to identify spurious visual cues in Multimodal LLMs (MLLMs). The authors introduce "Perception Accuracy (PA) Gap" and "Hallucination Rate (HR) Gap" to quantify model reliance on these cues. While reviewers acknowledged the utility of a scalable, automated diagnostic tool and the extensive experiments across various models (Qwen, LLaVA, GPT-4o-mini), the consensus leans towards rejection primarily due to concerns regarding incremental novelty and limited scope. Multiple reviewers noted that the method is largely an automation of existing frameworks (like Spurious ImageNet) or a direct extension of a prior workshop paper with identical core methodologies. Furthermore, the analysis is restricted to simple object existence/hallucination, neglecting the complex reasoning and relational capabilities that define modern MLLMs. The dependence on upstream models (GPT-4, OWLv2) also introduces a chain of potential errors that, while validated to some extent, remains a methodological constraint.

**Reviewer Concerns:**

Addressed Concerns:
- Tool Reliability: The authors successfully addressed concerns regarding the reliability of the OWLv2 detector (Reviewer yEFM, DJ5w) by adding a human verification study (Appendix F) and comparing against other detectors like GroundingDINO (Appendix O).
- Dataset Criteria: Reviewer wHCG’s concern about the lack of criteria for "diverse datasets" was addressed by the addition of a heuristic procedure in Appendix P.
- Token Dropping: The authors clarified the implementation of token dropping (preserving positional embeddings) and added comparisons to zero-filling to address Reviewer DJ5w’s technical questions.

Outstanding Concerns:
- Novelty and Incrementalism: This remains the most significant hurdle. Reviewer g5KA explicitly noted that the core method and metrics are nearly identical to the authors' ICLR 2025 Workshop paper, viewing this submission as a "polished version" rather than a substantial leap. Reviewer yEFM also categorized the work as an automation of existing ideas (Spurious ImageNet, RAVL) rather than a conceptually new contribution.
- Scope of Analysis: Reviewers DJ5w and yEFM highlighted that the evaluation is limited to object recognition (presence/absence). This fails to capture the broader spectrum of MLLM capabilities, such as relational reasoning, action understanding, or counting, limiting the insights into "multimodal" spuriousness beyond simple visual co-occurrence.
- Mitigation: The paper acts primarily as a diagnostic tool. The "mitigation" section largely reports negative results (prompting doesn't help), which, while scientifically valid, left reviewers like g5KA and yEFM looking for more constructive solutions or training-time interventions to actually fix the identified biases.

**Reviewer Scores:**

- Reviewer wHCG (Score: 6): Likely would have maintained their score or dropped slightly to a 5 upon realizing the extent of the overlap with the workshop paper, as their review focused heavily on the "soundness" which is somewhat undercut by the incremental nature.
- Reviewer yEFM (Score: 4): Would likely maintain a 4. Their concern about "limited conceptual novelty" and the method being an "automation of existing ideas" is fundamental and was not fully resolved by the additional experiments.
- Reviewer DJ5w (Score: 6): Might have lowered their score to a 5. While satisfied with the technical clarifications on token dropping, the limitation that the method does not apply to complex MLLM tasks (actions/relations) remains a valid structural weakness they identified.
- Reviewer g5KA (Score: 4): Would almost certainly maintain a 4 (or drop lower). The rebuttal confirmed the existence of the prior workshop paper with the same core methodology, validating their primary reason for the low score.

---

### Decision · Program_Chairs · 2026-01-26

Reject